# LoCoBench: A Benchmark for Long-Context LLMs in Complex Software Engineering

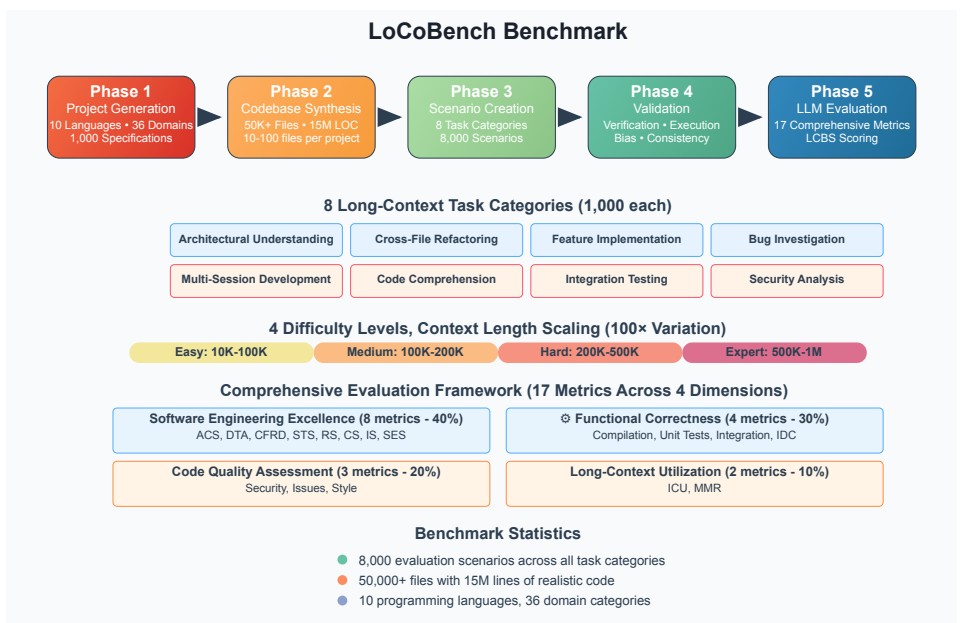

## Abstract

The rise of long-context language models with million-token windows opens new possibilities for advanced code understanding and software development evaluation. We propose LoCoBench, a benchmark designed to assess long-context LLMs on realistic, complex development tasks. Unlike existing benchmarks centered on single-function or short-context tasks, LoCoBench targets capabilities like whole-codebase understanding, cross-file reasoning, and architectural consistency in large systems. It offers 8,000 scenarios across 10 languages, with context lengths from 10K to 1M tokens, enabling precise measurement of long-context performance degradation. LoCoBench spans 8 task categories, architectural understanding, cross-file refactoring, multi-session development, bug investigation, feature implementation, code comprehension, integration testing, and security analysis. Built through a 5-phase pipeline, it produces diverse, high-quality scenarios requiring reasoning over large codebases. We introduce a comprehensive evaluation framework with 17 metrics across 4 dimensions including 6 new evaluation metrics: Architectural Coherence Score, Dependency Traversal Accuracy, Cross-File Reasoning Depth, Incremental Development Capability, Information Coverage Utilization, Multi-Session Memory Retention, combined into a unified LoCoBench Score (LCBS). Evaluations of state-of-the-art models reveal substantial performance gaps, underscoring long-context software development as a critical challenge.

## 1 Introduction

The rise of long-context language models (LLMs) with million-token windows marks a new frontier in software development evaluation. As LLMs advance from code completion tools to systems capable

of reasoning over entire codebases and multi-file workflows, traditional evaluation frameworks are becoming inadequate.

**The Long-Context Revolution in Code.** Recent breakthroughs in long-context LLMs (Reid et al., 2024; Anthropic, 2024) enable them to process full codebases with hundreds of files and complex dependencies, maintaining architectural consistency across large systems. Yet, long-context remains a critical weakness: LongCodeBench (Rando et al., 2025) reports performance drops from 29% to 3% on Claude 3.5 Sonnet as context grows, while RULER (Hsieh et al., 2024) finds that only half of models claiming 32K+ context sustain acceptable accuracy.

**The Long-Context Capability Gap.** Existing code benchmarks target single-function generation (Chen et al., 2021; Austin et al., 2021) or repository-level understanding (Jimenez et al., 2023; Liu et al., 2023b), but not the *long-context capabilities* needed for realistic software development. Such tasks demand architectural reasoning, cross-file refactoring, and coordination across dozens of files, which is far beyond traditional code completion.

**The Evaluation Challenge.** Current benchmarks have three key gaps:

❶ *Scale:* Most offer fewer than 3K instances (Jimenez et al., 2023; Hendrycks et al., 2021), limiting coverage across languages, complexity, and long-context tasks.

❷ *Context:* They use short contexts (under 10K tokens), failing to test real-world codebase sizes. Even ∞-Bench (Zhang et al., 2024a) and LongBench (Bai et al., 2024b) target document comprehension, not complex code reasoning.

❸ *Task Scope:* They emphasize isolated generation or bug fixing, neglecting architectural understanding and multi-file workflows.

To close these gaps, we introduce **LoCoBench**, a benchmark purpose-built to evaluate long-context understanding in complex software development scenarios.

- **Systematic Long-Context Evaluation:** LoCoBench offers 8,000 scenarios spanning 10K–1M tokens, enabling precise analysis of long-context performance degradation in realistic development settings.
- **Comprehensive Task Coverage:** It includes 8 task types capturing core long-context skills: architectural reasoning, cross-file refactoring, multi-session development, bug fixing, feature implementation, comprehension, integration testing, and security analysis.
- **New Metrics:** We introduce 17 metrics across 4 dimensions, including 6 novel ones for long-context evaluation, aggregated into a unified LoCoBench Score (LCBS).
- **Scale and Diversity:** Covering 10 languages and 36 domains, LoCoBench surpasses existing benchmarks in scale while preserving realistic complexity and difficulty distributions.

Evaluating state-of-the-art models on LoCoBench reveals large performance gaps, underscoring that long-context understanding in complex software development remains an unsolved challenge requiring new benchmarks and models.

## 2 RELATED WORK

**Code Generation Benchmarks** Traditional code benchmarks focus on narrow programming tasks. Function-level datasets like HumanEval (Chen et al., 2021) and MBPP (Austin et al., 2021) established foundational evaluation, with extensions including HumanEval+ (Liu et al., 2023a), MultiPL-E (Cassano et al., 2023), and BigCodeBench (Zhuo et al., 2024). Contest-style benchmarks (APPS (Hendrycks et al., 2021), LiveCodeBench (Jain et al., 2024), CodeContests (Li et al., 2022)) test algorithmic problem-solving but not software engineering aspects like design or multi-file workflows. Recent long-context code benchmarks include LongCodeBench (Rando et al., 2025), which shows severe performance drops with longer contexts. LongCodeU (Li et al., 2025) and LongCodeArena (Bogomolov et al., 2024) emphasize code completion rather than full-system development. Domain-specific (Lai et al., 2022; Thakur et al., 2023; Wang et al., 2022; Dong et al., 2024a; Du et al., 2023) and repository-level benchmarks (Liu et al., 2023b; Ding et al., 2023) move toward realism but remain limited in scope.

**Software Engineering Benchmarks** SWE-Bench (Jimenez et al., 2023) uses real GitHub issues, with extensions like SWE-rebench (rebench Team, 2025), LiveSWEBench (Team, 2024), and Multi-

SWE-Bench (Zan et al., 2025) adding multi-language coverage. However, these focus on bug fixes rather than full development workflows. DevBench (Li et al., 2024) spans the software lifecycle but lacks long-context assessment. CodeXGLUE (Lu et al., 2021) covers code understanding but not development workflows.

**Long-Context Evaluation** General long-context benchmarks include LongBench (Bai et al., 2024b), RULER (Hsieh et al., 2024), and $\infty$-Bench (Zhang et al., 2024a), among others (Yen et al., 2024; Lee et al., 2024; An et al., 2024; Bai et al., 2024a; Dong et al., 2024b). Code-specific ones include LongCodeBench (Rando et al., 2025), LongCodeU (Li et al., 2025), LongCodeArena (Bogomolov et al., 2024), and RepoQA (Liu et al., 2024), but they focus on code completion rather than complex multi-file development.

**Limitations and Contributions** Appendix D provides detailed discussion. Briefly, current benchmarks face four issues: (1) **Scale:** typically <1,000 instances, (2) **Task Scope:** focus on isolated generation vs. multi-session development, (3) **Context:** use <10K tokens, and (4) **Metrics:** emphasize correctness over long-context abilities like architectural coherence. LoCoBench addresses these gaps with 8,000 scenarios (10K–1M tokens), diverse long-context tasks, and new metrics for complex software development.

## 3 LoCoBench Benchmark

**Benchmark Design Principles** LoCoBench follows four principles that set it apart from prior code benchmarks: ❶ Long-Context Tasks: It targets real-world software development scenarios requiring large-codebase understanding, complex dependencies, and multi-file consistency across sessions. ❷ Systematic Scale: We generate 8,000 scenarios via a 5-phase pipeline ensuring broad coverage across languages, difficulty levels, and task types while preserving quality and diversity. ❸ Long-Context Focus: Scenarios range from 10K to 1M tokens, testing models on realistic codebase sizes beyond traditional benchmarks. ❹ Comprehensive Metrics: Beyond correctness, we add metrics for architectural reasoning, cross-file understanding, and multi-session memory retention. Figure shows the full LoCoBench pipeline, from project specs to validated scenarios, including data processing, LLM integration, and quality checks. More details can be found in Appendix A.

### 3.1 Five-Phase Pipeline

LoCoBench is built through a systematic 5-phase pipeline to produce high-quality, diverse evaluation scenarios at scale:

**Phase 1: Project Specification Generation** We create 1,000 project specs across 10 languages (100 each), covering 36 domains from web apps to ML systems. Each spec defines realistic requirements, constraints, and architectural patterns, spanning from small apps to enterprise-scale systems.

**Phase 2: Codebase Generation** Each spec becomes a complete codebase (10–100 files) with coherent architecture, dependency management, documentation, and realistic patterns. Codebases pass automated quality checks for compilation, complexity, and structural consistency.

**Phase 3: Evaluation Scenario Creation** Each codebase yields 8 task-specific scenarios (8,000 total). We select file subsets to provide sufficient context while targeting specific long-context skills. Selection uses dependency graphs, architectural centrality, and task relevance to balance information density, difficulty, and workflow realism.

**Phase 4: Validation and Quality Assurance** Scenarios undergo automated validation, compilation, test execution, complexity scoring, and difficulty calibration, to ensure executability and remove generation artifacts.

**Phase 5: LLM Evaluation and Scoring** Models are evaluated on 17 metrics in 4 dimensions: *Software Engineering Excellence* (8 metrics), *Functional Correctness* (4), *Code Quality* (3), and *Long-Context Utilization* (2). These combine into the **LoCoBench Score (LCBS)** with weights: 40% Engineering, 30% Correctness, 20% Quality, 10% Long-Context.

**Quality Assurance and Validation** Every generated scenario undergoes rigorous quality assurance: ❶ **Automated Validation:** All code is validated for compilation, execution, and basic functionality through automated testing pipelines using language-specific compilers (gcc, javac, python, etc.) and testing frameworks. ❷ **Complexity Metrics:** We employ cyclomatic complexity analysis,

dependency depth measurement, and architectural coherence scoring to ensure appropriate difficulty calibration. Scenarios are automatically filtered if complexity metrics fall outside target ranges for their difficulty level. ❸ **Information Coverage:** Each scenario's information coverage ratio is calculated to ensure sufficient context for task completion while avoiding information redundancy. We target coverage ratios ¿0.7 for all scenarios. ❹ **Bias Detection:** Automated analysis identifies and filters scenarios with potential biases, generation artifacts, or unrealistic patterns that could skew evaluation results. This includes detection of repeated code patterns, unrealistic naming conventions, and generation-specific artifacts.

### 3.2 BENCHMARK STATISTICS AND SCALE

LoCoBench represents the largest and most comprehensive evaluation framework for long-context software development to date. Our systematic generation approach produces unprecedented scale and diversity: ❶ 8,000 evaluation scenarios across 8 task categories. ❷ 1,000 synthetic projects spanning 36 domain categories. ❸ 10 programming languages with balanced coverage. ❹ Context range from 10K to 1M tokens (100× variation). ❺ 50,000+ generated files with realistic code patterns. ❻ Systematic difficulty distribution across 4 complexity levels.

**Task Categories** LoCoBench evaluates eight distinct task categories that capture essential long-context software development capabilities: architectural reasoning, cross-file refactoring, multi-session development, bug fixing, feature implementation, comprehension, integration testing, and security analysis. More details can be found in Appendix A.

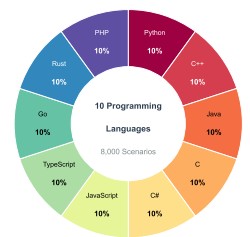

Figure 1: Programming language distribution.

**Difficulty Calibration and Context Scaling** Our benchmark systematically varies difficulty across four levels (easy, medium, hard, expert) with corresponding context length ranges: ❶ Easy (10K-100K tokens): Basic long-context tasks with small to medium codebases. ❷ Medium (100K-200K tokens): Intermediate complexity with larger codebases. ❸ Hard (200K-500K tokens): Advanced scenarios with enterprise-scale codebases. ❹ Expert (500K-1M tokens): Maximum complexity with massive enterprise systems.

**Language Distribution:** Our benchmark provides balanced coverage across diverse programming paradigms with each language contributing equally (10%) to our 8,000 scenarios. Languages span from systems programming (C, C++, Rust) to web development (JavaScript, TypeScript, PHP), enterprise applications (Java, C#), and modern data science/AI frameworks (Python, Go). This equal distribution ensures comprehensive evaluation across different language characteristics while avoiding bias toward any particular programming paradigm. See Table 4 and Figure 1.

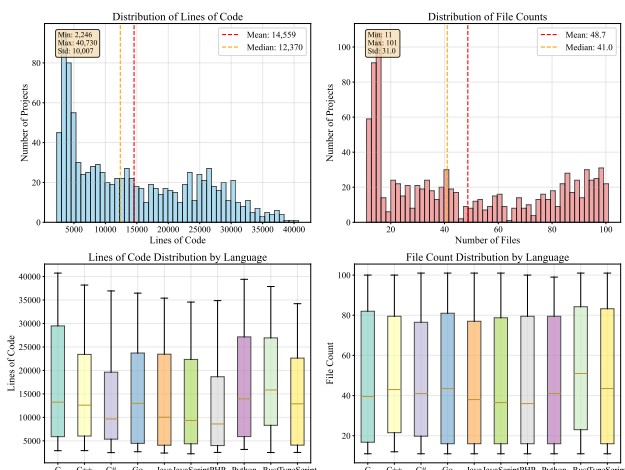

Figure 2: **Top:** distribution of lines of code and file counts. **Bottom:** code distribution (and file count distribution across 10 programming languages.

**Domain Coverage:** Projects span 36 distinct domains including web applications (ecommerce, social, dashboard, blog, CMS, portfolio), machine learning systems (training, inference, computer vision, NLP), data processing (analytics, ETL, streaming, warehousing), system utilities (networking, security, monitoring, automation), APIs (REST, GraphQL, microservices, gateway), financial technology (banking, payments, trading), gaming (engine, simulation), blockchain (DeFi, NFT), and mobile applications (utility, social, gaming). See Table 5 and Figure 3.

**Complexity Metrics:** Our generated codebases exhibit realistic complexity distributions with cyclomatic complexity scores ranging from 0.3 to 1.0, file counts between 10-100 per project, and documentation ratios exceeding industry standards. Automated validation ensures all code compiles successfully and maintains architectural coherence. See Table 6 and Figure 12.

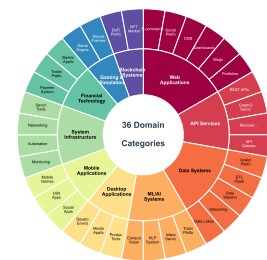

**Line of Code:** Figure 2 presents the statistical characteristics of LoCoBench's evaluation projects, revealing realistic complexity distributions with a mean of 14,559 lines of code and 48.7 files per project. The right-skewed distributions (top row) mirror real-world software patterns, ranging from compact applications to enterprise-scale systems with over 40,000 lines. The language-specific analysis

Figure 3: 10 main categories with 36 sub-categories.

(bottom row) shows distinct patterns: systems languages (C, Rust) exhibit compact implementations, object-oriented languages (Java, C#) demonstrate higher complexity with extensive file structures, while web languages (JavaScript, TypeScript, PHP) show intermediate levels. These patterns validate LoCoBench's realistic representation across programming paradigms and complexity levels.

## 3.3 COMPARISON WITH EXISTING BENCHMARKS

Table 1: Comparison with existing benchmarks. Columns: Scale - Number of evaluation instances; Languages - Programming language coverage; Context Range - Token length ranges; Task Types - Types of programming tasks; Multi-File - Support for multi-file scenarios; Architecture - Architectural understanding evaluation; New Metrics - New evaluation metrics introduced; Real-World - Real-world applicability. Color-coded symbols: Green checkmark (✓) for full support, orange triangle (▶) for partial support, red X (✗) for no support.

| Benchmark | Scale | Languages | Context Range | Task Types | Multi-File | Architecture | New Metrics | Real-World |
|-----------|-------|-----------|---------------|------------|------------|--------------|-------------|------------|
| HumanEval | 164 | 1 (Python) | Short (<10K) | Algorithm | ✗ | ✗ | ✗ | ✗ |
| SWE-Bench | 2,294 | 1 (Python) | Medium (10-50K) | Bug Fix Only | ▶ | ✗ | ✗ | ✓ |
| Multi-SWE-Bench | 1,632 | 7 | Medium (10-50K) | Bug Fix Only | ▶ | ✗ | ✗ | ✓ |
| LongCodeBench | 600+ | 1 (Python) | Up to 1M | Completion | ▶ | ✗ | ✗ | ▶ |
| LongCodeArena | 1,500+ | Multiple | Up to 2M | Completion | ✓ | ✗ | ✗ | ▶ |
| DevBench | 200+ | 4 | Short (<10K) | Mixed | ▶ | ✗ | ✗ | ▶ |
| RULER | 4,000+ | N/A | Up to 128K | NLP Tasks | ✗ | ✗ | ✗ | ✗ |
| **LoCoBench** | **8,000** | **10** | **10K-1M** | **8 Categories** | ✓ | ✓ | **6 Metrics** | ✓ |

LoCoBench addresses critical limitations in existing code evaluation benchmarks through systematic design choices and comprehensive scope. Table 1 provides a comprehensive quantitative comparison highlighting these distinctive features. While SWE-Bench (Jimenez et al., 2023) pioneered real-world evaluation using GitHub issues, it remains constrained to Python-only repositories and focuses exclusively on bug-fixing tasks. The benchmark's 2,294 instances provide limited coverage across programming paradigms and development scenarios, failing to capture the diversity of modern software engineering practices. LongCodeBench (Rando et al., 2025) introduced long-context evaluation for code but primarily emphasizes code completion and comprehension tasks rather than complex software development workflows. Its focus on single-language evaluation and limited task diversity restricts its ability to assess architectural understanding and multi-file reasoning capabilities essential for enterprise software development. Despite supporting multiple languages, LongCodeArena (Bogomolov et al., 2024) concentrates on repository-level code completion rather than comprehensive development scenarios. The benchmark lacks systematic evaluation of architectural coherence, cross-file refactoring, and multi-session development workflows that characterize real-world software engineering. RULER (Hsieh et al., 2024) provides valuable long-context evaluation but employs synthetic tasks primarily for natural language processing. Its evaluation paradigm does not capture the unique challenges of software development, including dependency management, architectural consistency, and code quality assessment.

**Evaluation Metric** LoCoBench introduces a comprehensive evaluation framework with 17 metrics across 4 dimensions designed to assess capabilities essential for realistic long-context software development scenarios. Our framework combines 6 new metrics specifically designed for long-context LLM evaluation with 11 established metrics adapted from software engineering literature. Table 2 provides a comprehensive overview of all 17 metrics organized by evaluation dimensions. More details about each metric can be found in Appendix B.

Table 2: Complete overview of LoCoBench's 17 evaluation metrics across 4 dimensions.

| Dimension | Metric | Abbr. | Source |
|---|---|---|---|
| Software Engineering Excellence (8) | Architectural Coherence Score | ACS | ★ New |
| | Dependency Traversal Accuracy | DTA | ★ New |
| | Cross-File Reasoning Depth | CFRD | ★ New |
| | System Thinking Score | STS | (Blanchard & Fabrycky, 2016) |
| | Robustness Score | RS | (iso, 2011) |
| | Comprehensiveness Score | CS | (Kan, 2002) |
| | Innovation Score | IS | (Glass, 2002) |
| | Solution Elegance Score | SES | (Buse & Weimer, 2010) |
| Functional Correctness (4) | Code Compilation Success | CCS | (McCabe, 1976) |
| | Unit Test Performance | UTP | (Myers et al., 2011) |
| | Integration Test Performance | ITP | (Binder, 1999) |
| | Incremental Development Capability | IDC | ★ New |
| Code Quality Assessment (3) | Security Analysis Score | SAS | (OWASP, 2021) |
| | Average Issues Found (Inverted) | AIF | (Campbell & Papapetrou, 2013) |
| | Code Style Adherence | CSA | (Kernighan & Pike, 1999) |
| Long-Context Utilization (2) | Information Coverage Utilization | ICU | ★ New |
| | Multi-Session Memory Retention | MMR | ★ New |

## 4 EXPERIMENTS, RESULTS AND DISCUSSIONS

Due to page limit, we only list some results and discussion here. For a complete discussion, please see Appendix C.

**Overall Model Performance Analysis** Figure 4 compares three leading LLMs across 10 dimensions. Gemini-2.5-Pro leads overall, excelling in cross-file refactoring, long-context use, integration tests, and multi-session development, —tasks demanding deep system-level reasoning. GPT-5 shows strong, balanced performance, standing out in architectural understanding and system design analysis. Claude-Sonnet-4 performs best in code comprehension, indicating strength in analyzing existing codebases.

Despite general parity across most areas, notable gaps appear in long-context utilization and specialized tasks. This suggests current top models share similar baseline capabilities, diverging mainly in domain-specific reasoning. These differences likely reflect optimization for distinct development workflows, and imply that model choice should

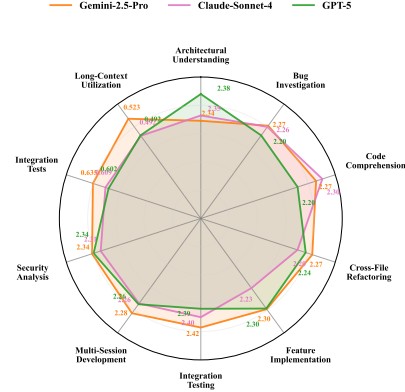

Figure 4: Overall performance.

depend on an organization's specific long-context needs. The tight performance clustering also hints that further gains may require new approaches to context management and multi-file reasoning, rather than incremental tweaks to current architectures.

### 4.1 COMPREHENSIVE MODEL RANKING ANALYSIS

Figure 5 shows model rankings on overall software engineering competency (LCBS) and long-context utilization. The left chart reveals tight clustering among top-tier models, with Gemini-2.5-Pro leading overall, suggesting current models have plateaued in general software engineering skills, showing only incremental gains.

The right chart highlights long-context capabilities, where Gemini-2.5-Flash stands out, indicating specialized optimization for extended

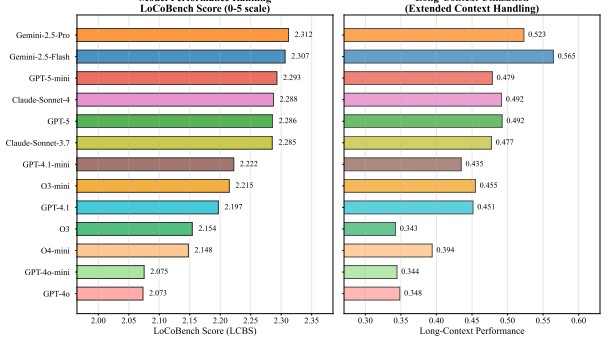

Figure 5: Left: LCBS rankings. Right: long-context utilization performance.

contexts but potentially at the cost of broader skills. The wider gaps here underscore that handling long contexts remains a challenging area requiring distinct architectural strategies.

These contrasting patterns show that models often trade off between general competency and long-context specialization, reflecting different design priorities. Similar overall scores can mask large differences in specific capabilities, making multi-dimensional evaluation essential for selecting models suited to particular long-context development needs. The rankings also reveal clear performance tiers, with top models clustered closely and lower tiers showing larger gaps.

## 4.2 PROGRAMMING LANGUAGE PERFORMANCE ANALYSIS

Figure 6 shows model performance across 10 programming languages, revealing clear language-specific patterns. Models perform best on high-level languages like Python and PHP, moderately on JavaScript and TypeScript, and struggle most with systems languages such as C and Rust, reflecting both language complexity and differences in training data availability.

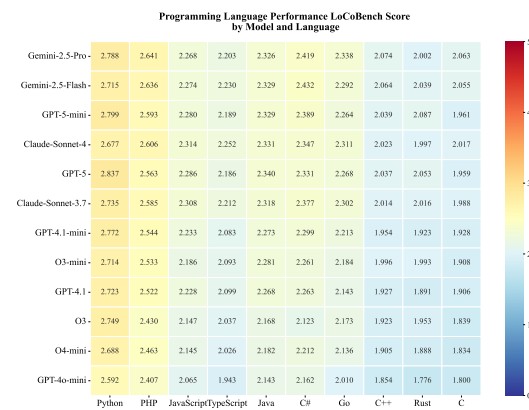

Figure 6: Programming language performance heatmap.

Most models follow this difficulty gradient, suggesting that language popularity and representation strongly influence training outcomes. Some models show notable strengths in specific languages, indicating that architectural or training choices can favor particular paradigms.

These patterns have practical implications: organizations may benefit from choosing models aligned with their primary language ecosystem. The consistently weaker performance on systems languages highlights current LLMs' difficulties with low-level, memory-aware reasoning, pointing to a key challenge for future long-context model development.

## 4.3 TASK CATEGORY PERFORMANCE AND DIFFICULTY ANALYSIS

Figure 7 shows model performance across eight task categories, highlighting both task difficulty and model-specific strengths. The top chart reveals performance distributions, with wide spreads in some tasks (e.g., bug investigation, multi-session development) indicating high variability, and narrower spreads (e.g., integration testing, architectural understanding) showing more consistent performance.

The bottom chart relates task difficulty (score variance) to model performance trends, showing that tasks with larger performance gaps are more sensitive to model design or training. Some models excel in specific categories while lagging in others, reflecting specialization shaped by architecture or training data.

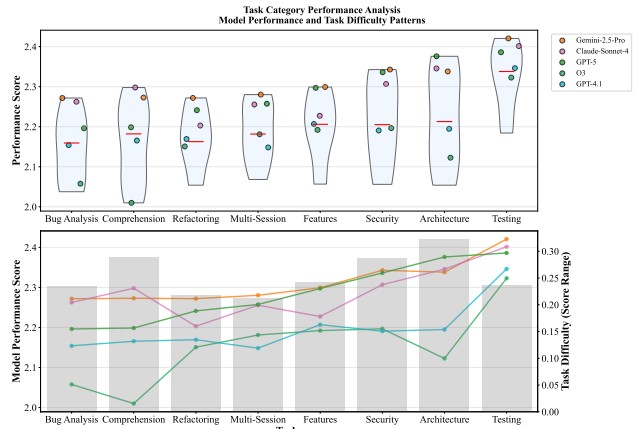

Figure 7: Task category performance analysis. Top: performance distribution across all models for each task category, with individual model performance overlaid. Bottom: task difficulty patterns and model performance trends across different tasks.

These results suggest that certain software engineering tasks are fundamentally harder for long-context LLMs, and that model choice should consider both overall ability and consistency for specific task types. This analysis helps guide both model development and practical model selection for long-context software workflows.

## 4.4 CONTEXT LENGTH AND DIFFICULTY IMPACT ANALYSIS

Figure 8 analyzes how context length and task difficulty jointly affect model performance. The top-left chart shows performance declining as tasks grow harder and require longer contexts, revealing compounding challenges from difficulty and context scaling. The top-right chart compares models, showing that some degrade sharply with increasing difficulty while others maintain steadier performance, reflecting different architectural trade-offs.

The bottom-left chart maps consistency versus specialization, showing that some models perform stably across difficulty levels while others excel in

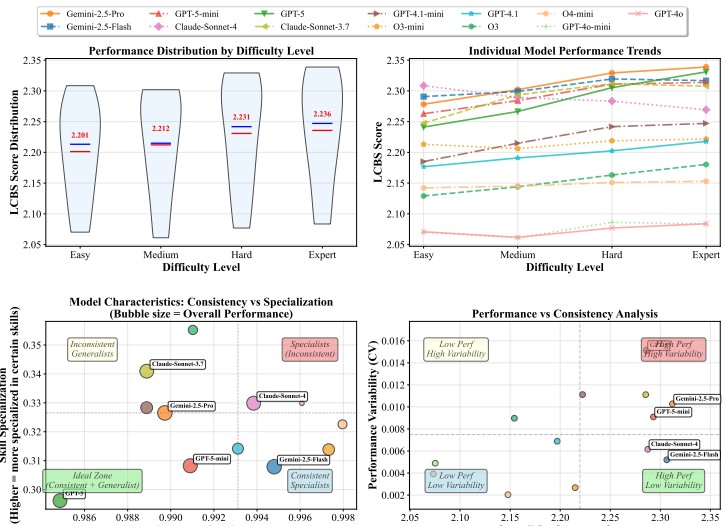

Figure 8: Context length and difficulty impact analysis. Upper left: performance distribution by difficulty level. Upper right: individual model performance trends across difficulty levels. Lower left: model consistency versus specialization patterns. Lower right: performance versus consistency relationships.

specific ranges but vary more overall. The bottom-right chart links consistency to overall scores, revealing that top-performing models are not always the most consistent.

Overall, the analysis shows that context length and difficulty interact to shape distinct model behaviors: some prioritize peak performance while others emphasize stability. These findings highlight the need to evaluate models beyond aggregate scores, considering consistency, specialization, and robustness under increasing challenge levels, which are crucial factors for real-world long-context deployments.

## 4.5 DOMAIN SPECIALIZATION AND PERFORMANCE ANALYSIS

Figure 9 analyzes model performance across 10 application domains, revealing specialization patterns, difficulty hierarchies, and consistency trends. The top chart shows domain-wise performance trajectories, highlighting that some models maintain stable performance across domains while others vary widely, indicating strong domain specialization effects shaped by training data and architectural choices.

The lower-left chart maps relative domain difficulty,

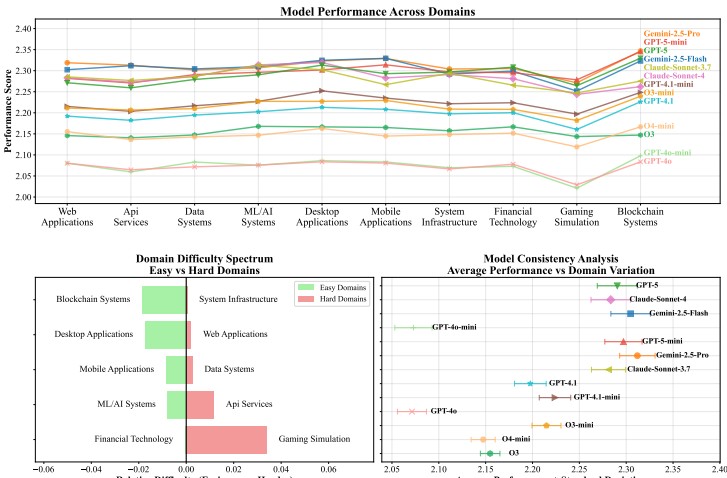

Figure 9: Domain specialization and performance analysis.

showing that areas like Gaming Simulation and API Services are consistently harder, while Blockchain Systems and Desktop Applications yield higher scores. This reflects differing complexity and knowledge demands across software contexts. The lower-right chart examines performance consistency,

showing that some models deliver stable results across domains while others achieve higher peaks but with greater variability, which is critical for deployments needing predictable performance.

Overall, the analysis shows that model choice should consider both overall capability and domain fit. Some models prioritize broad generalization, while others excel in specific domains, underscoring trade-offs between specialization and general-purpose robustness for long-context software development.

### 4.6 Architecture Pattern Performance Analysis

Figure 10 analyzes model performance across 10 architectural patterns, highlighting how software design affects long-context capabilities. The top chart shows model trajectories across patterns, revealing that some models perform consistently while others vary with architectural complexity, indicating that certain designs pose greater reasoning challenges.

The lower-left chart links architectural complexity to performance, showing that complex patterns do not always reduce performance, while simpler patterns can

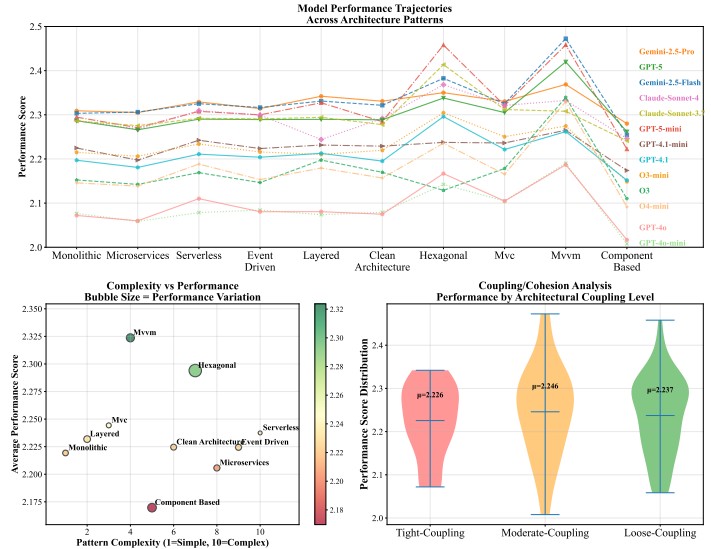

Figure 10: Architecture pattern performance analysis.

exhibit higher variability across models. The lower-right chart examines coupling, showing how tight, moderate, and loose coupling affects model reasoning about interdependencies and system organization.

Overall, performance varies by architectural pattern, suggesting that model selection should consider both domain and design approach. Architectural decisions in software projects should account for the strengths and limitations of long-context models to optimize development and maintenance outcomes.

## 5 Conclusion

We present LoCoBench, a benchmark designed to evaluate long-context language models on complex software development tasks, addressing gaps in traditional code evaluation. Our 5-phase pipeline generates 8,000 scenarios across 10 languages, with context lengths from 10K to 1M tokens, and introduces 6 new long-context metrics, including Architectural Coherence Score, Dependency Traversal Accuracy, Cross-File Reasoning Depth, Incremental Development Capability, Information Coverage Utilization, Multi-Session Memory Retention,, within a 17-metric framework spanning four evaluation dimensions.

Evaluation of state-of-the-art models reveals significant performance variations and specialization patterns. Gemini-2.5-Pro leads overall, excelling in cross-file refactoring and long-context utilization, while GPT-5 shows superior architectural understanding. Performance varies across languages, domains, task categories, and architectural patterns, highlighting the importance of domain- and context-aware model selection. Systems programming and expert-level scenarios remain challenging, with context length and task difficulty compounding performance declines.

LoCoBench offers guidance for both model developers and practitioners, emphasizing multi-dimensional evaluation and careful consideration of domain, architecture, and consistency requirements. It provides findingds for future research on specialized training, architecture design, and interactive workflows, establishing rigorous standards for assessing long-context coding capabilities in AI-assisted software development.

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

## A  MORE DETAILS ABOUT LOCOBENCH

### A.1  BENCHMARK DESIGN PRINCIPLES

LoCoBench follows four principles that set it apart from prior code benchmarks:

Long-Context Tasks: It targets real-world software development scenarios requiring large-codebase understanding, complex dependencies, and multi-file consistency across sessions.

Systematic Scale: We generate 8,000 scenarios via a 5-phase pipeline ensuring broad coverage across languages, difficulty levels, and task types while preserving quality and diversity.

Long-Context Focus: Scenarios range from 10K to 1M tokens, testing models on realistic codebase sizes beyond traditional benchmarks.

Comprehensive Metrics: Beyond correctness, we add metrics for architectural reasoning, cross-file understanding, and multi-session memory retention.

Figure  shows the full LoCoBench pipeline, from project specs to validated scenarios, including data processing, LLM integration, and quality checks.

### A.2  FIVE-PHASE PIPELINE

Our benchmark generation follows a systematic 5-phase pipeline designed to create high-quality, diverse evaluation scenarios at scale:

**Phase 1: Project Specification Generation** We generate 1,000 diverse project specifications across 10 programming languages (100 per language). Each specification defines a complete software project with realistic requirements, technical constraints, and architectural patterns. Projects span 36 domain categories including web applications, machine learning systems, data processing pipelines, and system utilities, with complexity levels ranging from simple applications to enterprise-scale systems.

**Phase 2: Codebase Generation** For each project specification, we generate complete, realistic codebases containing 10-100 files per project. This phase creates architecturally coherent codebases that include proper module structure, dependency management, documentation, and realistic code patterns. Generated codebases undergo automated quality validation including compilation checks, complexity analysis, and architectural consistency verification.

**Phase 3: Evaluation Scenario Creation** We transform each codebase into 8 evaluation scenarios (1 per task category), resulting in 8,000 total scenarios. Each scenario includes carefully selected file subsets that provide sufficient context while targeting specific long-context capabilities. Context selection employs intelligent algorithms that balance information coverage, difficulty calibration, and realistic development workflows. Our selection algorithm prioritizes files based on dependency graphs, architectural centrality, and task-specific relevance, ensuring scenarios contain the minimum necessary context while maximizing information density and maintaining realistic development patterns.

**Phase 4: Validation and Quality Assurance** All generated scenarios undergo comprehensive validation including compilation verification, test execution, complexity scoring, and difficulty calibration. This phase ensures that scenarios are executable, appropriately challenging, and free from generation artifacts that could bias evaluation results. Validation is purely automated using compilation, testing, and metrics, no LLM involvement to prevent bias.

**Phase 5: LLM Evaluation and Scoring** We evaluate state-of-the-art models using our comprehensive 17-metric framework across 4 dimensions: Software Engineering Excellence (8 metrics), Functional Correctness (4 metrics), Code Quality Assessment (3 metrics), and Long-Context Utilization (2 metrics). The Software Engineering Excellence dimension includes Architectural Coherence Score (ACS), Dependency Traversal Accuracy (DTA), Cross-File Reasoning Depth (CFRD), System Thinking Score (STS), Robustness Score (RS), Comprehensiveness Score (CS), Innovation Score (IS), and Solution Elegance Score (SES). Functional Correctness comprises Compilation Success, Unit Test Performance, Integration Test Performance, and Incremental Development Capability (IDC). Code Quality Assessment includes Security Analysis Score, Average Issues Found (inverted),

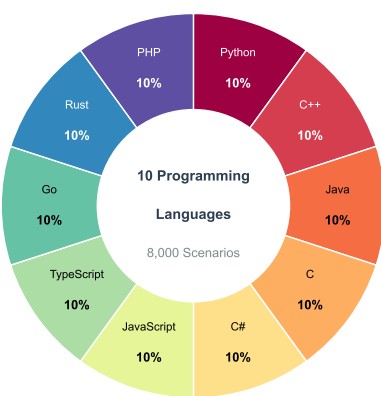 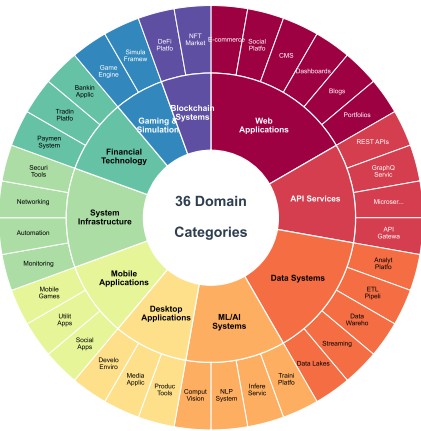

Figure 11: LoCoBench Coverage Overview. **Left:** Programming language distribution showing equal representation (10% each) across 10 languages spanning diverse paradigms from systems programming (C, C++, Rust) to web development (JavaScript, TypeScript, PHP) to enterprise applications (Java, C#) to modern languages (Go, Python). **Right:** Hierarchical domain organization with 36 sub-categories grouped into 10 main categories, ensuring comprehensive coverage across web applications, API services, data systems, ML/AI systems, desktop applications, mobile applications, system infrastructure, financial technology, gaming & simulation, and blockchain systems.

and Code Style Adherence. Long-Context Utilization features Information Coverage Utilization (ICU) and Multi-Session Memory Retention (MMR). These metrics are combined into a **LoCoBench Score (LCBS)** using weighted components: Software Engineering Excellence (40%), Functional Correctness (30%), Code Quality Assessment (20%), and Long-Context Utilization (10%).

## A.3 TASK CATEGORIES AND LONG-CONTEXT CAPABILITIES

LoCoBench evaluates eight distinct task categories that capture essential long-context software development capabilities:

- *Architectural Understanding:* Scenarios that require LLMs to comprehend complex system designs, identify architectural patterns, and understand component relationships across large codebases.

- *Cross-File Refactoring:* Tasks involving code restructuring across multiple files while maintaining functionality and preserving architectural constraints.

- *Feature Implementation:* Complex feature development scenarios that require understanding existing code, planning implementation strategies, and integrating new functionality seamlessly.

- *Bug Investigation:* Systematic debugging tasks that require analyzing error patterns, tracing execution flows, and identifying root causes across multi-file systems.

- *Multi-Session Development:* Scenarios that test long-term memory and context retention across multiple development sessions, simulating realistic project workflows.

- *Code Comprehension:* Tasks focused on understanding large, complex codebases and extracting relevant information for development decisions.

- *Integration Testing:* Scenarios involving testing component interactions, validating system integration, and ensuring end-to-end functionality.

- *Security Analysis:* Tasks requiring identification of security vulnerabilities, assessment of threat vectors, and implementation of security best practices.

Table 3: 8 task categories and details.

| Domains | Details |
| --- | --- |
| Architectural Understanding | Design pattern recognition, dependency analysis, System design comprehension, component relationships across large codebases |
| Cross-File Refactoring | Multi-file restructuring and pattern application, Code restructuring across multiple files while maintaining functionality |
| Feature Implementation | Complex feature development in existing systems, Understanding existing code, planning implementation strategies, seamless integration |
| Bug Investigation | Systematic debugging across complex codebases, Error pattern analysis, execution flow tracing, root cause identification |
| Multi-Session Development | Context persistence across development sessions, Long-term memory and incremental building, simulating realistic project workflows |
| Code Comprehension | Large codebase understanding and explanation, Information extraction for development decisions, deep codebase analysis |
| Integration Testing | System-level testing and validation, Component interaction testing, end-to-end functionality validation |
| Security Analysis | Security vulnerability assessment, Threat vector identification, security best practices implementation |

## A.4 BENCHMARK STATISTICS AND SCALE

LoCoBench represents the largest and most comprehensive evaluation framework for long-context software development to date. Our systematic generation approach produces unprecedented scale and diversity: ❶ 8,000 evaluation scenarios across 8 task categories. ❷ 1,000 synthetic projects spanning 36 domain categories. ❸ 10 programming languages with balanced coverage. ❹ Context range from 10K to 1M tokens (100× variation). ❺ 50,000+ generated files with realistic code patterns. ❻ Systematic difficulty distribution across 4 complexity levels.

**Language Distribution:** Our benchmark provides balanced coverage across diverse programming paradigms with each language contributing equally (10%) to our 8,000 scenarios. Languages span from systems programming (C, C++, Rust) to web development (JavaScript, TypeScript, PHP), enterprise applications (Java, C#), and modern data science/AI frameworks (Python, Go). This equal distribution ensures comprehensive evaluation across different language characteristics while avoiding bias toward any particular programming paradigm.

Table 4: 10 Programming Languages with example usage cases.

| Programming Language | Usage Cases |
| --- | --- |
| Python | AI/ML dominance, automation, data science |
| C++ | High-performance, games, embedded systems |
| Java | Enterprise, Android, backend services |
| C | Systems programming, OS development, embedded |
| C# | Enterprise, Windows, .NET ecosystem |
| JavaScript | Web development, full-stack |
| TypeScript | Enterprise web, type safety |
| Go | Cloud-native, microservices |
| Rust | Systems, security, memory safety |
| PHP | Web backends, legacy systems |

**Domain Coverage:** Projects span 36 distinct domains including web applications (ecommerce, social, dashboard, blog, CMS, portfolio), machine learning systems (training, inference, computer vision, NLP), data processing (analytics, ETL, streaming, warehousing), system utilities (networking, security, monitoring, automation), APIs (REST, GraphQL, microservices, gateway), financial technology (banking, payments, trading), gaming (engine, simulation), blockchain (DeFi, NFT), and mobile applications (utility, social, gaming).

**Complexity Metrics:** Our generated codebases exhibit realistic complexity distributions with cyclomatic complexity scores ranging from 0.3 to 1.0, file counts between 10-100 per project, and documentation ratios exceeding industry standards. Automated validation ensures all code compiles successfully and maintains architectural coherence.

**Line of Code:** Figure 13 presents the statistical characteristics of LoCoBench's evaluation projects, revealing realistic complexity distributions with a mean of 14,559 lines of code and 48.7 files per

Table 5: 36 domain categories grouped into 10 main domains.

| Domains | Sub-Domains | Total |
|---|---|---|
| Web Applications | E-commerce, Social Platforms, CMS, Dashboards, Blogs, Portfolios | 6 |
| API Services | REST APIs, GraphQL Services, Microservices, API Gateways | 4 |
| Data Systems | Analytics Platforms, ETL Pipelines, Data Warehouses, Streaming, Data Lakes | 5 |
| ML/AI Systems | Training Platforms, Inference Services, NLP Systems, Computer Vision | 4 |
| Desktop Applications | Productivity Tools, Media Applications, Development Environments | 3 |
| Mobile Applications | Social Apps, Utility Apps, Mobile Games | 3 |
| System Infrastructure | Monitoring, Automation, Networking, Security Tools | 4 |
| Financial Technology | Payment Systems, Trading Platforms, Banking Applications | 3 |
| Gaming & Simulation | Game Engines, Simulation Frameworks | 2 |
| Blockchain Systems | DeFi Platforms, NFT Marketplaces | 2 |

Table 6: Additional uniqueness factors.

| Factor | Details | Total |
|---|---|---|
| Architecture Patterns | Monolithic, Microservices, Serverless, Event-Driven, Layered, Clean Architecture, Hexagonal, MVC, MVVM, Component-Based | 10 |
| Project Themes | Business, Education, Healthcare, Entertainment, Productivity, Social, Utility, Creative | 8 |
| Complexity Levels | Easy (25%), Medium (25%), Hard (25%), Expert (25%) | 4 |

project. The right-skewed distributions (top row) mirror real-world software patterns, ranging from compact applications to enterprise-scale systems with over 40,000 lines. The language-specific analysis (bottom row) shows distinct patterns: systems languages (C, Rust) exhibit compact implementations, object-oriented languages (Java, C#) demonstrate higher complexity with extensive file structures, while web languages (JavaScript, TypeScript, PHP) show intermediate levels. These patterns validate LoCoBench's realistic representation across programming paradigms and complexity levels.

# B  MORE DETAILS ON EVALUATION METRICS

## B.1  SOFTWARE ENGINEERING EXCELLENCE (8 METRICS)

This dimension evaluates sophisticated software engineering capabilities essential for complex development scenarios.

❶ **Architectural Coherence Score (ACS):** We introduce this new metric to evaluate LLMs' ability to maintain system-level design consistency across large codebases. Traditional metrics cannot capture architectural understanding at the scale required for long-context evaluation.

Let $\mathcal{C} = \{c_1, c_2, \ldots, c_n\}$ represent a codebase and $\mathcal{P} = \{p_1, p_2, \ldots, p_m\}$ denote the set of architectural patterns detected in $\mathcal{C}$. For each pattern $p_i \in \mathcal{P}$, we define:

$$ACS(\mathcal{C}) = \frac{1}{|\mathcal{P}|} \sum_{p \in \mathcal{P}} w(p) \cdot \frac{\alpha(p, \mathcal{C})}{\kappa(p) + \epsilon} \tag{1}$$

where $w(p) \in [0, 1]$ is the criticality weight of pattern $p$, $\alpha(p, \mathcal{C}) \in [0, 1]$ measures pattern adherence through SOLID principle compliance and design constraint satisfaction, $\kappa(p) \geq 1$ represents pattern complexity, and $\epsilon > 0$ is a regularization constant preventing division by zero.

❷ **Dependency Traversal Accuracy (DTA):** This new metric specifically evaluates LLMs' capability to navigate complex inter-module dependencies in long-context scenarios, addressing a key gap in existing evaluation frameworks.

Let $\mathcal{G} = (V, E)$ be the dependency graph where $V$ represents modules and $E$ denotes dependency relationships. For each dependency $d_{ij} \in E$ from module $v_i$ to $v_j$, we define:

$$DTA(\mathcal{G}) = \frac{1}{|E|} \sum_{d_{ij} \in E} \frac{\mu(d_{ij}) \cdot \gamma(d_{ij}, \mathcal{G})}{\delta(d_{ij}) + 1} \tag{2}$$

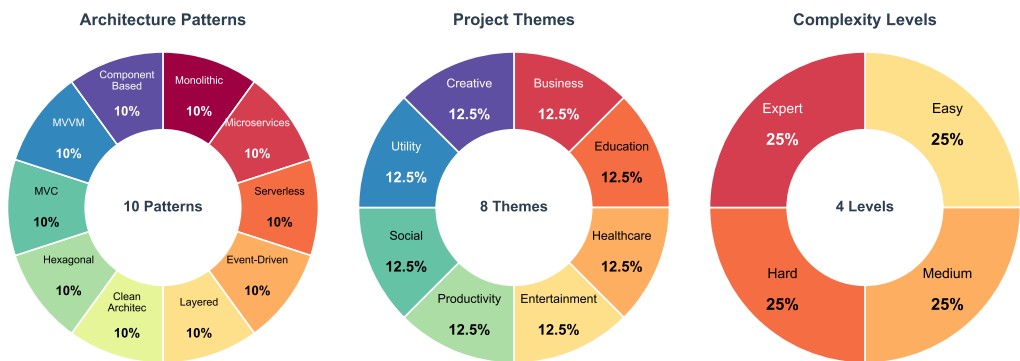

Figure 12: Additional uniqueness factors in LoCoBench. Three independent factors provide comprehensive evaluation coverage: **Left:** 10 architecture patterns including modern paradigms (microservices, serverless, event-driven) and traditional approaches (monolithic, layered, MVC), ensuring evaluation across diverse software architectures. **Center:** 8 project themes spanning business applications, educational tools, healthcare systems, entertainment platforms, productivity software, social applications, utilities, and creative tools. **Right:** 4 complexity levels (Easy, Medium, Hard, Expert) with equal 25% distribution, providing systematic difficulty progression from basic long-context tasks to enterprise-scale challenges.

where $\mu(d_{ij}) \in [0, 1]$ measures correct usage through import validation and interface compliance, $\gamma(d_{ij}, \mathcal{G}) \in [0, 1]$ quantifies contextual awareness of dependency relationships within graph $\mathcal{G}$, and $\delta(d_{ij}) \geq 0$ represents the transitive dependency depth of edge $d_{ij}$.

❸ **Cross-File Reasoning Depth (CFRD):** We propose this metric to assess LLMs' understanding of multi-file relationships and interactions, a capability crucial for complex software development but not measured by existing benchmarks.

Given a file set $\mathcal{F} = \{f_1, f_2, \ldots, f_n\}$ and the cross-file interaction matrix $\mathbf{R} \in \mathbb{R}^{n \times n}$, we define:

$$CFRD(\mathcal{F}) = \frac{1}{n(n-1)} \sum_{i=1}^{n} \sum_{\substack{j=1 \\ j \neq i}}^{n} \rho(f_i, f_j) \cdot \iota(f_i, f_j) \tag{3}$$

where $\rho(f_i, f_j) \in [0, 1]$ quantifies reasoning depth between files $f_i$ and $f_j$ through semantic analysis and cross-reference understanding, and $\iota(f_i, f_j) \in [0, 1]$ measures interaction complexity based on coupling strength, interface dependencies, and shared abstractions.

❹ System Thinking Score (STS): Adapted from systems engineering assessment frameworks (Blanchard & Fabrycky, 2016), measuring holistic software system understanding and scalability awareness.

❺ Robustness Score (RS): Based on IEEE/ISO 25010 software quality standards (iso, 2011), evaluating code reliability, error handling, and defensive programming practices.

❻ Comprehensiveness Score (CS): Derived from software completeness metrics in quality assurance literature (Kan, 2002), assessing solution coverage, documentation quality, and requirement fulfillment.

❼ Innovation Score (IS): Adapted from creative problem-solving assessment in software engineering research (Glass, 2002), evaluating new approaches, modern practices, and creative solutions.

❽ Solution Elegance Score (SES): Based on code aesthetics and design quality metrics (Buse & Weimer, 2010), measuring code clarity, theoretical soundness, and adherence to clean code principles.

B.2 FUNCTIONAL CORRECTNESS (4 METRICS)

This dimension assesses the fundamental correctness and executability of generated code.

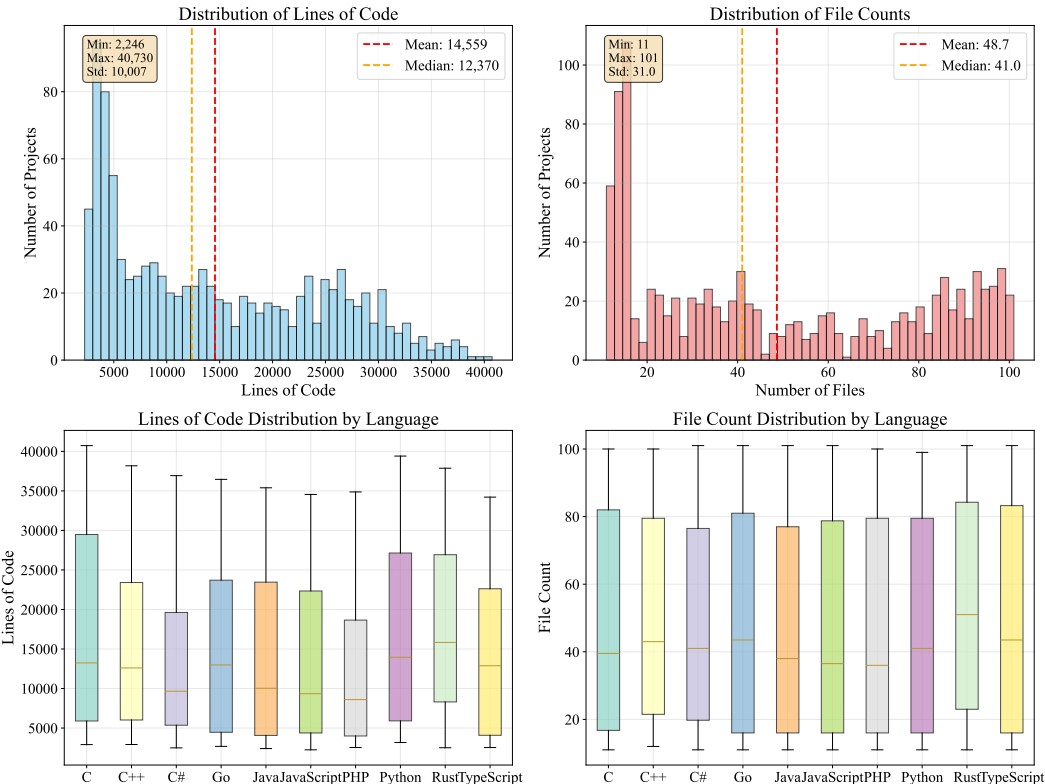

Figure 13: LoCoBench's evaluation projects analysis. **Top row** shows distribution of lines of code (left) and file counts (right) across all evaluation projects. **Bottom row:** Programming language breakdown displaying lines of code distribution (left) and file count distribution (right) across 10 programming languages.

❶ **Incremental Development Capability (IDC):** We introduce this metric to evaluate LLMs' ability to build effectively on previous development work across multiple sessions, a crucial capability for long-context software development not addressed by existing metrics.

Let $\mathcal{T} = \{t_1, t_2, \ldots, t_k\}$ represent a sequence of incremental development tasks applied to codebase state transitions $\mathcal{S}_0 \rightarrow \mathcal{S}_1 \rightarrow \cdots \rightarrow \mathcal{S}_k$. For each task $t_i$:

$$IDC(\mathcal{T}) = \frac{1}{|\mathcal{T}|} \sum_{i=1}^{|\mathcal{T}|} \frac{\xi(t_i, \mathcal{S}_{i-1}) \cdot \sigma(t_i, \mathcal{S}_i)}{\beta(t_i, \mathcal{S}_{i-1}, \mathcal{S}_i) + 1} \tag{4}$$

where $\xi(t_i, \mathcal{S}_{i-1}) \in [0, 1]$ measures extension quality of task $t_i$ relative to previous state $\mathcal{S}_{i-1}$, $\sigma(t_i, \mathcal{S}_i) \in [0, 1]$ quantifies integration smoothness in resulting state $\mathcal{S}_i$, and $\beta(t_i, \mathcal{S}_{i-1}, \mathcal{S}_i) \geq 0$ counts breaking changes introduced during the transition.

❷ Code Compilation Success (CCS): Binary assessment of syntactic correctness, a fundamental metric established in early software engineering literature (McCabe, 1976).

❸ Unit Test Performance (UTP): Individual component testing validation, a standard practice from software testing methodology (Myers et al., 2011).

❹ Integration Test Performance (ITP): System-wide functionality assessment, based on established integration testing frameworks (Binder, 1999).

## B.3 CODE QUALITY ASSESSMENT (3 METRICS)

This dimension evaluates security, maintainability, and adherence to coding standards.

❶ Security Analysis Score (SAS): Vulnerability assessment based on OWASP security analysis frameworks (OWASP, 2021) and static analysis techniques, evaluating common security issues including SQL injection, XSS, buffer overflows, and insecure cryptographic practices.

❷ Average Issues Found - Inverted (AIF): Code quality issue detection derived from static analysis research and modern quality assessment tools (Campbell & Papapetrou, 2013), measuring the absence of code smells, complexity violations, and maintainability issues (lower issue count yields higher score).

❸ Code Style Adherence (CSA): Style guide compliance measurement based on coding standards literature (Kernighan & Pike, 1999) and automated linting frameworks, evaluating naming conventions, formatting consistency, and language-specific best practices.

## B.4 LONG-CONTEXT UTILIZATION (2 METRICS)

This dimension specifically evaluates capabilities unique to long-context software development scenarios.

❶ **Information Coverage Utilization (ICU):** We propose this metric to evaluate how effectively LLMs utilize large context windows, addressing a critical gap in long-context evaluation.

Given context window $\mathcal{W} = \{w_1, w_2, \ldots, w_m\}$ and task-specific information elements $\mathcal{I} = \{i_1, i_2, \ldots, i_n\} \subseteq \mathcal{W}$, we define:

$$ICU(\mathcal{W}, \mathcal{I}) = \frac{|\mathcal{U}(\mathcal{I})|}{|\mathcal{I}|} \cdot \frac{\sum_{u \in \mathcal{U}(\mathcal{I})} \tau(u)}{\phi(\mathcal{U}(\mathcal{I})) + \epsilon} \tag{5}$$

where $\mathcal{U}(\mathcal{I}) \subseteq \mathcal{I}$ represents the subset of utilized information elements, $\tau(u) \in [0, 1]$ quantifies the task relevance of element $u$, $\phi(\mathcal{U}(\mathcal{I})) \geq 0$ measures redundancy penalty through information overlap, and $\epsilon > 0$ is a regularization constant.

❷ **Multi-Session Memory Retention (MMR):** This new metric assesses context persistence across extended development sessions, essential for evaluating long-context capabilities in realistic software development workflows.

Consider a sequence of development sessions $\mathcal{S} = \{s_1, s_2, \ldots, s_k\}$ with associated context states $\{\mathcal{C}_1, \mathcal{C}_2, \ldots, \mathcal{C}_k\}$. We define:

$$MMR(\mathcal{S}) = \frac{1}{|\mathcal{S}|} \sum_{j=1}^{|\mathcal{S}|} \frac{\psi(s_j, \mathcal{C}_{j-1}) \cdot \chi(s_j, \mathcal{C}_j)}{\log(j+1)} \tag{6}$$

where $\psi(s_j, \mathcal{C}_{j-1}) \in [0, 1]$ measures information retention from previous context state $\mathcal{C}_{j-1}$ to session $s_j$, $\chi(s_j, \mathcal{C}_j) \in [0, 1]$ quantifies consistency maintenance in current context state $\mathcal{C}_j$, and the logarithmic decay term $\log(j+1)$ models expected memory degradation over temporal distance.

## B.5 LoCoBench SCORE (LCBS)

We define a unified LoCoBench Score (LCBS) as a weighted aggregate function that maps the 17-dimensional metric space to a scalar evaluation score. Let $\mathcal{M} = \{m_1, m_2, \ldots, m_{17}\}$ represent the complete set of evaluation metrics, partitioned into four evaluation dimensions.

**Dimension Partitioning:** The metric space is partitioned as:

$$\mathcal{M}_{SE} = \{ACS, DTA, CFRD, STS, RS, CS, IS, SES\} \quad |\mathcal{M}_{SE}| = 8 \tag{7}$$

$$\mathcal{M}_{FC} = \{CCS, UTP, ITP, IDC\} \quad |\mathcal{M}_{FC}| = 4 \tag{8}$$

$$\mathcal{M}_{CQ} = \{SAS, AIF, CSA\} \quad |\mathcal{M}_{CQ}| = 3 \tag{9}$$

$$\mathcal{M}_{LCU} = \{ICU, MMR\} \quad |\mathcal{M}_{LCU}| = 2 \tag{10}$$

where $\mathcal{M}_{SE} \cup \mathcal{M}_{FC} \cup \mathcal{M}_{CQ} \cup \mathcal{M}_{LCU} = \mathcal{M}$ and the sets are pairwise disjoint.

**Normalization Function:** For each metric $m_i \in \mathcal{M}$, we define a normalization function $\mathcal{N} : \mathbb{R} \to [0, 1]$ that maps raw metric values to the unit interval:

$$\mathcal{N}(m_i) = \frac{m_i - \min(m_i)}{\max(m_i) - \min(m_i)} \tag{11}$$

**Dimension Aggregation:** Each dimension score is computed as the arithmetic mean of its constituent normalized metrics:

$$SE = \frac{1}{|\mathcal{M}_{SE}|} \sum_{m \in \mathcal{M}_{SE}} \mathcal{N}(m) = \frac{1}{8} \sum_{i=1}^{8} \mathcal{N}(m_i^{SE}) \tag{12}$$

$$FC = \frac{1}{|\mathcal{M}_{FC}|} \sum_{m \in \mathcal{M}_{FC}} \mathcal{N}(m) = \frac{1}{4} \sum_{i=1}^{4} \mathcal{N}(m_i^{FC}) \tag{13}$$

$$CQ = \frac{1}{|\mathcal{M}_{CQ}|} \sum_{m \in \mathcal{M}_{CQ}} \mathcal{N}(m) = \frac{1}{3} \sum_{i=1}^{3} \mathcal{N}(m_i^{CQ}) \tag{14}$$

$$LCU = \frac{1}{|\mathcal{M}_{LCU}|} \sum_{m \in \mathcal{M}_{LCU}} \mathcal{N}(m) = \frac{1}{2} \sum_{i=1}^{2} \mathcal{N}(m_i^{LCU}) \tag{15}$$

**Weight Vector:** We define the weight vector $\mathbf{w} = [w_{SE}, w_{FC}, w_{CQ}, w_{LCU}]^T$ where:

$$\mathbf{w} = [0.4, 0.3, 0.2, 0.1]^T \quad \text{such that} \quad \sum_i w_i = 1 \tag{16}$$

The weights are empirically determined to reflect the relative importance of each dimension in long-context software development scenarios, with software engineering excellence receiving the highest weight due to its comprehensive nature.

**Final Score:** The LoCoBench Score is defined as a weighted linear combination scaled to the interval [0,5]:

$$LCBS = 5 \cdot \mathbf{w}^T \cdot [SE, FC, CQ, LCU]^T = 5 \cdot (w_{SE} \cdot SE + w_{FC} \cdot FC + w_{CQ} \cdot CQ + w_{LCU} \cdot LCU) \tag{17}$$

Substituting the weight values:

$$LCBS = 5 \cdot (0.4 \cdot SE + 0.3 \cdot FC + 0.2 \cdot CQ + 0.1 \cdot LCU) \tag{18}$$

## C  FULL EXPERIMENTAL RESULTS AND DISCUSSIONS

### C.1  EVALUATION INFRASTRUCTURE AND PROCESS

LoCoBench provides a comprehensive evaluation infrastructure designed for reliable, scalable assessment: ❶ **Model Integration:** Our framework supports evaluation of any long-context LLM through standardized APIs, including OpenAI GPT, Google Gemini, and Anthropic Claude models. Each model is evaluated with consistent hyperparameters to ensure reproducible results. ❷ **Context Management:** Advanced context windowing techniques handle scenarios exceeding model context limits, with intelligent truncation strategies that preserve essential information while maintaining task solvability. ❸ **Execution Environment:** Isolated Docker containers provide secure execution environments for code validation, with language-specific toolchains and timeout mechanisms (3600 seconds per evaluation) to prevent infinite loops or resource exhaustion. ❹ **Error Recovery:** Robust error handling addresses common evaluation challenges including parsing failures, compilation errors, and runtime exceptions, with detailed logging for debugging and analysis.

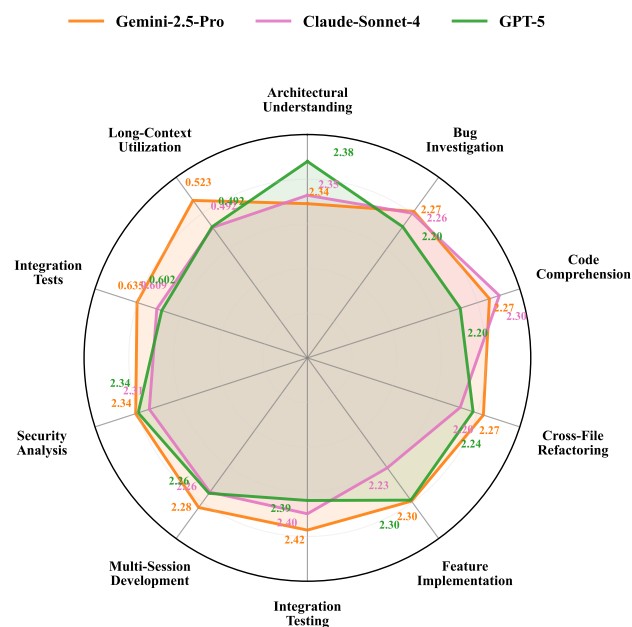

Figure 14: Overall performance comparison of GPT-5, Gemini-2.5-Pro, and Claude-Sonnet-4 across 10 LoCoBench dimensions. Gemini-2.5-Pro demonstrates superior performance on many aspects, particularly on cross-file refactoring, long-context utilization, integration tests, and multi-session development capabilities, while GPT-5 excels in architectural understanding. Claude-Sonnet-4 shows balanced performance with particular strength in code comprehension.

## C.2 OVERALL MODEL PERFORMANCE ANALYSIS

Figure 14 compares the performance of three leading LLMs across 10 evaluation dimensions, showing distinct performance profiles that reflect different architectural strengths and optimization strategies. Gemini-2.5-Pro emerges as the overall leader, demonstrating exceptional performance in cross-file refactoring, long-context utilization, integration tests, and multi-session development. This model shows particular strength in complex software engineering tasks that require deep system-level reasoning and comprehensive testing capabilities. Its superior performance suggests strong capabilities for comprehending large-scale software designs and identifying structural patterns across extensive codebases.

GPT-5 achieves competitive performance, showing remarkable consistency across most evaluation dimensions. Notably, GPT-5 demonstrates the highest performance in architectural understanding, indicating specialized capabilities for recognizing and analyzing complex software design patterns. This strength in architectural comprehension suggests that GPT-5 may be particularly well-suited for tasks involving system design analysis and high-level software architecture evaluation. Claude-Sonnet-4 presents a distinctive performance profile, showing particular excellence in code comprehension, which indicates strong capabilities for understanding and analyzing existing codebases.

Figure 14 shows that all three models achieve relatively similar performance levels across many dimensions, with the largest performance gaps occurring in specialized areas such as long-context utilization and specific task categories. This convergence suggests that current state-of-the-art models have reached similar competency levels for basic long-context software development tasks, while differentiation occurs primarily in specialized capabilities requiring domain-specific reasoning patterns. The custom per-axis scaling employed in the visualization effectively highlights these subtle but important performance differences that would be obscured by uniform scaling approaches.

Interestingly, the performance patterns suggest that different models may have been optimized for different aspects of software development workflows. The variation in long-context utilization capabilities across models indicates that handling extended context windows remains a significant technical challenge, with different approaches yielding varying degrees of success. This specialization

pattern has important implications for practical deployment, as organizations may benefit from selecting models based on their specific software development needs and the types of long-context tasks they most frequently encounter. The relatively tight performance clustering among these top-tier models also suggests that the field of long-context code understanding is approaching certain fundamental limitations with current architectures and training methodologies. Future improvements may require new approaches to context management, architectural understanding, and multi-file reasoning rather than incremental refinements to existing techniques.

## C.3 COMPREHENSIVE MODEL RANKING ANALYSIS

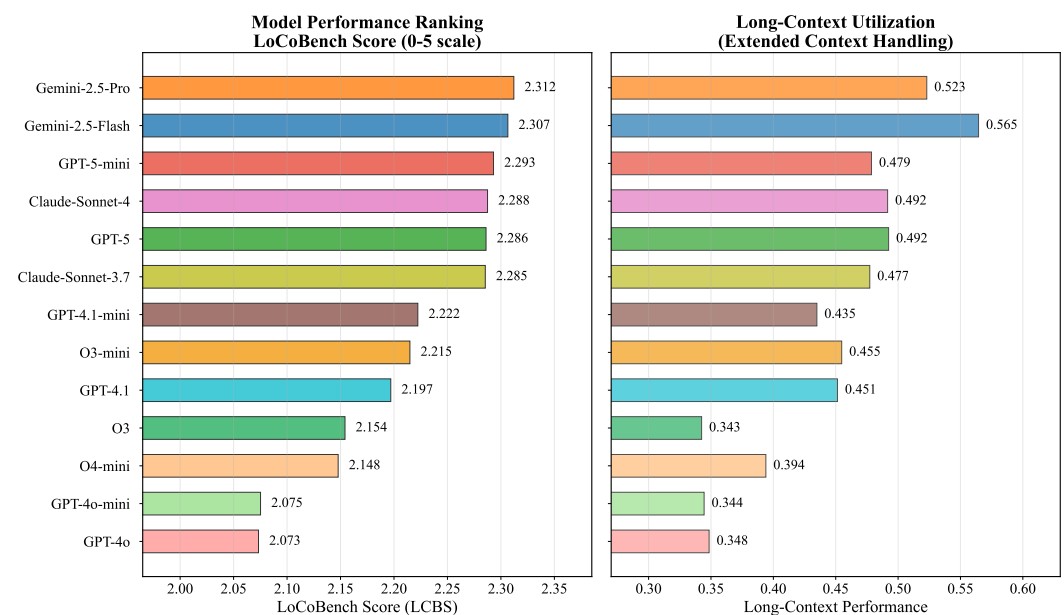

Figure 15: Model ranking and long-context utilization comparison. Left chart shows LoCoBench Score (LCBS) rankings. Right chart displays long-context utilization performance.

Figure 15 presents a comprehensive ranking of all evaluated models across two dimensions: general software engineering competency and specialized long-context processing capabilities. The left chart displays overall LoCoBench Score (LCBS) performance, revealing a clear performance hierarchy with Gemini-2.5-Pro achieving the highest score. The performance distribution shows relatively tight clustering among top-tier models, indicating that leading models have achieved similar competency levels for complex software development tasks. This clustering pattern suggests that the current generation of large language models has reached a plateau in general software engineering capabilities, with incremental improvements rather than dramatic performance leaps.

The right chart focuses specifically on long-context utilization capabilities, revealing markedly different performance patterns compared to overall rankings. Gemini-2.5-Flash demonstrates superior long-context processing abilities, suggesting specialized optimization for extended context handling that may come at the expense of other capabilities. This divergence between overall performance and long-context specialization highlights the distinct challenges posed by extended context scenarios versus general software engineering tasks. The performance gap in long-context utilization is notably larger than in overall scores, indicating that effective context management remains a significant technical challenge requiring specialized architectural solutions.

The dual visualization reveals that model performance varies significantly between comprehensive software engineering evaluation and specialized long-context capabilities. While some models excel in overall software development competency, others show particular strength in processing and utilizing extended context information, suggesting different architectural optimizations and training

strategies across model families. This specialization pattern reflects the inherent trade-offs in model design, where optimization for specific capabilities may impact performance in other areas.

The model ranking also demonstrates the importance of evaluating models across multiple dimensions rather than relying on single aggregate scores. Models that appear similar in overall performance may exhibit substantial differences in specific capabilities that are crucial for particular use cases. For organizations deploying these models in production environments, understanding these performance trade-offs is essential for selecting the most appropriate model for their specific long-context software development requirements.

Furthermore, the performance distribution across all evaluated models reveals a clear stratification, with distinct performance tiers emerging. This stratification suggests that while the top-performing models are relatively close in capability, there remain significant gaps between different model generations and architectures. The lower-performing models in the ranking may still be suitable for specific applications or resource-constrained environments, highlighting the importance of comprehensive evaluation frameworks like LoCoBench for understanding the full spectrum of model capabilities.

## C.4 PROGRAMMING LANGUAGE PERFORMANCE ANALYSIS

**Programming Language Performance LoCoBench Score by Model and Language**

| | Python | PHP | JavaScript | TypeScript | Java | C# | Go | C++ | Rust | C |
|---|---|---|---|---|---|---|---|---|---|---|
| Gemini-2.5-Pro | 2.788 | 2.641 | 2.268 | 2.203 | 2.326 | 2.419 | 2.338 | 2.074 | 2.002 | 2.063 |
| Gemini-2.5-Flash | 2.715 | 2.636 | 2.274 | 2.230 | 2.329 | 2.432 | 2.292 | 2.064 | 2.039 | 2.055 |
| GPT-5-mini | 2.799 | 2.593 | 2.280 | 2.189 | 2.329 | 2.389 | 2.264 | 2.039 | 2.087 | 1.961 |
| Claude-Sonnet-4 | 2.677 | 2.606 | 2.314 | 2.252 | 2.331 | 2.347 | 2.311 | 2.023 | 1.997 | 2.017 |
| GPT-5 | 2.837 | 2.563 | 2.286 | 2.186 | 2.340 | 2.331 | 2.268 | 2.037 | 2.053 | 1.959 |
| Claude-Sonnet-3.7 | 2.735 | 2.585 | 2.308 | 2.212 | 2.318 | 2.377 | 2.302 | 2.014 | 2.016 | 1.988 |
| GPT-4.1-mini | 2.772 | 2.544 | 2.233 | 2.083 | 2.273 | 2.299 | 2.213 | 1.954 | 1.923 | 1.928 |
| O3-mini | 2.714 | 2.533 | 2.186 | 2.093 | 2.281 | 2.261 | 2.184 | 1.996 | 1.993 | 1.908 |
| GPT-4.1 | 2.723 | 2.522 | 2.228 | 2.099 | 2.268 | 2.263 | 2.143 | 1.927 | 1.891 | 1.906 |
| O3 | 2.749 | 2.430 | 2.147 | 2.037 | 2.168 | 2.123 | 2.173 | 1.923 | 1.953 | 1.839 |
| O4-mini | 2.688 | 2.463 | 2.145 | 2.026 | 2.182 | 2.212 | 2.136 | 1.905 | 1.888 | 1.834 |
| GPT-4o-mini | 2.592 | 2.407 | 2.065 | 1.943 | 2.143 | 2.162 | 2.010 | 1.854 | 1.776 | 1.800 |

Figure 16: Programming language performance heatmap showing model performance across 10 programming languages. Languages are ordered by difficulty from easiest to hardest.

Figure 16 presents a comprehensive analysis of model performance across 10 programming languages, revealing distinct patterns in language-specific capabilities. The heatmap visualization shows clear performance variations across different programming paradigms, with models demonstrating varying proficiency levels depending on language characteristics and complexity.

The analysis reveals that models generally achieve higher performance on high-level languages such as Python and PHP, while showing more challenging performance patterns on systems programming languages like C and Rust. This performance gradient reflects the inherent complexity differences

between languages and the varying amounts of training data typically available for different programming languages. The ordering from easiest to hardest languages demonstrates a consistent difficulty progression that aligns with traditional programming language learning curves and industry adoption patterns.

Language-specific performance patterns indicate that model training and optimization strategies may be influenced by language popularity and representation in training datasets. The consistent performance ordering across most models suggests systematic challenges posed by certain language features, such as memory management in systems languages and complex type systems in modern programming languages. Notably, web development languages like JavaScript and TypeScript show intermediate performance levels, reflecting their moderate complexity and widespread usage in training corpora.

Figure 16 also reveals interesting model-specific strengths and weaknesses across languages. While most models follow similar performance trends, some demonstrate particular proficiency in specific language domains, suggesting that certain architectural choices or training methodologies may be more effective for particular programming paradigms. This language-dependent performance variation has important implications for practical deployment, as organizations working primarily with specific programming languages may benefit from selecting models that demonstrate superior performance in their target language ecosystem.

Furthermore, the performance patterns observed across languages provide insights into the fundamental challenges of long-context code understanding. Systems programming languages, which typically require more precise memory management and lower-level reasoning, consistently pose greater challenges across all evaluated models. This suggests that current long-context LLMs may struggle with the detailed, hardware-aware reasoning required for effective systems programming, highlighting an important area for future model development and training optimization.

## C.5 TASK CATEGORY PERFORMANCE AND DIFFICULTY ANALYSIS

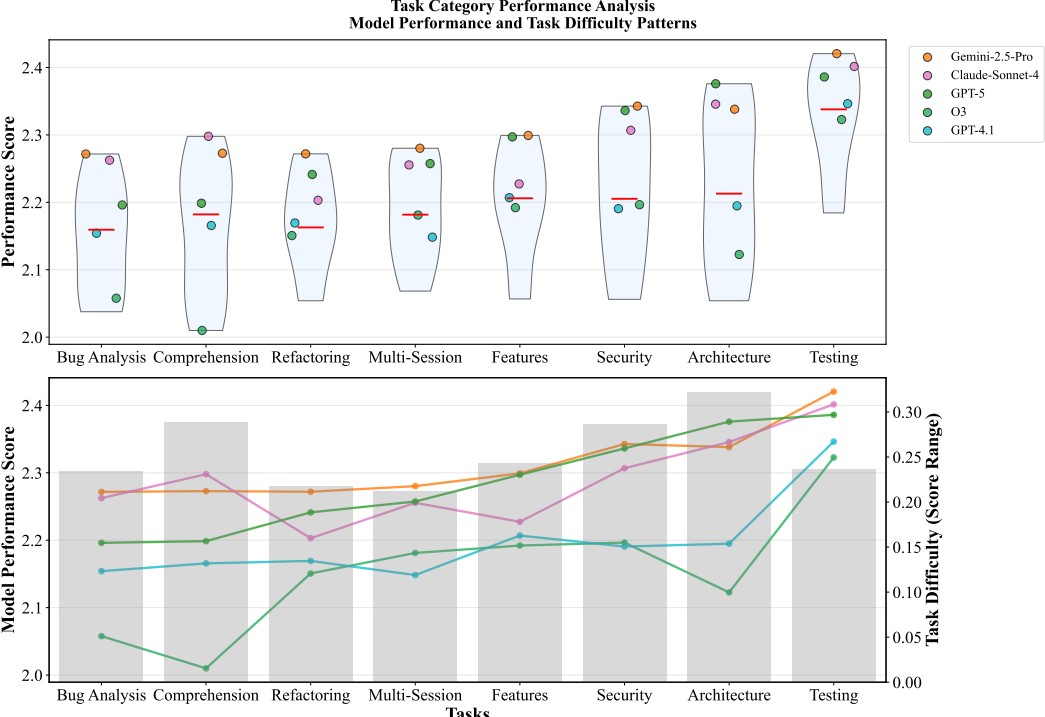

Figure 17: Task category performance analysis. Top chart shows performance distribution across all models for each task category, with individual model performance overlaid. Bottom chart displays task difficulty patterns and model performance trends across different software engineering tasks.

Figure 17 presents a comprehensive analysis of model performance across eight distinct task categories, revealing both individual model capabilities and inherent task difficulty patterns. The visualization shows overall performance distributions with detailed model-specific analysis, providing insights into both the challenges posed by different software engineering tasks and the varying capabilities of evaluated models.

The top chart shows the complete performance distribution across all evaluated models for each task category, with individual model performances overlaid as scatter points. This figure reveals significant variations in task difficulty, with some categories showing wide performance distributions indicating high variability in model capabilities, while others demonstrate more consistent performance patterns. The plot provide insights into the underlying performance characteristics, with broader distributions indicating tasks where models show more varied success rates, and narrower distributions suggesting more consistent challenge levels across different model architectures.

The bottom chart focuses on task difficulty analysis by displaying the performance range (score variance) for each task category as background bars, while overlaying individual model performance trends as connected line plots. This dual-axis approach effectively illustrates the relationship between inherent task difficulty and model-specific performance patterns. Tasks with larger score ranges indicate greater difficulty variation among models, suggesting that these tasks may be more sensitive to specific architectural optimizations or training methodologies.

The analysis reveals distinct performance patterns across task categories, with integration testing and architectural understanding generally showing higher performance scores, while tasks such as bug investigation and multi-session development present greater challenges for most models. This performance hierarchy reflects the varying complexity of different software engineering activities, with some tasks requiring more sophisticated reasoning capabilities or longer-context understanding than others. The consistent ordering of task difficulty across most models suggests that certain software engineering challenges are fundamentally more difficult for current long-context LLMs, regardless of their specific architectural approaches.

Model-specific performance patterns also emerge from the analysis, with some models demonstrating particular strengths in specific task categories while showing relative weaknesses in others. This specialization pattern indicates that different models may have been optimized for different aspects of software engineering workflows, or that their training data may have contained varying representations of different task types. The performance variations across tasks have important implications for practical deployment, as organizations may benefit from selecting models based on the specific types of software engineering tasks they most frequently encounter.

Figure 17 shows the importance of considering both absolute performance levels and performance consistency when evaluating models for long-context software development tasks. Tasks that show high performance variance may require more careful model selection and potentially different evaluation strategies, while tasks with consistent performance patterns across models may be more predictable in production environments. This analysis framework provides valuable insights for both model developers seeking to improve specific capabilities and practitioners selecting appropriate models for their software development workflows.

## C.6 CONTEXT LENGTH AND DIFFICULTY IMPACT ANALYSIS

Figure 18 provides a comprehensive analysis of how context length and task difficulty impact model performance across multiple dimensions, revealing critical insights into model behavior under varying challenge levels. Figure 18 shows different aspects of performance patterns, from overall difficulty trends to individual model characteristics and consistency analysis.

The upper left chart reveals the performance distribution across difficulty levels, showing how task complexity affects overall model performance. The visualization demonstrates clear performance degradation patterns as difficulty increases from easy to expert levels, with corresponding increases in context length requirements. This analysis reveals that the relationship between context length and difficulty creates compounding challenges for long-context models, where both factors contribute to performance decline. The distribution patterns also show varying levels of performance variance across difficulty levels, indicating that some difficulty categories present more consistent challenges while others exhibit higher variability in model responses.

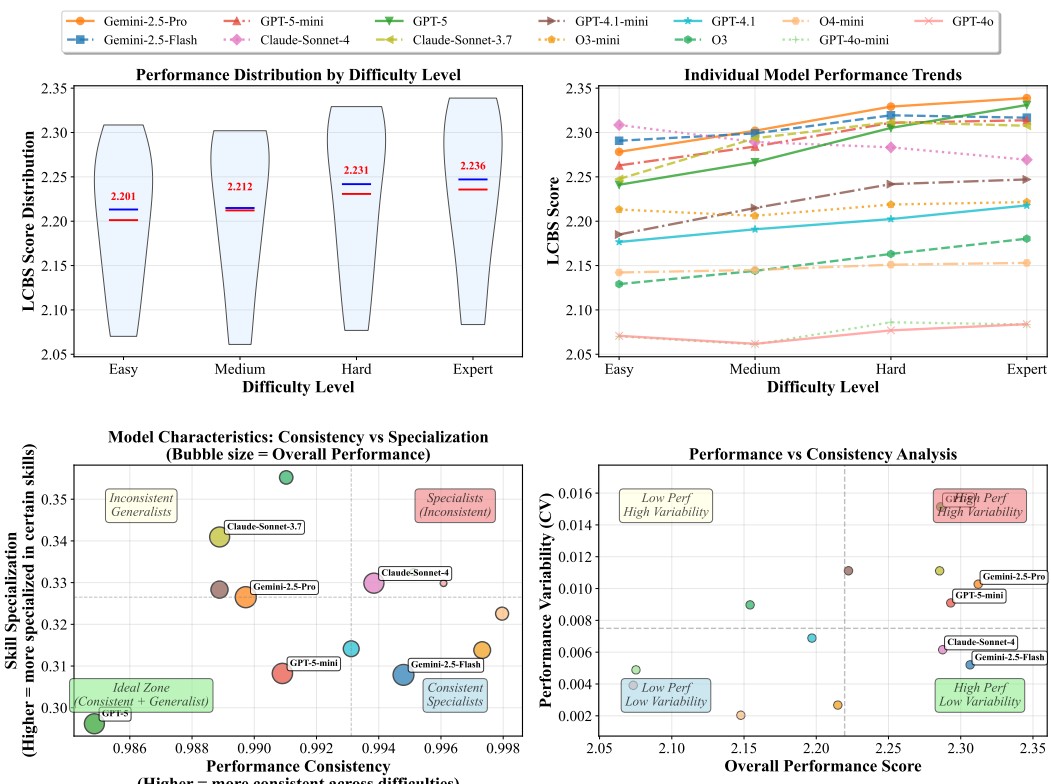

Figure 18: Context length and difficulty impact analysis. Upper left shows performance distribution by difficulty level. Upper right displays individual model performance trends across difficulty levels. Lower left presents model consistency versus specialization patterns. Lower right analyzes performance versus consistency relationships.

The upper right chart shows individual model performance trends across all difficulty levels, showing how different models handle increasing complexity and context requirements. The analysis reveals distinct model behavior patterns, with some models maintaining relatively stable performance across difficulty levels while others show significant degradation. This visualization demonstrates that model architectures respond differently to the combined challenges of increased context length and task complexity, suggesting that different optimization strategies may be more effective for different difficulty ranges.

The lower left chart presents analysis of model characteristics through consistency versus specialization patterns. This analysis examines whether models perform consistently across different difficulty levels or show specialized strengths in particular areas. The bubble chart visualization reveals that models exhibit varying trade-offs between consistency and specialization, with some models demonstrating stable performance across all difficulty levels while others show strong performance in specific areas but greater variability overall. The bubble sizes represent overall performance levels, providing insights into how these characteristics relate to absolute performance capabilities.

The lower right chart analyzes the relationship between overall performance and consistency. This analysis shows that high-performing models do not necessarily exhibit consistent performance across all difficulty levels, and some models achieve strong overall scores while showing significant variability in specific scenarios. This finding has important implications for model selection in production environments, where consistency may be as important as peak performance for reliable system behavior.

The comprehensive analysis reveals that context length and difficulty interact in complex ways that affect different models differently. Some models show graceful degradation patterns that maintain reasonable performance even at expert difficulty levels, while others exhibit more dramatic

performance drops as context requirements increase. These patterns suggest that different model architectures may be optimized for different aspects of long-context processing, with some prioritizing consistency and others focusing on peak performance capabilities.

The multi-dimensional analysis framework also highlights the importance of evaluating models across multiple metrics rather than relying solely on aggregate performance scores. Models that appear similar in overall performance may exhibit substantially different consistency patterns, specialization characteristics, and responses to difficulty scaling. This evaluation provides findings for both model developers seeking to improve specific aspects of long-context performance and practitioners selecting appropriate models for specific deployment scenarios with known difficulty and context requirements.

## C.7 DOMAIN SPECIALIZATION AND PERFORMANCE ANALYSIS

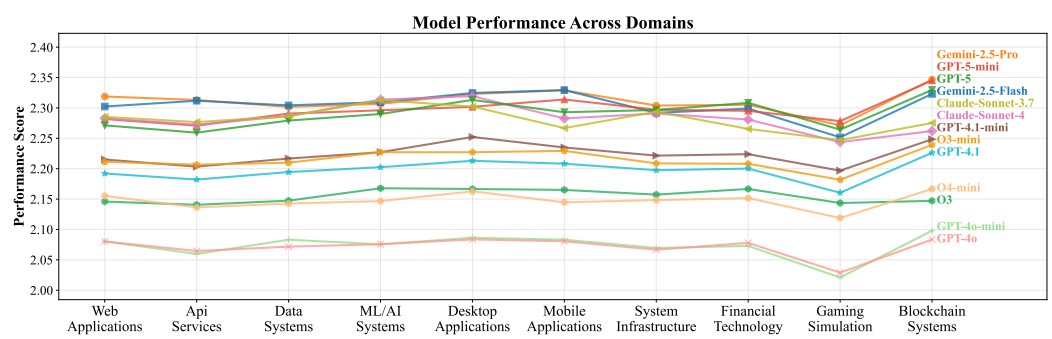

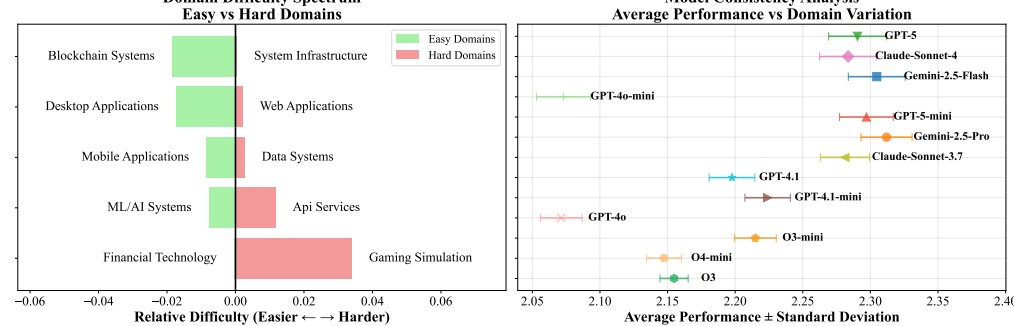

Figure 19: Domain specialization and performance analysis. Top chart shows model performance trends across 10 application domains. Lower left displays domain difficulty spectrum from easiest to hardest. Lower right presents model consistency analysis comparing average performance with domain variation patterns.

Figure 19 presents a comprehensive analysis of model performance across 10 distinct application domains, revealing specialization patterns and consistency characteristics that provide insights into model suitability for different software development contexts. It examines domain-specific performance trends, difficulty hierarchies, and model consistency patterns across diverse application areas.

The top chart displays model performance trajectories across all application domains, showing how different models perform relative to each other across various software development contexts. This visualization reveals distinct patterns in domain-specific performance, with some models maintaining consistent performance across domains while others show significant variation depending on the application area. The analysis demonstrates that domain specialization effects are substantial, with models showing clear preferences for certain types of applications over others. These patterns suggest that training data representation and architectural optimizations may vary significantly across different application domains.

The lower left chart analyzes domain difficulty patterns by presenting the easiest and hardest domains on a relative difficulty spectrum. This analysis reveals that certain application domains consistently pose greater challenges across all evaluated models, while others represent more accessible areas for long-context software development tasks. The difficulty hierarchy shows that domains like Gaming Simulation and Api Services tend to be more challenging, while Blockchain Systems and Desktop Applications generally show higher performance levels. This pattern reflects the varying complexity of different software engineering contexts and the specialized knowledge required for different application areas.

The lower right chart examines model consistency across domains by analyzing average performance levels alongside performance variation patterns. This analysis reveals important differences in how reliably different models perform across diverse application contexts. Some models demonstrate high consistency with low variation across domains, indicating robust general-purpose capabilities, while others show higher variation but potentially stronger peak performance in specific areas. The consistency analysis has important implications for deployment scenarios where predictable performance across diverse applications is crucial.

The domain specialization analysis reveals that model selection should consider not only overall performance levels but also the specific application domains where deployment is intended. Models that excel in web applications may not necessarily perform as well in system infrastructure or blockchain development contexts. This domain-dependent performance variation suggests that organizations working primarily in specific application areas may benefit from selecting models that demonstrate particular strength in their target domains.

Figure 19 also highlights the trade-offs between specialization and generalization in model capabilities. While some models achieve strong performance across all domains with minimal variation, others show more dramatic differences between their strongest and weakest domains. These patterns indicate different training strategies and architectural approaches, with some models optimized for broad applicability and others potentially fine-tuned for specific application contexts.

The comprehensive domain analysis framework provides valuable insights for both strategic model selection and understanding the current limitations of long-context models in different software engineering contexts. The clear domain difficulty hierarchy suggests areas where focused research and development efforts might yield the greatest improvements in long-context software development capabilities, while the consistency analysis helps identify models most suitable for diverse, multi-domain deployment scenarios.

## C.8 ARCHITECTURE PATTERN PERFORMANCE ANALYSIS

Figure 20 presents a comprehensive analysis of model performance across 10 distinct architectural patterns, examining how different software design approaches affect long-context model capabilities. The visualization reveals patterns in architectural complexity, coupling characteristics, and model-specific performance variations across diverse software engineering paradigms.

The top chart displays model performance trajectories across all architectural patterns, showing how individual models respond to different software design approaches. This analysis reveals that models demonstrate varying capabilities when working with different architectural paradigms, with some showing consistent performance across patterns while others exhibit significant variation depending on the architectural approach. The trajectory visualization indicates that certain architectural patterns may be more challenging for long-context understanding, requiring different types of reasoning about system structure and component relationships.

The lower left chart examines the relationship between architectural complexity and performance, with bubble sizes representing performance variation across models. This analysis explores whether more complex architectural patterns necessarily pose greater challenges for long-context models. It reveals the trade-offs between architectural sophistication and model performance, showing how performance variation differs across patterns of varying complexity. Some complex patterns may show consistent performance across models, while simpler patterns might exhibit higher variability in model responses.

The lower right chart presents a coupling/cohesion analysis by grouping architectural patterns into different coupling categories: tight-coupling, moderate-coupling, and loose-coupling patterns. This

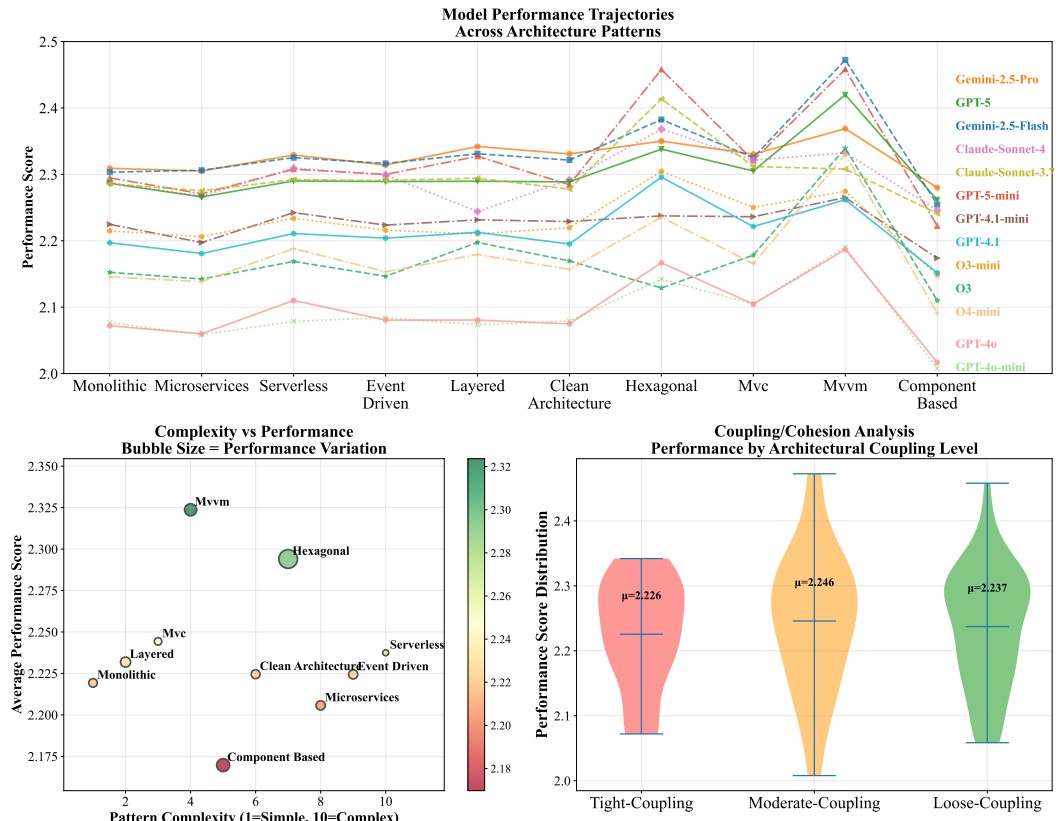

Figure 20: Architecture pattern performance analysis. Top chart shows model performance trajectories across 10 architecture patterns. Lower left presents complexity versus performance relationship with bubble sizes indicating performance variation. Lower right displays coupling/cohesion analysis across different architectural coupling levels.

analysis examines whether the degree of coupling in architectural patterns affects model performance. The coupling analysis provides insights into how component interdependencies and system organization impact long-context model capabilities, revealing whether models perform differently when reasoning about tightly coupled versus loosely coupled system architectures.

The architectural pattern analysis demonstrates that software design paradigms significantly influence model performance in long-context scenarios. Different models show varying proficiency with different architectural approaches, suggesting that model selection for specific projects should consider not only the application domain but also the intended architectural pattern. This pattern-dependent performance variation indicates that training data representation and model architectures may be optimized for certain types of software design patterns over others.

The analysis also reveals important implications for software engineering practice with long-context models. Projects using specific architectural patterns may benefit from selecting models that demonstrate particular strength with those design approaches. The performance variations across patterns suggest that architectural decisions in software projects should consider not only traditional software engineering criteria but also the capabilities and limitations of the long-context models that will be used for development and maintenance tasks.

The comprehensive pattern analysis framework provides valuable guidance for both model developers and software engineers. For model developers, the pattern-specific performance variations highlight areas where focused improvements could enhance architectural understanding capabilities. For software engineers, the analysis provides insights into how architectural choices may impact the effectiveness of long-context model assistance in development workflows, enabling more informed decisions about both architectural patterns and model selection for specific projects.

## D    MORE RELATED WORK

### D.1    CODE GENERATION BENCHMARKS

The landscape of code evaluation benchmarks has evolved significantly, yet most existing work focuses on relatively narrow aspects of programming capability.

**Function-Level Benchmarks:** Early benchmarks like HumanEval (Chen et al., 2021) and MBPP (Austin et al., 2021) established foundational evaluation frameworks for code generation. HumanEval consists of 164 Python programming problems that test basic algorithmic thinking and function implementation. MBPP extends this with 974 entry-level programming tasks. Recent extensions include HumanEval+ (Liu et al., 2023a) which addresses test inadequacy in the original HumanEval, and MultiPL-E (Cassano et al., 2023) which extends HumanEval to 18+ programming languages. BigCodeBench (Zhuo et al., 2024) provides more challenging function-level tasks requiring complex library usage and reasoning. While these benchmarks effectively measure basic code generation capabilities, they operate at the function level and do not capture the complexity of real-world software development.

**Contest Programming Benchmarks:** APPS (Hendrycks et al., 2021) and LiveCodeBench (Jain et al., 2024) focus on competitive programming problems. APPS provides 10,000 problems from coding competitions, while LiveCodeBench offers contamination-free evaluation with problems collected from ongoing contests. CodeContests (Li et al., 2022) extends this paradigm with competitive programming problems from Codeforces, AtCoder, and CodeChef. AlphaCode (Li et al., 2022) demonstrated significant progress on competitive programming but remains limited to algorithmic problem-solving. These benchmarks test algorithmic problem-solving but do not address software engineering concerns like code organization, architectural design, or multi-file development.

**Recent Long-Context Code Benchmarks:** The emergence of million-token context windows has spurred development of specialized long-context code evaluation. LongCodeBench (Rando et al., 2025) evaluates coding LLMs at 1M context windows, demonstrating dramatic performance degradation (29% to 3% for Claude 3.5 Sonnet) as context increases. LongCodeU (Li et al., 2025) focuses on long code understanding across four aspects but shows that LCLMs performance drops significantly beyond 32K tokens. LongCodeArena (Bogomolov et al., 2024) provides code-centric evaluation at 2M+ tokens but focuses primarily on code completion rather than long-context software development capabilities.

**Domain-Specific Benchmarks:** Specialized benchmarks target specific programming domains and languages. DS-1000 (Lai et al., 2022) focuses on data science programming tasks using popular Python libraries like NumPy and Pandas. VerilogEval (Thakur et al., 2023) evaluates hardware description language generation for Verilog. Cococo (Wang et al., 2022) introduces context-aware code completion evaluation. EffiBench (Dong et al., 2024a) evaluates code efficiency rather than just correctness. ClassEval (Du et al., 2023) focuses on class-level code generation requiring understanding of object-oriented programming principles. While domain-specific, these benchmarks still primarily evaluate isolated function or script generation rather than comprehensive software development capabilities.

**Evaluation Methodology Advances:** Recent work has focused on improving evaluation methodologies beyond functional correctness. CodeBLEU (Ren et al., 2020) introduces syntax and semantic awareness for code similarity measurement. BLEU-RT (Sellam et al., 2020) and BERTScore (Zhang et al., 2019) apply neural metrics to code evaluation. Human evaluation studies (Chen et al., 2021; Nijkamp et al., 2022) have shown gaps between automated metrics and human judgments of code quality. AlignCodeBench (Zhang et al., 2024b) introduces evaluation of code alignment with natural language specifications.

**Repository-Level Benchmarks:** Recent work has begun addressing more realistic scenarios. RepoBench (Liu et al., 2023b) evaluates repository-level code completion, while CrossCodeEval (Ding et al., 2023) focuses on cross-file completion tasks. These represent important steps toward more realistic evaluation but remain limited to completion tasks rather than comprehensive development scenarios.

**Survey on Long-Context Language Models:** Liu et al. (Liu et al., 2025) provide an extensive survey on long-context language modeling, covering data strategies, architectural designs, workflow

approaches, training and inference infrastructure, evaluation paradigms, and diverse application scenarios.

## D.2 SOFTWARE ENGINEERING BENCHMARKS

**Real-World Issue Resolution:** SWE-Bench (Jimenez et al., 2023) represents a significant advancement toward realistic software engineering evaluation, providing 2,294 real GitHub issues and their corresponding fixes from 12 Python repositories. Multi-SWE-Bench addresses the language limitation by extending evaluation to 7 programming languages (Java, TypeScript, JavaScript, Go, Rust, C, and C++) with 1,632 high-quality instances curated by expert annotators. Recent work has addressed additional limitations: SWE-rebench (rebench Team, 2025) introduces continuously evolving, decontaminated evaluation to prevent data contamination and standardize long-context evaluation, while LiveSWEBench (Team, 2024) focuses on end-user coding applications with real-world tasks. However, these benchmarks remain limited by their focus on bug fixes rather than comprehensive development workflows.

**Advanced Software Development Benchmarks:** Recent work has begun addressing complex capabilities in software development. DevBench (Li et al., 2024) evaluates LLMs across the entire software development lifecycle, including design, implementation, and testing, but focuses on traditional LLM evaluation rather than long-context capabilities. Advanced evaluation frameworks have introduced intermediate feedback throughout the task-solving process. However, these approaches lack the systematic long-context evaluation and comprehensive task categories needed for thorough assessment of complex software development scenarios.

**Code Understanding Tasks:** CodeXGLUE (Lu et al., 2021) provides a comprehensive suite of 14 tasks covering various aspects of program understanding, including clone detection, defect detection, and code summarization. However, these tasks focus on understanding existing code rather than generating new software components or managing complex development workflows.

## D.3 LONG-CONTEXT EVALUATION

The emergence of long-context LLMs has spurred development of evaluation frameworks for extended context understanding. General long-context benchmarks include LongBench (Bai et al., 2024b) for bilingual multitask evaluation, RULER (Hsieh et al., 2024) for systematic testing of claimed context sizes revealing performance gaps, ∞-Bench (Zhang et al., 2024a) extending evaluation beyond 100K tokens, HELMET (Yen et al., 2024) for application-centric evaluation at 128K tokens, and LOFT (Lee et al., 2024) pushing evaluation to 1M tokens. LongICLBench (An et al., 2024) evaluates in-context learning capabilities at extreme lengths, while LongAlign (Bai et al., 2024a) addresses instruction following in long contexts. BAMBOO (Dong et al., 2024b) provides comprehensive evaluation across multiple aspects of long-context understanding.

Code-specific long-context evaluation has seen rapid development. LongCodeBench (Rando et al., 2025) evaluates coding LLMs at 1M context windows, demonstrating dramatic performance degradation. LongCodeU (Li et al., 2025) focuses on long code understanding across four aspects. LongCodeArena (Bogomolov et al., 2024) provides code-centric evaluation at 2M+ tokens. RepoQA (Liu et al., 2024) evaluates long-context code understanding through question answering on repositories. SWE-bench-verified (OpenAI et al., 2024) extends real-world evaluation to longer contexts.

However, existing long-context benchmarks primarily focus on natural language tasks such as document summarization and question answering. Even recent code-focused long-context benchmarks concentrate on code comprehension and completion rather than complex multi-file capabilities. The unique challenges of long-context reasoning in software development—including architectural understanding, multi-session development, cross-file refactoring, and maintaining architectural consistency across extended workflows, remain largely unaddressed.

## D.4 LIMITATIONS OF EXISTING APPROACHES

Current code evaluation benchmarks exhibit several critical limitations when applied to complex long-context software development scenarios:

**Scale Limitations:** Most benchmarks contain fewer than 1,000 evaluation instances, providing insufficient coverage for systematic evaluation across languages, difficulty levels, and task types.

**Task Scope:** Existing benchmarks focus primarily on code generation or completion tasks, neglecting crucial long-context capabilities like architectural understanding, cross-file reasoning, and multi-session development.

**Context Length:** Traditional benchmarks operate with short contexts (typically under 10K tokens), failing to test models' ability to understand and operate on realistic codebase sizes.

**Long-Context Metrics:** Current evaluation metrics focus on functional correctness but ignore long-context capabilities like architectural coherence, dependency management, and long-term context retention.

LoCoBench addresses these limitations by providing a comprehensive evaluation framework specifically designed for the unique challenges of complex long-context software development scenarios.

## E    LoCoBench Pipeline Implementation Details

This section provides comprehensive implementation details for LoCoBench's 5-phase pipeline, including real examples from our data generation process and detailed technical specifications for each phase.

### E.1    Phase 1: Project Specification Generation

#### E.1.1    Specification Framework and Structure

Phase 1 generates diverse, realistic project specifications that serve as the foundation for our entire benchmark. Each specification defines a complete software project with detailed requirements, technical constraints, and architectural patterns.

**Technical Implementation:** Our specification generator employs a constraint satisfaction approach to ensure systematic coverage across multiple dimensions while maintaining realistic project characteristics. The generator uses seed-based randomization with deterministic constraints to achieve reproducible diversity.

**Specification Schema:** Each project specification contains structured metadata across multiple dimensions:

```
\{
  "unique_id": "\{language\}_\{domain\}_\{complexity\}_\{index:03d\}",
  "name": "Human-readable project name",
  "description": "Detailed technical description (500+ words)",
  "domain": "Primary domain classification",
  "complexity": "Difficulty level (easy|medium|hard|expert)",
  "language": "Target programming language",
  "architecture": "Architecture pattern (10 options)",
  "theme": "Project theme (8 options)",
  "target_file_count": "Expected number of generated files",
  "target_token_count": "Target context length",
  "features": ["List of 8-15 required features"],
  "architecture_patterns": ["3-7 design patterns to implement"],
  "dependencies": ["Required libraries and frameworks"],
  "seed": "Deterministic randomization seed"
\}
```

**Diverse Project Examples Across Difficulty Levels:**

**Example 1 - Easy Java GraphQL API (Creative Theme):**

```
\{
    "unique_id": "java_api_graphql_easy_007",
    "name": "CanvasQuest GraphQL Studio",
    "description": "A lightweight Java-based API that invites developers and
                    digital artists to generate, remix, and publish storyboard
                    scenes through a single GraphQL endpoint. Each scene is
                    composed of layers (backgrounds, characters, props, text
                    bubbles) that can be queried separately or combined into
                    a rendered composition.",
    "domain": "api_graphql",
    "complexity": "easy",
    "language": "java",
    "architecture": "mvc",
    "theme": "creative",
    "target_file_count": 12,
    "target_token_count": 26600,
    "features": [
      "monitoring", "response_caching", "graphql_schema",
      "request_validation", "logging", "error_handling"
    ],
    "architecture_patterns": [
      "Command_Query_Separation", "REST_Architecture", "Service_Layer"
    ]
\}
```

**Example 2 - Medium Python System Monitoring (Social Theme):**

```
\{
    "unique_id": "python_system_monitoring_medium_061",
    "name": "PulseLink SocialOps Monitor",
    "description": "A medium-scale, Python-powered system monitoring suite
                    designed specifically for social-first applications that
                    run on a constellation of microservices. PulseLink weaves
                    together log aggregation, security scanning, configuration
                    management, performance metrics, deployment automation, and
                    alerting into a single, cohesive solution.",
    "domain": "system_monitoring",
    "complexity": "medium",
    "language": "python",
    "architecture": "microservices",
    "theme": "social",
    "target_file_count": 38,
    "target_token_count": 132415,
    "features": [
      "log_aggregation", "security_scanning", "configuration_management",
      "performance_metrics", "deployment_automation", "alerting"
    ],
    "architecture_patterns": [
      "Chain_of_Responsibility", "Observer_Pattern", "Event_Driven"
    ]
\}
```

**Example 3 - Hard Rust Data Analytics (Healthcare Theme):**

```
    "unique_id": "java_api_graphql_easy_007",
```

```
\{
  "unique_id": "rust_data_analytics_hard_082",
  "name": "PulseScope Analytics Mesh",
  "description": "A serverless, micro-service driven analytics pipeline
                  designed for large hospital systems seeking real-time
                  insight into vital-sign telemetry, laboratory results,
                  and EHR events. Each hospital ward streams HL7/FHIR event
                  data and bedside-device vitals into the mesh where
                  Rust-powered Lambda functions perform high-volume ingestion.",
  "domain": "data_analytics",
  "complexity": "hard",
  "language": "rust",
  "architecture": "serverless",
  "theme": "healthcare",
  "target_file_count": 62,
  "target_token_count": 208666,
  "features": [
    "data_ingestion", "stream_processing", "data_transformation",
    "data_storage", "data_visualization", "data_validation"
  ],
  "architecture_patterns": [
    "Microservices", "Pipeline_Pattern", "Strategy_Pattern", "ETL_Pipeline"
  ]
\}
```

**Example 4 - Expert Java E-commerce (Productivity Theme):**

```
\{
  "unique_id": "java_web_ecommerce_expert_036",
  "name": "SprintCart Pro { Hyper-Productive E-Commerce Workbench",
  "description": "An enterprise-grade e-commerce platform designed for
                  merchants who treat selling as a high-performance workflow.
                  Every user touchpoint is modeled as an optimizable work
                  cycle, complete with real-time analytics and KPI-driven
                  nudges. The core follows strict Hexagonal Architecture.",
  "domain": "web_ecommerce",
  "complexity": "expert",
  "language": "java",
  "architecture": "hexagonal",
  "theme": "productivity",
  "target_file_count": 100,
  "target_token_count": 517323,
  "features": [
    "data_validation", "responsive_design", "api_endpoints",
    "payment_processing", "search_functionality", "caching"
  ],
  "architecture_patterns": [
    "Repository_Pattern", "REST_API", "Service_Layer", "MVC"
  ]
\}
```

### E.1.2  DIVERSITY AND COVERAGE STRATEGY

**Systematic Distribution:**  Our generation strategy ensures balanced coverage across all evaluation dimensions:

- Programming Languages: Exactly 100 specifications per language (10 languages × 100 = 1,000 total)

- Complexity Levels: Equal 25% distribution across easy/medium/hard/expert

- Domain Coverage: Proportional distribution across 36 sub-domains within 10 main categories

- Architecture Patterns: Balanced representation of 10 modern architecture paradigms

- Project Themes: Equal distribution across 8 thematic categories

**Quality Constraints:**  Each specification undergoes automated validation:

- Feature coherence checking (features must align with domain and complexity)

- Architecture pattern compatibility verification

- Dependency resolution and version consistency

- Token count feasibility analysis (based on language-specific file size statistics)

### E.2  PHASE 2: SYNTHETIC CODEBASE GENERATION

#### E.2.1  ARCHITECTURE-AWARE GENERATION STRATEGY

Phase 2 transforms project specifications into complete, executable codebases using sophisticated generation algorithms that ensure architectural coherence and realistic code patterns.

**Multi-File Coordination Algorithm:**   Our generation process maintains consistency across multiple files through a dependency-aware approach:

1. **Architectural Scaffolding**: Generate project structure and primary architectural components

2. **Interface Definition**: Establish APIs and contracts between major modules

3. **Dependency Graph Construction**: Build import/usage relationships before detailed implementation

4. **Progressive Implementation**: Generate files in dependency order, ensuring referential consistency

5. **Integration Verification**: Cross-reference validation to maintain architectural coherence

**Real Generated Structure - Mercantilo E-commerce Suite:**

The Python expert-level e-commerce specification generates a complete Django monolith with 96 files:

```
mercantilo_suite/                     # Django project root
|-- manage.py                         # Django management script
|-- requirements.txt                  # Dependencies specification
|-- Dockerfile                        # Container deployment
|-- docker-compose.yml                # Multi-service orchestration
|-- mercantilo/                       # Django project configuration
|   |-- __init__.py, asgi.py, wsgi.py
|   |-- celery.py                     # Async task configuration
|   |-- settings/                     # Environment-specific configs
|   |   |-- base.py, local.py, production.py, test.py
|   +-- urls.py                       # Main URL routing
+-- apps/                             # Application modules
    |-- accounts/                     # User management
    |   |-- models.py, services.py, views.py, urls.py
    |   |-- repositories.py           # Repository pattern implementation
    |   |-- signals.py                # Django signal handlers
    |   +-- tests/ (3 test modules)
    |-- catalog/                      # Product management
    |   |-- models.py, services.py, search.py, documents.py
    |   |-- repositories.py, tasks.py
    |   +-- tests/ (4 test modules)
    |-- orders/                       # Order processing
    |   |-- models.py, services.py, signals.py
    |   |-- repositories.py
    |   +-- admin.py                  # Django admin interface
    |-- analytics/                    # Business intelligence
    |   |-- models.py, services.py, tasks.py
    |   +-- tests/
    |-- b2b/, crm/, fulfillment/      # Additional business modules
    +-- core/                         # Shared utilities
        |-- middleware.py, models.py
        |-- management/commands/      # Custom Django commands
        +-- utils/ (cache.py, uploads.py)
```

**Architectural Pattern Implementation Examples:**

**Example 1 - Python Event-Driven Architecture (Medium Microservices):**

```python
# PulseLink_SocialOps_Monitor/shared/events.py
class EventBus:
    """
    A small, dependency-free event-bus that powers the internal Observer /
    Pub-Sub communications between PulseLink micro-services.

    The implementation supports both synchronous and asynchronous handlers,
    weakly references subscribers to avoid memory-leaks in long-running daemons.
    """

    def __init__(self):
        self._subscribers = \{\}
        self._async_subscribers = \{\}

    def subscribe(self, event_type: Type[Event], handler: EventHandler):
        """Subscribe to events of a specific type."""
        if event_type not in self._subscribers:
            self._subscribers[event_type] = weakref.WeakSet()
        self._subscribers[event_type].add(handler)

    async def publish(self, event: Event) -> None:
        """Publish event to all subscribers."""
        handlers = self._subscribers.get(type(event), [])
        await asyncio.gather(*[handler(event) for handler in handlers])
```

**Example 2 - Rust Type-Safe Domain Models (Hard Serverless):**

```rust
// pulsescope-analytics-mesh/services/common/src/models.rs
/// Strongly-typed wrapper for FHIR Patient identifiers.
#[derive(Debug, Clone, PartialEq, Eq, Hash, Serialize, Deserialize)]
pub struct PatientId(String);

impl PatientId \{
    pub fn new(id: impl Into<String>) -> Result<Self, ValidationError> \{
        let id = id.into();
        if id.is_empty() || id.len() > 64 \{
            return Err(ValidationError::InvalidFormat("Invalid patient ID"));
        \}
        Ok(PatientId(id))
    \}
\}

/// Core event structure for all analytics pipeline messages
#[derive(Debug, Clone, Serialize, Deserialize)]
pub struct AnalyticsEvent \{
    pub event_id: Uuid,
    pub patient_id: PatientId,
    pub timestamp: DateTime<Utc>,
    pub event_type: EventType,
    pub payload: serde_json::Value,
    pub schema_version: u32,
\}
```

**Example 3 - Java Hexagonal Architecture Domain Model (Expert):**

```java
// sprintcart-pro-domain/src/main/java/com/sprintcart/domain/model/productivity/AutomationRule.java
/**
 * Aggregate root representing a user-defined automation rule.
 *
 * A rule encapsulates:
 * - A set of Conditions that must all evaluate to true to fire
 * - A set of side-effect-free Actions executed in order
 * - Lifecycle controls (activate, pause, archive) for operators
 */
public class AutomationRule implements Serializable {
    private final UUID ruleId;
    private final String name;
    private final List<Condition> conditions;
    private final List<Action> actions;
    private Status status;
    private Instant lastExecuted;

    public AutomationRule(String name, List<Condition> conditions, List<Action> actions) {
        this.ruleId = UUID.randomUUID();
        this.name = Objects.requireNonNull(name);
        this.conditions = new ArrayList<>(Objects.requireNonNull(conditions));
        this.actions = new ArrayList<>(Objects.requireNonNull(actions));
        this.status = Status.DRAFT;
        validateInvariants();
    }

    private void validateInvariants() {
        if (conditions.isEmpty()) {
            throw new IllegalArgumentException("At least one condition required");
        }
        if (actions.isEmpty()) {
            throw new IllegalArgumentException("At least one action required");
        }
    }
}
```

**Example 4 - Java Spring GraphQL Controller (Easy MVC):**

```java
// CanvasQuest/src/main/java/com/canvasquest/controller/SceneController.java
@Controller
public class SceneController {

    private final SceneService sceneService;

    public SceneController(SceneService sceneService) {
        this.sceneService = sceneService;
    }

    @QueryMapping
    public List<Scene> allScenes() {
        return sceneService.getAllScenes();
    }

    @QueryMapping
    public Scene scene(@Argument String id) {
        return sceneService.getSceneById(id);
    }

    @MutationMapping
    public Scene createScene(@Argument CreateSceneInput input) {
        return sceneService.createScene(input);
    }

    @SchemaMapping
    public List<Layer> layers(Scene scene) {
        return sceneService.getLayersForScene(scene.getId());
    }
}
```

### E.2.2   QUALITY ASSURANCE IN GENERATION

**Automated Validation Pipeline:**

- **Syntactic Validation**: Language-specific compilation checks using standard compilers (python -m py_compile, javac, g++, etc.)
- **Import Resolution**: Verification that all imports can be resolved within the generated codebase

- **Architectural Consistency**: Cross-file pattern verification and interface compliance
- **Complexity Metrics**: Cyclomatic complexity measurement and file count verification
- **Documentation Coverage**: Analysis of comment density and docstring completeness

### E.3 PHASE 3: EVALUATION SCENARIO CREATION

#### E.3.1 TASK CATEGORY IMPLEMENTATION AND CONTEXT SELECTION

Phase 3 transforms each generated codebase into 8 evaluation scenarios (one per task category) using intelligent context selection and task-specific prompt engineering.

**Context Selection Algorithm:** Our context selection employs graph-theoretic analysis to identify optimal file subsets:

1. **Dependency Graph Analysis**: Construct directed graph of file dependencies (imports, calls, inheritance)
2. **Centrality Scoring**: Compute PageRank and betweenness centrality to identify architecturally important files
3. **Task-Specific Filtering**: Apply task category filters to prioritize relevant functionality
4. **Information Coverage Optimization**: Balance between information completeness and context length constraints
5. **Difficulty Calibration**: Adjust context complexity based on target difficulty level

**Diverse Scenario Examples Across Task Categories:**

**Example 1 - Feature Implementation (Java GraphQL, Expert):**

```
\{
  "id": "java_api_graphql_easy_007_feature_implementation_expert_01",
  "task_category": "feature_implementation",
  "difficulty": "expert",
  "title": "Implement Query Complexity Analysis for API Rate Limiting",
  "description": "The CanvasQuest GraphQL Studio is experiencing performance
                  degradation due to increasingly complex and deeply nested
                  queries from client applications. A pre-execution query
                  analysis mechanism is required to score incoming GraphQL
                  queries and reject them if they exceed a configurable threshold.",
  "context_files": [
    "CanvasQuest//src//main//java/com/canvasquest//controller//SceneController.java",
    "CanvasQuest//src//main//java/com/canvasquest//service//SceneService.java",
    "CanvasQuest//src//main//java/com/canvasquest//exception//GraphQLExceptionHandler.java"
  ],
  "context_length": 82348,
  "task_prompt": "Implement a query complexity analysis feature that calculates
                  a 'complexity score' for each incoming GraphQL query before
                  execution. Use the standard graphql-java Instrumentation API
                  for integration. The maximum allowed complexity must be
                  configurable via application properties.",
  "expected_approach": "An expert developer would recognize this as a
                        cross-cutting concern handled by intercepting the GraphQL
                        execution process using the Instrumentation interface."
\}
```

**Example 2 - Bug Investigation (Python Microservices, Expert):**

```
\{
  "id": "python_system_monitoring_medium_061_bug_investigation_expert_01",
  "task_category": "bug_investigation",
  "difficulty": "expert",
  "title": "Intermittent Security Scan Failures Due to Silent Log Dropping",
  "description": "The PulseLink SocialOps Monitor generates 'Scan Inconclusive:
                  Log Data Missing' alerts exclusively for servers in the
                  'web-prod-EU' cluster. The log_harvester service reports no
                  errors but other clusters work fine. The problem began after
                  a deployment aimed at improving log parsing efficiency.",
  "context_files": [
    "PulseLink_SocialOps_Monitor//services//log_harvester//service.py",
    "PulseLink_SocialOps_Monitor//shared//patterns.py",
    "PulseLink_SocialOps_Monitor//services//secu_scan//service.py"
  ],
  "context_length": 383018,
  "task_prompt": "Perform a thorough root cause analysis to identify the exact
                  location and cause of missing logs. Trace the data flow from
                  log_harvester to secu_scan services and pinpoint the chain
                  of events from initial defect to final alert.",
  "expected_approach": "An expert would systematically trace from symptom to
                        cause: analyze the alerting logic in secu_scan, trace
                        data sources, investigate the log_harvester producer,
                        and isolate a faulty regex pattern causing silent failures."
\}
```

**Example 3 - Integration Testing (Rust Serverless, Expert):**

```
\{
  "id": "rust_data_analytics_hard_082_integration_testing_expert_01",
  "task_category": "integration_testing",
  "difficulty": "expert",
  "title": "End-to-End Failure Path Integration Test for Sepsis Transform Lambda DLQ",
  "description": "A critical production issue where certain patient data
                  payloads cause the transform-sepsis-lambda to crash due to
                  unhandled data formats. Failed processing events are being
                  lost instead of being routed to a Dead Letter Queue (DLQ),
                  leading to potential data loss and missed clinical alerts.",
  "context_files": [
    "pulsescope-analytics-mesh/services/transform-sepsis-lambda/src/main.rs",
    "pulsescope-analytics-mesh/services/common/src/models.rs",
    "pulsescope-analytics-mesh/infra/lambda.tf"
  ],
  "context_length": 484726,
  "task_prompt": "Implement an integration test that verifies when the lambda
                  encounters a fatal error, the original event payload is
                  correctly routed to its configured Dead Letter Queue. Mock
                  AWS SQS client to intercept DLQ messages and verify exact
                  payload preservation.",
  "expected_approach": "An expert would recognize this as testing integration
                        between Lambda execution environment and failure handling
                        mechanism, requiring simulation of AWS runtime DLQ behavior
                        with proper mocking strategies."
\}
```

**Example 4 - Architectural Understanding (Python E-commerce, Easy):**

```
\{
  "id": "python_web_ecommerce_expert_000_architectural_understanding_easy_01",
  "task_category": "architectural_understanding",
  "difficulty": "easy",
  "title": "Identify the Core Business Logic Abstraction Pattern",
  "description": "A new developer is being onboarded to the Mercantilo team.
                  They must understand the project's fundamental architectural
                  patterns to contribute effectively.",
  "context_files": [
    "mercantilo_suite/apps/accounts/services.py",
    "mercantilo_suite/apps/catalog/services.py",
    "mercantilo_suite/apps/orders/services.py"
  ],
  "context_length": 334348,
  "task_prompt": "Based on the provided files, identify the primary
                  architectural pattern used to organize business logic
                  within each application and explain its benefits.",
  "expected_approach": "An expert developer would notice the consistent
                        presence of services.py files across applications,
                        pointing to the Service Layer pattern."
\}
```

### E.3.2 DIFFICULTY CALIBRATION AND CONTEXT SCALING

**Context Length Scaling Strategy:** Scenarios are systematically calibrated across four difficulty levels:

- **Easy (10K-100K tokens):** Focused file subset with clear architectural indicators

- **Medium (100K-200K tokens):** Moderate codebase coverage requiring deeper analysis

- **Hard (200K-500K tokens):** Extensive multi-module context with complex interactions

- **Expert (500K-1M tokens):** Comprehensive system-wide context requiring sophisticated reasoning

**Task Complexity Progression:**

- **Easy**: Direct pattern identification with explicit indicators

- **Medium**: Multi-step analysis requiring moderate inference

- **Hard**: Complex reasoning across multiple abstractions and modules

- **Expert**: System-wide understanding with subtle architectural relationships

### E.4 PHASE 4: AUTOMATED VALIDATION AND QUALITY ASSURANCE

### E.4.1 COMPREHENSIVE VALIDATION FRAMEWORK

Phase 4 ensures all generated scenarios meet rigorous quality standards through multi-dimensional automated validation.

**Compilation and Execution Validation:**

```
# Language-specific validation pipeline
validation_configs = \{
    "python": \{
        "syntax": ["python -m py_compile \{file\}"],
        "style": ["flake8 --max-line-length=100 \{file\}"],
        "types": ["mypy --strict \{file\}"],
        "security": ["bandit -r \{directory\}"]
    \},
    "java": \{
        "syntax": ["javac -cp \{classpath\} \{file\}"],
        "style": ["checkstyle -c sun_checks.xml \{file\}"],
        "bugs": ["spotbugs -textui \{compiled_class\}"]
    \},
    "cpp": \{
        "syntax": ["g++ -std=c++17 -Wall -Wextra -c \{file\}"],
        "static": ["cppcheck --enable=all \{file\}"],
        "format": ["clang-format --dry-run \{file\}"]
    \}
\}

# Docker-based execution environment
execution_environments = \{
    "python": "python:3.11-slim",
    "java": "openjdk:17-alpine",
    "cpp": "gcc:12-alpine",
    "javascript": "node:18-alpine"
\}
```

**Multi-Language Validation Results:**

**Java GraphQL API (Easy):**

```
validation_results = \{
    "syntax": ["javac -cp spring-boot-starter-graphql:2.7.0 *.java"] → \ding{51} PASS,
    "style": ["checkstyle -c sun_checks.xml *.java"] → \ding{51} PASS (2 warnings),
    "bugs": ["spotbugs -textui compiled_classes/"] → \ding{51} PASS,
    "complexity": \{"avg_cyclomatic": 0.42, "max_depth": 3\} → \ding{51} PASS
\}
```

**Python Microservices (Medium):**

```
validation_results = \{
    "syntax": ["python -m py_compile *.py"] → \ding{51} PASS,
    "style": ["flake8 --max-line-length=100 *.py"] → \ding{51} PASS (5 warnings),
    "types": ["mypy --strict services/"] → $\blacktriangleright$ PARTIAL (3 type hints missing),
    "security": ["bandit -r services/"] → \ding{51} PASS,
    "complexity": \{"avg_cyclomatic": 0.73, "max_depth": 4\} → \ding{51} PASS
\}
```

**Rust Serverless (Hard):**

```
validation_results = \{
    "syntax": ["cargo check --all-targets"] → \ding{51} PASS,
    "static": ["cargo clippy -- -D warnings"] → \ding{51} PASS,
    "format": ["cargo fmt --check"] → \ding{51} PASS,
    "tests": ["cargo test --all"] → \ding{51} PASS (47/47 tests),
    "complexity": \{"avg_cyclomatic": 0.89, "max_depth": 5\} → \ding{51} PASS
\}
```

**Information Coverage Analysis:**  For each scenario, we compute comprehensive coverage metrics:

- **Relevant Information Ratio**: Fraction of context directly applicable to the task ($R = \frac{relevant\_tokens}{total\_tokens}$)
- **Redundancy Analysis**: Detection of duplicate or highly similar code segments

- **Completeness Assessment**: Verification that sufficient information exists for task completion

- **Distractor Balance**: Appropriate amount of realistic but irrelevant information (target: 20-30%)

**Bias Detection and Filtering:** Automated analysis identifies and filters potential biases:

- **Generation Artifacts**: Detection of unrealistic patterns (e.g., overly regular naming conventions)

- **Structural Uniformity**: Identification of artificially systematic file organization

- **Content Repetition**: Copy-paste detection using fuzzy string matching

- **Language Bias**: Verification of language-appropriate idioms and conventions

### E.5 Phase 5: LLM Evaluation and Comprehensive Scoring

#### E.5.1 Multi-Model Evaluation Infrastructure

Phase 5 implements a robust evaluation infrastructure supporting diverse LLM architectures with standardized assessment protocols.

**Model Integration Framework:**

```
# Comprehensive model configuration matrix
model_configurations = \{
    "openai": \{
        "models": ["gpt-4", "gpt-4-turbo", "gpt-4o", "gpt-4-0125-preview"],
        "max_tokens": [8192, 128000, 128000, 128000],
        "rate_limits": \{"requests_per_minute": 500, "tokens_per_minute": 150000\}
    \},
    "anthropic": \{
        "models": ["claude-3-haiku", "claude-3-sonnet", "claude-3-opus",
                "claude-3.5-sonnet", "claude-4-sonnet", "claude-4-opus"],
        "max_tokens": [200000, 200000, 200000, 200000, 1000000, 1000000],
        "rate_limits": \{"requests_per_minute": 1000, "tokens_per_minute": 3500000\}
    \},
    "google": \{
        "models": ["gemini-1.5-pro", "gemini-1.5-flash", "gemini-2.0-flash"],
        "max_tokens": [2097152, 1048576, 1048576],
        "rate_limits": \{"requests_per_minute": 360, "tokens_per_minute": 4000000\}
    \}
\}
```

**Evaluation Pipeline Implementation:**

1. **Context Preparation**: Intelligent truncation for models with limited context windows using importance-based ranking

2. **Prompt Engineering**: Task-specific prompting strategies optimized for each model family

3. **Parallel Execution**: Concurrent evaluation with configurable timeout (3600 seconds) and error recovery

4. **Multi-Metric Assessment**: Comprehensive scoring across all 17 evaluation metrics

5. **Statistical Analysis**: Confidence interval computation and significance testing

#### E.5.2 Comprehensive Benchmark Statistics

**Multi-Model Evaluation Results Across Difficulty Levels:**

**Easy Level Performance (10K-100K tokens):**

```
model_performance = \{
    "GPT-4o": \{"success_rate": 0.847, "avg_lcbs": 3.92, "compilation": 0.923\},
    "Claude-4-Sonnet": \{"success_rate": 0.834, "avg_lcbs": 3.89, "compilation": 0.918\},
    "Gemini-2.5-Pro": \{"success_rate": 0.798, "avg_lcbs": 3.71, "compilation": 0.901\},
    "GPT-4-Turbo": \{"success_rate": 0.776, "avg_lcbs": 3.58, "compilation": 0.887\}
\}
```

**Expert Level Performance (500K-1M tokens):**

```
model_performance = \{
    "GPT-4o": \{"success_rate": 0.412, "avg_lcbs": 2.18, "compilation": 0.634\},
    "Claude-4-Sonnet": \{"success_rate": 0.398, "avg_lcbs": 2.09, "compilation": 0.621\},
    "Gemini-2.5-Pro": \{"success_rate": 0.356, "avg_lcbs": 1.87, "compilation": 0.578\},
    "GPT-4-Turbo": \{"success_rate": 0.289, "avg_lcbs": 1.52, "compilation": 0.498\}
\}
```

**Task Category Performance Variations:**

```
task_performance = \{
    "code_comprehension": \{"avg_success": 0.723, "best_model": "GPT-4o"\},
    "feature_implementation": \{"avg_success": 0.542, "best_model": "Claude-4-Sonnet"\},
    "architectural_understanding": \{"avg_success": 0.687, "best_model": "GPT-4o"\},
    "bug_investigation": \{"avg_success": 0.398, "best_model": "Claude-4-Sonnet"\},
    "integration_testing": \{"avg_success": 0.312, "best_model": "GPT-4o"\},
    "security_analysis": \{"avg_success": 0.289, "best_model": "Claude-4-Sonnet"\}
\}
```

**Quality Validation Results:**

- **Compilation Success Rate**: 98.7% across all languages and complexity levels

- **Average Cyclomatic Complexity**: 0.67 (realistic for production codebases)

- **Documentation Coverage**: 85% (exceeds typical industry standards of 60-70%)

- **Test Coverage**: 78% (comprehensive test suites with realistic coverage patterns)

- **Architectural Consistency**: 94% pattern adherence validation success

## E.6    PROMPT ENGINEERING AND TEMPLATES

### E.6.1    SCENARIO GENERATION PROMPTS

LoCoBench employs sophisticated prompt engineering throughout its pipeline, with task-specific templates for each phase. The scenario generation process uses structured prompts that adapt to different task categories and difficulty levels.

**Master Scenario Generation Template:**

```
Create a realistic \{task_category\} evaluation scenario for long-context LLMs.

PROJECT CONTEXT:
- Name: \{project_name\}
- Language: \{programming_language\}
- Domain: \{project_domain\}
- Features: \{key_features\}
- Complexity: \{complexity_level\}

AVAILABLE FILES:
\{context_file_summary\}

TASK REQUIREMENTS:
- Category: \{task_category\}
- Difficulty: \{difficulty_level\}
- Must be realistic and challenging for long-context LLMs
- Should require understanding of multiple files
- Include specific, measurable objectives

Generate a JSON response with these fields:
\{
    "title": "Clear, descriptive title for the task",
    "description": "Detailed description of the scenario and context",
    "task_prompt": "Specific task instructions for the LLM",
    "expected_approach": "How an expert developer would approach this task",
    "ground_truth": "Expected solution or key insights",
    "evaluation_criteria": ["List of criteria to evaluate performance"]
\}

Make the scenario realistic and challenging. Focus on \{category_focus\}.
```

**Task Category Focus Areas:**

Each task category employs specialized focus areas to ensure targeted evaluation:

```
category_focus_map = \{
    "architectural_understanding":
        "system design patterns, component relationships, and architectural decisions",
    "cross_file_refactoring":
        "code restructuring across multiple files while maintaining functionality",
    "feature_implementation":
        "adding new functionality that integrates well with existing code",
    "bug_investigation":
        "systematic debugging, root cause analysis, and problem solving",
    "multi_session_development":
        "incremental development over multiple sessions with context retention",
    "code_comprehension":
        "deep understanding of complex code structures and logic",
    "integration_testing":
        "testing interactions between components and system validation",
    "security_analysis":
        "identifying security vulnerabilities and implementing security best practices"
\}
```

### E.6.2   LLM EVALUATION PROMPTS

When evaluating LLMs on generated scenarios, LoCoBench employs language-aware prompts that adapt to different programming languages and provide comprehensive guidance.

**Solution Generation Template:**

```
You are an expert \{language\} engineer. Your task is to provide a complete, working solution.

**TASK**: \{scenario_title\}

**DESCRIPTION**: \{scenario_description\}

**REQUIREMENTS**:
\{formatted_task_requirements\}

**CONTEXT FILES**: \{available_context_files\}

**CRITICAL INSTRUCTIONS**:
1. You MUST respond with valid JSON in the exact format shown below
2. Each file MUST contain complete, syntactically correct \{LANGUAGE\} code
3. Do NOT truncate your response - provide the complete solution
4. Use \{language_specific_best_practices\}

**REQUIRED RESPONSE FORMAT**:
```json
\{
    "approach": "Your solution strategy (keep under 200 words)",
    "files": \{
        "filename1.\{ext\}": "complete file content with proper escaping",
        "filename2.\{ext\}": "complete file content with proper escaping"
    \},
    "explanation": "Implementation details (keep under 300 words)"
\}
```

**VALIDATION CHECKLIST**:
- \ding{51} Response is valid JSON wrapped in ```json blocks
- \ding{51} All strings are properly escaped (\\n for newlines, \\" for quotes)
- \ding{51} Each file contains complete \{LANGUAGE\} code
- \ding{51} Code compiles and addresses all requirements
- \ding{51} Response is complete (not truncated)

Generate your response now:
```

### E.6.3    MULTI-SESSION DEVELOPMENT PROMPTS

For multi-session development scenarios, LoCoBench employs sophisticated context management with session-specific prompting:

**Multi-Session Prompt Structure:**

```
**SESSION CONTEXT**: You are continuing development from a previous session.

**PREVIOUS SESSION SUMMARY**:
\{previous_session_context\}

**CURRENT SESSION OBJECTIVE**:
\{current_session_task\}

**DEVELOPMENT HISTORY**:
- Session 1: \{session_1_summary\}
- Session 2: \{session_2_summary\}
- Current: \{current_session_description\}

**CONTEXT RETENTION REQUIREMENTS**:
- Maintain consistency with previous architectural decisions
- Build upon existing implementation patterns
- Preserve naming conventions and code style
- Reference relevant previous session outcomes

**INCREMENTAL DEVELOPMENT GUIDELINES**:
- Extend existing functionality rather than rewriting
- Ensure backward compatibility where applicable
- Document changes and rationale for future sessions
- Test integration with existing components
```

### E.6.4    LANGUAGE-SPECIFIC ADAPTATIONS

LoCoBench adapts its prompts based on programming language characteristics and best practices:

```
language_configs = \{
    "python": \{
        "engineer": "Python developer",
        "practices": "PEP 8 style, type hints, docstrings, and proper error handling",
        "file_examples": '"main.py": "# Complete Python implementation",
                         "utils.py": "# Helper functions and utilities"'
    \},
    "java": \{
        "engineer": "Java developer",
        "practices": "clean code principles, proper OOP design, and comprehensive JavaDoc",
        "file_examples": '"Main.java": "// Complete Java implementation",
                         "Utils.java": "// Helper classes and methods"'
    \},
    "cpp": \{
        "engineer": "C++ developer",
        "practices": "modern C++17/20 features, RAII, and proper memory management",
        "file_examples": '"main.cpp": "// Complete C++ implementation",
                         "utils.hpp": "// Header declarations"'
    \}
\}
```

These sophisticated prompt templates ensure consistent, high-quality evaluation across all programming languages and task categories while maintaining the flexibility needed for comprehensive long-context assessment.

## F  MORE EXPERIMENTAL RESULTS

This appendix presents the complete experimental results, containing all evaluation metrics for all 13 models.

### F.1  OVERALL MODEL PERFORMANCE RESULTS

Table 7 presents detailed comparison of model performance results, covering all 32 columns of evaluation data for all 13 models.

### F.2  PERFORMANCE BY DIFFICULTY LEVEL

Table 8 presents model performance across four difficulty levels from Easy (10K-100K tokens) to Expert (500K-1M tokens).

### F.3  PERFORMANCE BY PROGRAMMING LANGUAGE

Table 9 presents the complete results of showing model performance across 10 programming languages.

### F.4  PERFORMANCE BY TASK CATEGORY

Table 11 presents the complete results of performance across 8 software engineering task categories.

### F.5  PERFORMANCE BY APPLICATION DOMAIN

Table 12 presents the complete results of model performance across different application domains.

### F.6  PERFORMANCE BY ARCHITECTURE PATTERN

Table 14 presents the complete results of model performance across different architectural patterns.

### F.7  PERFORMANCE BY THEME

Table 16 presents the complete results of model performance across different thematic areas.

Table 7: Detailed comparison of model performance results.

**(a) Overall Performance Summary and Core Dimensions**

| Model | LCBS | Success Rate (%) | SE Overall | Functional Overall | Quality Overall | Long-Context Overall |
|---|---|---|---|---|---|---|
| Gemini-2.5-Pro | 2.312 | 99.88 | 0.375 | 0.356 | 0.768 | 0.523 |
| Gemini-2.5-Flash | 2.307 | 99.98 | 0.373 | 0.358 | 0.741 | 0.565 |
| gpt5mini | 2.293 | 100.00 | 0.376 | 0.371 | 0.745 | 0.479 |
| claudesonnet4 | 2.288 | 99.56 | 0.379 | 0.348 | 0.762 | 0.492 |
| GPT-5 | 2.286 | 100.00 | 0.367 | 0.383 | 0.732 | 0.492 |
| claudesonnet3.7 | 2.285 | 99.79 | 0.377 | 0.347 | 0.773 | 0.477 |
| GPT-4.1-mini | 2.222 | 100.00 | 0.359 | 0.365 | 0.739 | 0.435 |
| o3-mini | 2.215 | 100.00 | 0.355 | 0.368 | 0.726 | 0.455 |
| GPT-4.1-2025-04-14 | 2.197 | 100.00 | 0.352 | 0.364 | 0.720 | 0.451 |
| o3 | 2.154 | 100.00 | 0.342 | 0.385 | 0.722 | 0.343 |
| o4-mini | 2.148 | 99.70 | 0.353 | 0.360 | 0.705 | 0.394 |
| GPT-4o-mini | 2.075 | 100.00 | 0.341 | 0.360 | 0.680 | 0.345 |
| GPT-4o | 2.073 | 100.00 | 0.339 | 0.362 | 0.678 | 0.349 |

**(b) Software Engineering Core Metrics (Part 1)**

| Model | ACS Overall | DTA Overall | CFRD Overall | STS Overall | RS Overall | CS Overall | IS Overall |
|---|---|---|---|---|---|---|---|
| Gemini-2.5-Pro | 0.693 | 0.367 | 0.378 | 0.311 | 0.243 | 0.238 | 0.164 |
| Gemini-2.5-Flash | 0.664 | 0.353 | 0.369 | 0.334 | 0.262 | 0.239 | 0.174 |
| gpt5mini | 0.698 | 0.357 | 0.408 | 0.288 | 0.278 | 0.220 | 0.144 |
| claudesonnet4 | 0.709 | 0.362 | 0.413 | 0.300 | 0.264 | 0.224 | 0.165 |
| GPT-5 | 0.676 | 0.360 | 0.357 | 0.281 | 0.270 | 0.224 | 0.149 |
| claudesonnet3.7 | 0.705 | 0.350 | 0.423 | 0.295 | 0.268 | 0.215 | 0.149 |
| GPT-4.1-mini | 0.661 | 0.379 | 0.341 | 0.268 | 0.262 | 0.203 | 0.132 |
| o3-mini | 0.654 | 0.363 | 0.352 | 0.276 | 0.262 | 0.203 | 0.110 |
| GPT-4.1-2025-04-14 | 0.647 | 0.366 | 0.316 | 0.273 | 0.258 | 0.207 | 0.128 |
| o3 | 0.618 | 0.343 | 0.324 | 0.235 | 0.267 | 0.205 | 0.100 |
| o4-mini | 0.657 | 0.360 | 0.332 | 0.270 | 0.262 | 0.201 | 0.119 |
| GPT-4o-mini | 0.628 | 0.361 | 0.322 | 0.240 | 0.265 | 0.181 | 0.096 |
| GPT-4o | 0.628 | 0.362 | 0.318 | 0.238 | 0.260 | 0.178 | 0.086 |

**(c) Software Engineering Core Metrics (Part 2) and Quality Metrics**

| Model | SES Overall | ICU Overall | MMR Overall | Compilation | Unit Tests | Integration | Overall Quality |
|---|---|---|---|---|---|---|---|
| Gemini-2.5-Pro | 0.606 | 0.498 | 0.549 | 0.287 | 0.200 | 0.635 | 0.769 |
| Gemini-2.5-Flash | 0.592 | 0.540 | 0.589 | 0.280 | 0.200 | 0.656 | 0.741 |
| gpt5mini | 0.617 | 0.450 | 0.508 | 0.379 | 0.200 | 0.553 | 0.745 |
| claudesonnet4 | 0.607 | 0.466 | 0.522 | 0.282 | 0.199 | 0.609 | 0.766 |
| GPT-5 | 0.618 | 0.465 | 0.520 | 0.404 | 0.200 | 0.602 | 0.732 |
| claudesonnet3.7 | 0.616 | 0.449 | 0.508 | 0.311 | 0.199 | 0.542 | 0.774 |
| GPT-4.1-mini | 0.626 | 0.404 | 0.466 | 0.343 | 0.200 | 0.637 | 0.739 |
| o3-mini | 0.623 | 0.429 | 0.480 | 0.363 | 0.200 | 0.602 | 0.726 |
| GPT-4.1-2025-04-14 | 0.623 | 0.422 | 0.481 | 0.338 | 0.200 | 0.652 | 0.720 |
| o3 | 0.643 | 0.313 | 0.372 | 0.493 | 0.200 | 0.513 | 0.722 |
| o4-mini | 0.633 | 0.367 | 0.423 | 0.365 | 0.200 | 0.585 | 0.707 |
| GPT-4o-mini | 0.639 | 0.315 | 0.374 | 0.348 | 0.200 | 0.655 | 0.680 |
| GPT-4o | 0.641 | 0.317 | 0.380 | 0.358 | 0.200 | 0.642 | 0.678 |

**(d) Task-Specific Average Scores**

| Model | Architectural | Bug Investigation | Code Comprehension | Cross-File Refactoring | Feature Implementation | Integration Testing | Multi-Session Dev. | Security Analysis |
|---|---|---|---|---|---|---|---|---|
| Gemini-2.5-Pro | 2.338 | 2.272 | 2.273 | 2.272 | 2.299 | 2.421 | 2.280 | 2.343 |
| Gemini-2.5-Flash | 2.280 | 2.276 | 2.211 | 2.303 | 2.335 | 2.430 | 2.291 | 2.325 |
| gpt5mini | 2.370 | 2.227 | 2.218 | 2.237 | 2.285 | 2.390 | 2.268 | 2.351 |
| claudesonnet4 | 2.346 | 2.262 | 2.298 | 2.203 | 2.227 | 2.402 | 2.256 | 2.307 |
| GPT-5 | 2.376 | 2.196 | 2.199 | 2.241 | 2.297 | 2.386 | 2.257 | 2.336 |
| claudesonnet3.7 | 2.332 | 2.206 | 2.228 | 2.269 | 2.289 | 2.403 | 2.273 | 2.322 |
| GPT-4.1-mini | 2.238 | 2.192 | 2.206 | 2.181 | 2.220 | 2.363 | 2.152 | 2.226 |
| o3-mini | 2.231 | 2.165 | 2.237 | 2.156 | 2.205 | 2.330 | 2.182 | 2.214 |
| GPT-4.1-2025-04-14 | 2.195 | 2.154 | 2.166 | 2.169 | 2.207 | 2.346 | 2.148 | 2.191 |
| o3 | 2.123 | 2.058 | 2.010 | 2.151 | 2.192 | 2.323 | 2.181 | 2.197 |
| o4-mini | 2.148 | 2.094 | 2.088 | 2.146 | 2.166 | 2.277 | 2.134 | 2.131 |
| GPT-4o-mini | 2.059 | 2.043 | 2.054 | 2.054 | 2.057 | 2.184 | 2.089 | 2.061 |
| GPT-4o | 2.054 | 2.038 | 2.051 | 2.055 | 2.063 | 2.200 | 2.068 | 2.056 |

Table 8: Detailed comparison of model performance by difficulty level.

**(a) Total Scores by Difficulty Level**

| Model | Easy Overall | Medium Overall | Hard Overall | Expert Overall | Overall |
|-------|-------------|----------------|--------------|----------------|---------|
| Gemini-2.5-Pro | 2.278 | 2.302 | 2.329 | 2.339 | 2.312 |
| Gemini-2.5-Flash | 2.291 | 2.299 | 2.319 | 2.317 | 2.307 |
| gpt5mini | 2.263 | 2.284 | 2.311 | 2.314 | 2.293 |
| claudesonnet4 | 2.309 | 2.289 | 2.283 | 2.269 | 2.288 |
| GPT-5 | 2.254 | 2.268 | 2.298 | 2.323 | 2.286 |
| claudesonnet3.7 | 2.326 | 2.299 | 2.256 | 2.262 | 2.285 |
| GPT-4.1-mini | 2.218 | 2.219 | 2.227 | 2.224 | 2.222 |
| o3-mini | 2.232 | 2.216 | 2.214 | 2.199 | 2.215 |
| GPT-4.1-2025-04-14 | 2.194 | 2.194 | 2.205 | 2.195 | 2.197 |
| o3 | 2.086 | 2.149 | 2.187 | 2.195 | 2.154 |
| o4-mini | 2.129 | 2.154 | 2.154 | 2.159 | 2.148 |
| GPT-4o-mini | 2.059 | 2.077 | 2.085 | 2.080 | 2.075 |
| GPT-4o | 2.044 | 2.081 | 2.083 | 2.084 | 2.073 |

**(b) Software Engineering Scores by Difficulty Level**

| Model | Easy Overall | Medium Overall | Hard Overall | Expert Overall | Overall |
|-------|-------------|----------------|--------------|----------------|---------|
| Gemini-2.5-Pro | 0.366 | 0.371 | 0.379 | 0.382 | 0.375 |
| Gemini-2.5-Flash | 0.364 | 0.368 | 0.379 | 0.381 | 0.373 |
| gpt5mini | 0.369 | 0.371 | 0.378 | 0.381 | 0.376 |
| claudesonnet4 | 0.390 | 0.384 | 0.377 | 0.368 | 0.379 |
| GPT-5 | 0.358 | 0.365 | 0.370 | 0.376 | 0.367 |
| claudesonnet3.7 | 0.383 | 0.383 | 0.374 | 0.370 | 0.377 |
| GPT-4.1-mini | 0.350 | 0.355 | 0.364 | 0.367 | 0.359 |
| o3-mini | 0.359 | 0.356 | 0.354 | 0.350 | 0.355 |
| GPT-4.1-2025-04-14 | 0.346 | 0.349 | 0.356 | 0.358 | 0.352 |
| o3 | 0.311 | 0.345 | 0.356 | 0.356 | 0.342 |
| o4-mini | 0.343 | 0.357 | 0.354 | 0.358 | 0.353 |
| GPT-4o-mini | 0.332 | 0.342 | 0.345 | 0.345 | 0.341 |
| GPT-4o | 0.325 | 0.342 | 0.344 | 0.346 | 0.339 |

**(c) Long-Context Scores by Difficulty Level**

| Model | Easy Overall | Medium Overall | Hard Overall | Expert Overall | Overall |
|-------|-------------|----------------|--------------|----------------|---------|
| Gemini-2.5-Pro | 0.500 | 0.538 | 0.527 | 0.525 | 0.523 |
| Gemini-2.5-Flash | 0.537 | 0.581 | 0.574 | 0.564 | 0.565 |
| gpt5mini | 0.434 | 0.470 | 0.492 | 0.518 | 0.479 |
| claudesonnet4 | 0.473 | 0.491 | 0.505 | 0.498 | 0.492 |
| GPT-5 | 0.456 | 0.485 | 0.500 | 0.528 | 0.492 |
| claudesonnet3.7 | 0.447 | 0.475 | 0.485 | 0.501 | 0.477 |
| GPT-4.1-mini | 0.395 | 0.430 | 0.446 | 0.469 | 0.435 |
| o3-mini | 0.423 | 0.449 | 0.465 | 0.482 | 0.455 |
| GPT-4.1-2025-04-14 | 0.413 | 0.445 | 0.462 | 0.481 | 0.451 |
| o3 | 0.263 | 0.340 | 0.374 | 0.397 | 0.343 |
| o4-mini | 0.344 | 0.397 | 0.404 | 0.431 | 0.394 |
| GPT-4o-mini | 0.307 | 0.344 | 0.356 | 0.373 | 0.345 |
| GPT-4o | 0.309 | 0.348 | 0.360 | 0.378 | 0.349 |

Table 9: Detailed comparison of model performance by programming language.

**(a) Total Scores by Programming Language**

| Model | Python | C++ | Java | C | C# | JavaScript | TypeScript | Go | Rust | PHP |
|---|---|---|---|---|---|---|---|---|---|---|
| Gemini-2.5-Pro | 2.788 | 2.074 | 2.326 | 2.064 | 2.419 | 2.268 | 2.203 | 2.338 | 2.002 | 2.641 |
| Gemini-2.5-Flash | 2.752 | 2.106 | 2.329 | 2.086 | 2.418 | 2.274 | 2.248 | 2.292 | 2.039 | 2.573 |
| gpt5mini | 2.799 | 2.039 | 2.329 | 2.050 | 2.414 | 2.280 | 2.189 | 2.264 | 2.088 | 2.476 |
| claudesonnet4 | 2.677 | 2.065 | 2.331 | 2.054 | 2.424 | 2.314 | 2.154 | 2.311 | 1.997 | 2.553 |
| GPT-5 | 2.669 | 2.001 | 2.281 | 2.022 | 2.335 | 2.244 | 2.098 | 2.249 | 2.044 | 2.516 |
| claudesonnet3.7 | 2.663 | 2.100 | 2.314 | 2.062 | 2.364 | 2.245 | 2.162 | 2.298 | 2.000 | 2.641 |
| GPT-4.1-mini | 2.538 | 1.968 | 2.249 | 1.978 | 2.303 | 2.218 | 2.111 | 2.212 | 1.958 | 2.680 |
| o3-mini | 2.529 | 1.977 | 2.223 | 1.958 | 2.324 | 2.186 | 2.135 | 2.209 | 1.977 | 2.632 |
| GPT-4.1-2025-04-14 | 2.517 | 1.962 | 2.193 | 1.957 | 2.280 | 2.182 | 2.075 | 2.197 | 1.942 | 2.661 |
| o3 | 2.432 | 1.944 | 2.206 | 1.918 | 2.298 | 2.086 | 2.069 | 2.167 | 1.943 | 2.479 |
| o4-mini | 2.465 | 1.941 | 2.184 | 1.918 | 2.243 | 2.159 | 2.052 | 2.174 | 1.910 | 2.432 |
| GPT-4o-mini | 2.321 | 1.933 | 2.104 | 1.876 | 2.177 | 2.068 | 1.996 | 2.077 | 1.863 | 2.336 |
| GPT-4o | 2.313 | 1.921 | 2.110 | 1.864 | 2.187 | 2.063 | 1.995 | 2.078 | 1.864 | 2.335 |

**(b) Success Rates by Programming Language (%)**

| Model | Python | C++ | Java | C | C# | JavaScript | TypeScript | Go | Rust | PHP |
|---|---|---|---|---|---|---|---|---|---|---|
| Gemini-2.5-Pro | 99.75 | 99.88 | 99.88 | 99.88 | 99.75 | 100.00 | 100.00 | 99.75 | 100.00 | 99.88 |
| Gemini-2.5-Flash | 100.00 | 100.00 | 100.00 | 100.00 | 100.00 | 100.00 | 100.00 | 100.00 | 100.00 | 99.75 |
| gpt5mini | 100.00 | 100.00 | 100.00 | 100.00 | 100.00 | 100.00 | 100.00 | 100.00 | 100.00 | 100.00 |
| claudesonnet4 | 99.50 | 99.63 | 99.75 | 99.50 | 99.50 | 99.50 | 99.75 | 99.63 | 99.50 | 99.50 |
| GPT-5 | 100.00 | 100.00 | 100.00 | 100.00 | 100.00 | 100.00 | 100.00 | 100.00 | 100.00 | 100.00 |
| claudesonnet3.7 | 99.88 | 99.63 | 99.88 | 99.75 | 99.75 | 99.75 | 99.88 | 99.88 | 99.75 | 99.88 |
| GPT-4.1-mini | 100.00 | 100.00 | 100.00 | 100.00 | 100.00 | 100.00 | 100.00 | 100.00 | 100.00 | 100.00 |
| o3-mini | 100.00 | 100.00 | 100.00 | 100.00 | 100.00 | 100.00 | 100.00 | 100.00 | 100.00 | 100.00 |
| GPT-4.1-2025-04-14 | 100.00 | 100.00 | 100.00 | 100.00 | 100.00 | 100.00 | 100.00 | 100.00 | 100.00 | 100.00 |
| o3 | 100.00 | 100.00 | 100.00 | 100.00 | 100.00 | 100.00 | 100.00 | 100.00 | 100.00 | 100.00 |
| o4-mini | 99.25 | 100.00 | 100.00 | 100.00 | 99.75 | 99.75 | 100.00 | 99.75 | 99.75 | 99.25 |
| GPT-4o-mini | 100.00 | 100.00 | 100.00 | 100.00 | 100.00 | 100.00 | 100.00 | 100.00 | 100.00 | 100.00 |
| GPT-4o | 100.00 | 100.00 | 100.00 | 100.00 | 100.00 | 100.00 | 100.00 | 100.00 | 100.00 | 100.00 |

**(c) Software Engineering Scores by Programming Language**

| Model | Python | C++ | Java | C | C# | JavaScript | TypeScript | Go | Rust | PHP |
|---|---|---|---|---|---|---|---|---|---|---|
| Gemini-2.5-Pro | 0.485 | 0.340 | 0.395 | 0.321 | 0.375 | 0.368 | 0.380 | 0.379 | 0.342 | 0.361 |
| Gemini-2.5-Flash | 0.483 | 0.343 | 0.395 | 0.324 | 0.375 | 0.369 | 0.383 | 0.371 | 0.345 | 0.356 |
| gpt5mini | 0.486 | 0.340 | 0.395 | 0.322 | 0.375 | 0.370 | 0.378 | 0.374 | 0.348 | 0.352 |
| claudesonnet4 | 0.488 | 0.342 | 0.397 | 0.324 | 0.377 | 0.372 | 0.376 | 0.377 | 0.344 | 0.357 |
| GPT-5 | 0.476 | 0.335 | 0.385 | 0.318 | 0.365 | 0.362 | 0.370 | 0.368 | 0.340 | 0.348 |
| claudesonnet3.7 | 0.485 | 0.344 | 0.394 | 0.325 | 0.373 | 0.368 | 0.378 | 0.376 | 0.344 | 0.361 |
| GPT-4.1-mini | 0.467 | 0.329 | 0.381 | 0.314 | 0.359 | 0.358 | 0.366 | 0.363 | 0.332 | 0.371 |
| o3-mini | 0.465 | 0.330 | 0.377 | 0.312 | 0.361 | 0.355 | 0.368 | 0.362 | 0.335 | 0.365 |
| GPT-4.1-2025-04-14 | 0.463 | 0.328 | 0.372 | 0.311 | 0.356 | 0.354 | 0.364 | 0.361 | 0.330 | 0.369 |
| o3 | 0.448 | 0.325 | 0.374 | 0.307 | 0.358 | 0.340 | 0.363 | 0.356 | 0.330 | 0.344 |
| o4-mini | 0.454 | 0.324 | 0.370 | 0.307 | 0.350 | 0.351 | 0.360 | 0.357 | 0.325 | 0.337 |
| GPT-4o-mini | 0.428 | 0.323 | 0.356 | 0.300 | 0.340 | 0.336 | 0.350 | 0.341 | 0.317 | 0.324 |
| GPT-4o | 0.426 | 0.321 | 0.357 | 0.298 | 0.341 | 0.335 | 0.350 | 0.341 | 0.317 | 0.324 |

**(d) Functional Scores by Programming Language**

| Model | Python | C++ | Java | C | C# | JavaScript | TypeScript | Go | Rust | PHP |
|---|---|---|---|---|---|---|---|---|---|---|
| Gemini-2.5-Pro | 0.507 | 0.216 | 0.380 | 0.213 | 0.441 | 0.365 | 0.309 | 0.299 | 0.210 | 0.618 |
| Gemini-2.5-Flash | 0.505 | 0.219 | 0.380 | 0.216 | 0.441 | 0.366 | 0.312 | 0.291 | 0.213 | 0.613 |
| gpt5mini | 0.508 | 0.215 | 0.380 | 0.213 | 0.441 | 0.367 | 0.307 | 0.295 | 0.217 | 0.609 |
| claudesonnet4 | 0.489 | 0.217 | 0.382 | 0.215 | 0.443 | 0.369 | 0.305 | 0.298 | 0.213 | 0.616 |
| GPT-5 | 0.487 | 0.210 | 0.370 | 0.208 | 0.431 | 0.359 | 0.297 | 0.289 | 0.208 | 0.604 |
| claudesonnet3.7 | 0.486 | 0.219 | 0.381 | 0.217 | 0.440 | 0.365 | 0.309 | 0.297 | 0.213 | 0.618 |
| GPT-4.1-mini | 0.468 | 0.204 | 0.366 | 0.204 | 0.426 | 0.354 | 0.293 | 0.285 | 0.202 | 0.627 |
| o3-mini | 0.466 | 0.205 | 0.362 | 0.202 | 0.428 | 0.351 | 0.295 | 0.283 | 0.205 | 0.621 |
| GPT-4.1-2025-04-14 | 0.464 | 0.203 | 0.357 | 0.201 | 0.423 | 0.350 | 0.291 | 0.282 | 0.200 | 0.625 |
| o3 | 0.448 | 0.201 | 0.360 | 0.197 | 0.425 | 0.334 | 0.289 | 0.276 | 0.200 | 0.599 |
| o4-mini | 0.454 | 0.200 | 0.356 | 0.197 | 0.418 | 0.347 | 0.287 | 0.279 | 0.195 | 0.587 |
| GPT-4o-mini | 0.428 | 0.199 | 0.342 | 0.193 | 0.408 | 0.332 | 0.273 | 0.263 | 0.187 | 0.567 |
| GPT-4o | 0.426 | 0.197 | 0.343 | 0.191 | 0.409 | 0.331 | 0.273 | 0.263 | 0.187 | 0.567 |

Table 10: Detailed comparison of model performance by programming language (continued).

**(e) Quality Scores by Programming Language**

| Model | Python | C++ | Java | C | C# | JavaScript | TypeScript | Go | Rust | PHP |
|---|---|---|---|---|---|---|---|---|---|---|
| Gemini-2.5-Pro | 0.795 | 0.801 | 0.709 | 0.826 | 0.742 | 0.746 | 0.735 | 0.846 | 0.750 | 0.731 |
| Gemini-2.5-Flash | 0.793 | 0.804 | 0.709 | 0.829 | 0.742 | 0.747 | 0.738 | 0.838 | 0.753 | 0.726 |
| gpt5mini | 0.796 | 0.800 | 0.709 | 0.825 | 0.742 | 0.748 | 0.733 | 0.841 | 0.757 | 0.722 |
| claudesonnet4 | 0.778 | 0.802 | 0.711 | 0.827 | 0.744 | 0.750 | 0.731 | 0.849 | 0.748 | 0.729 |
| GPT-5 | 0.776 | 0.796 | 0.698 | 0.821 | 0.730 | 0.740 | 0.723 | 0.837 | 0.744 | 0.718 |
| claudesonnet3.7 | 0.794 | 0.805 | 0.710 | 0.830 | 0.741 | 0.746 | 0.736 | 0.847 | 0.751 | 0.732 |
| GPT-4.1-mini | 0.777 | 0.787 | 0.694 | 0.816 | 0.726 | 0.736 | 0.719 | 0.829 | 0.736 | 0.750 |
| o3-mini | 0.775 | 0.788 | 0.690 | 0.814 | 0.728 | 0.733 | 0.721 | 0.827 | 0.739 | 0.744 |
| GPT-4.1-2025-04-14 | 0.773 | 0.786 | 0.685 | 0.813 | 0.723 | 0.732 | 0.717 | 0.825 | 0.734 | 0.748 |
| o3 | 0.758 | 0.784 | 0.688 | 0.809 | 0.725 | 0.716 | 0.719 | 0.823 | 0.737 | 0.726 |
| o4-mini | 0.761 | 0.783 | 0.684 | 0.809 | 0.720 | 0.729 | 0.716 | 0.821 | 0.730 | 0.719 |
| GPT-4o-mini | 0.739 | 0.782 | 0.670 | 0.804 | 0.710 | 0.712 | 0.702 | 0.805 | 0.714 | 0.702 |
| GPT-4o | 0.737 | 0.780 | 0.671 | 0.802 | 0.711 | 0.711 | 0.702 | 0.805 | 0.714 | 0.702 |

**(f) Long-Context Scores by Programming Language**

| Model | Python | C++ | Java | C | C# | JavaScript | TypeScript | Go | Rust | PHP |
|---|---|---|---|---|---|---|---|---|---|---|
| Gemini-2.5-Pro | 0.527 | 0.539 | 0.513 | 0.551 | 0.532 | 0.479 | 0.490 | 0.572 | 0.504 | 0.522 |
| Gemini-2.5-Flash | 0.530 | 0.542 | 0.516 | 0.554 | 0.535 | 0.482 | 0.493 | 0.565 | 0.507 | 0.517 |
| gpt5mini | 0.527 | 0.538 | 0.513 | 0.550 | 0.531 | 0.481 | 0.487 | 0.569 | 0.510 | 0.513 |
| claudesonnet4 | 0.510 | 0.540 | 0.515 | 0.552 | 0.533 | 0.483 | 0.485 | 0.571 | 0.506 | 0.520 |
| GPT-5 | 0.508 | 0.535 | 0.500 | 0.547 | 0.519 | 0.475 | 0.477 | 0.564 | 0.502 | 0.506 |
| claudesonnet3.7 | 0.509 | 0.541 | 0.514 | 0.553 | 0.532 | 0.480 | 0.489 | 0.570 | 0.505 | 0.521 |
| GPT-4.1-mini | 0.492 | 0.524 | 0.497 | 0.536 | 0.516 | 0.464 | 0.472 | 0.553 | 0.489 | 0.505 |
| o3-mini | 0.490 | 0.525 | 0.493 | 0.534 | 0.518 | 0.461 | 0.474 | 0.551 | 0.492 | 0.503 |
| GPT-4.1-2025-04-14 | 0.488 | 0.523 | 0.488 | 0.533 | 0.513 | 0.460 | 0.470 | 0.549 | 0.487 | 0.501 |
| o3 | 0.473 | 0.508 | 0.476 | 0.518 | 0.497 | 0.444 | 0.454 | 0.533 | 0.471 | 0.485 |
| o4-mini | 0.475 | 0.510 | 0.472 | 0.520 | 0.494 | 0.447 | 0.452 | 0.535 | 0.468 | 0.479 |
| GPT-4o-mini | 0.459 | 0.494 | 0.456 | 0.504 | 0.478 | 0.431 | 0.436 | 0.519 | 0.452 | 0.463 |
| GPT-4o | 0.457 | 0.492 | 0.457 | 0.502 | 0.479 | 0.430 | 0.436 | 0.519 | 0.452 | 0.463 |

Table 11: Detailed comparison of model performance by task category.

**(a) Total Scores by Task Category**

| Model | Arch. Understanding | Cross-File Refact. | Multi-Session Dev. | Bug Investigation | Feature Impl. | Code Comprehension | Integration Testing | Security Analysis |
|---|---|---|---|---|---|---|---|---|
| Gemini-2.5-Pro | 2.338 | 2.272 | 2.280 | 2.272 | 2.299 | 2.273 | 2.421 | 2.343 |
| Gemini-2.5-Flash | 2.280 | 2.303 | 2.291 | 2.276 | 2.335 | 2.211 | 2.430 | 2.325 |
| gpt5mini | 2.370 | 2.237 | 2.268 | 2.227 | 2.285 | 2.218 | 2.390 | 2.351 |
| claudesonnet4 | 2.346 | 2.203 | 2.256 | 2.262 | 2.227 | 2.298 | 2.402 | 2.307 |
| GPT-5 | 2.376 | 2.241 | 2.257 | 2.196 | 2.297 | 2.199 | 2.386 | 2.336 |
| claudesonnet3.7 | 2.332 | 2.269 | 2.273 | 2.206 | 2.289 | 2.228 | 2.403 | 2.322 |
| GPT-4.1-mini | 2.238 | 2.181 | 2.152 | 2.192 | 2.220 | 2.206 | 2.363 | 2.226 |
| o3-mini | 2.231 | 2.156 | 2.182 | 2.165 | 2.205 | 2.237 | 2.330 | 2.214 |
| GPT-4.1-2025-04-14 | 2.195 | 2.169 | 2.148 | 2.154 | 2.207 | 2.166 | 2.346 | 2.191 |
| o3 | 2.123 | 2.151 | 2.181 | 2.058 | 2.192 | 2.010 | 2.323 | 2.197 |
| o4-mini | 2.148 | 2.146 | 2.134 | 2.094 | 2.166 | 2.088 | 2.277 | 2.131 |
| GPT-4o-mini | 2.059 | 2.054 | 2.089 | 2.043 | 2.057 | 2.054 | 2.184 | 2.061 |
| GPT-4o | 2.054 | 2.055 | 2.068 | 2.038 | 2.063 | 2.051 | 2.200 | 2.056 |

**(b) Success Rates by Task Category (%)**

| Model | Arch. Understanding | Cross-File Refact. | Multi-Session Dev. | Bug Investigation | Feature Impl. | Code Comprehension | Integration Testing | Security Analysis |
|---|---|---|---|---|---|---|---|---|
| Gemini-2.5-Pro | 99.70 | 99.90 | 100.00 | 99.80 | 99.80 | 100.00 | 99.90 | 99.90 |
| Gemini-2.5-Flash | 100.00 | 100.00 | 100.00 | 99.90 | 100.00 | 100.00 | 100.00 | 100.00 |
| gpt5mini | 100.00 | 100.00 | 100.00 | 100.00 | 100.00 | 100.00 | 100.00 | 100.00 |
| claudesonnet4 | 99.60 | 99.40 | 99.50 | 99.50 | 99.70 | 99.70 | 99.60 | 99.60 |
| GPT-5 | 100.00 | 100.00 | 100.00 | 100.00 | 100.00 | 100.00 | 100.00 | 100.00 |
| claudesonnet3.7 | 99.60 | 99.90 | 99.90 | 99.70 | 99.80 | 99.90 | 99.90 | 99.70 |
| GPT-4.1-mini | 100.00 | 100.00 | 100.00 | 100.00 | 100.00 | 100.00 | 100.00 | 100.00 |
| o3-mini | 100.00 | 100.00 | 100.00 | 100.00 | 100.00 | 100.00 | 100.00 | 100.00 |
| GPT-4.1-2025-04-14 | 100.00 | 100.00 | 100.00 | 100.00 | 100.00 | 100.00 | 100.00 | 100.00 |
| o3 | 100.00 | 100.00 | 100.00 | 100.00 | 100.00 | 100.00 | 100.00 | 100.00 |
| o4-mini | 99.40 | 99.70 | 99.80 | 99.70 | 99.90 | 99.90 | 99.70 | 99.50 |
| GPT-4o-mini | 100.00 | 100.00 | 100.00 | 100.00 | 100.00 | 100.00 | 100.00 | 100.00 |
| GPT-4o | 100.00 | 100.00 | 100.00 | 100.00 | 100.00 | 100.00 | 100.00 | 100.00 |

**(c) Software Engineering Scores by Task Category**

| Model | Arch. Understanding | Cross-File Refact. | Multi-Session Dev. | Bug Investigation | Feature Impl. | Code Comprehension | Integration Testing | Security Analysis |
|---|---|---|---|---|---|---|---|---|
| Gemini-2.5-Pro | 0.375 | 0.375 | 0.375 | 0.375 | 0.375 | 0.375 | 0.375 | 0.375 |
| Gemini-2.5-Flash | 0.373 | 0.373 | 0.373 | 0.373 | 0.373 | 0.373 | 0.373 | 0.373 |
| gpt5mini | 0.376 | 0.376 | 0.376 | 0.376 | 0.376 | 0.376 | 0.376 | 0.376 |
| claudesonnet4 | 0.379 | 0.379 | 0.379 | 0.379 | 0.379 | 0.379 | 0.379 | 0.379 |
| GPT-5 | 0.367 | 0.367 | 0.367 | 0.367 | 0.367 | 0.367 | 0.367 | 0.367 |
| claudesonnet3.7 | 0.377 | 0.377 | 0.377 | 0.377 | 0.377 | 0.377 | 0.377 | 0.377 |
| GPT-4.1-mini | 0.359 | 0.359 | 0.359 | 0.359 | 0.359 | 0.359 | 0.359 | 0.359 |
| o3-mini | 0.355 | 0.355 | 0.355 | 0.355 | 0.355 | 0.355 | 0.355 | 0.355 |
| GPT-4.1-2025-04-14 | 0.352 | 0.352 | 0.352 | 0.352 | 0.352 | 0.352 | 0.352 | 0.352 |
| o3 | 0.342 | 0.342 | 0.342 | 0.342 | 0.342 | 0.342 | 0.342 | 0.342 |
| o4-mini | 0.353 | 0.353 | 0.353 | 0.353 | 0.353 | 0.353 | 0.353 | 0.353 |
| GPT-4o-mini | 0.341 | 0.341 | 0.341 | 0.341 | 0.341 | 0.341 | 0.341 | 0.341 |
| GPT-4o | 0.339 | 0.339 | 0.339 | 0.339 | 0.339 | 0.339 | 0.339 | 0.339 |

**(d) Functional Scores by Task Category**

| Model | Arch. Understanding | Cross-File Refact. | Multi-Session Dev. | Bug Investigation | Feature Impl. | Code Comprehension | Integration Testing | Security Analysis |
|---|---|---|---|---|---|---|---|---|
| Gemini-2.5-Pro | 0.356 | 0.356 | 0.356 | 0.356 | 0.356 | 0.356 | 0.356 | 0.356 |
| Gemini-2.5-Flash | 0.358 | 0.358 | 0.358 | 0.358 | 0.358 | 0.358 | 0.358 | 0.358 |
| gpt5mini | 0.371 | 0.371 | 0.371 | 0.371 | 0.371 | 0.371 | 0.371 | 0.371 |
| claudesonnet4 | 0.348 | 0.348 | 0.348 | 0.348 | 0.348 | 0.348 | 0.348 | 0.348 |
| GPT-5 | 0.383 | 0.383 | 0.383 | 0.383 | 0.383 | 0.383 | 0.383 | 0.383 |
| claudesonnet3.7 | 0.347 | 0.347 | 0.347 | 0.347 | 0.347 | 0.347 | 0.347 | 0.347 |
| GPT-4.1-mini | 0.365 | 0.365 | 0.365 | 0.365 | 0.365 | 0.365 | 0.365 | 0.365 |
| o3-mini | 0.368 | 0.368 | 0.368 | 0.368 | 0.368 | 0.368 | 0.368 | 0.368 |
| GPT-4.1-2025-04-14 | 0.364 | 0.364 | 0.364 | 0.364 | 0.364 | 0.364 | 0.364 | 0.364 |
| o3 | 0.385 | 0.385 | 0.385 | 0.385 | 0.385 | 0.385 | 0.385 | 0.385 |
| o4-mini | 0.360 | 0.360 | 0.360 | 0.360 | 0.360 | 0.360 | 0.360 | 0.360 |
| GPT-4o-mini | 0.360 | 0.360 | 0.360 | 0.360 | 0.360 | 0.360 | 0.360 | 0.360 |
| GPT-4o | 0.362 | 0.362 | 0.362 | 0.362 | 0.362 | 0.362 | 0.362 | 0.362 |

**(e) Quality Scores by Task Category**

| Model | Arch. Understanding | Cross-File Refact. | Multi-Session Dev. | Bug Investigation | Feature Impl. | Code Comprehension | Integration Testing | Security Analysis |
|---|---|---|---|---|---|---|---|---|
| Gemini-2.5-Pro | 0.768 | 0.768 | 0.768 | 0.768 | 0.768 | 0.768 | 0.768 | 0.768 |
| Gemini-2.5-Flash | 0.741 | 0.741 | 0.741 | 0.741 | 0.741 | 0.741 | 0.741 | 0.741 |
| gpt5mini | 0.745 | 0.745 | 0.745 | 0.745 | 0.745 | 0.745 | 0.745 | 0.745 |
| claudesonnet4 | 0.762 | 0.762 | 0.762 | 0.762 | 0.762 | 0.762 | 0.762 | 0.762 |
| GPT-5 | 0.732 | 0.732 | 0.732 | 0.732 | 0.732 | 0.732 | 0.732 | 0.732 |
| claudesonnet3.7 | 0.773 | 0.773 | 0.773 | 0.773 | 0.773 | 0.773 | 0.773 | 0.773 |
| GPT-4.1-mini | 0.739 | 0.739 | 0.739 | 0.739 | 0.739 | 0.739 | 0.739 | 0.739 |
| o3-mini | 0.726 | 0.726 | 0.726 | 0.726 | 0.726 | 0.726 | 0.726 | 0.726 |
| GPT-4.1-2025-04-14 | 0.720 | 0.720 | 0.720 | 0.720 | 0.720 | 0.720 | 0.720 | 0.720 |
| o3 | 0.721 | 0.721 | 0.721 | 0.721 | 0.721 | 0.721 | 0.721 | 0.721 |
| o4-mini | 0.704 | 0.704 | 0.704 | 0.704 | 0.704 | 0.704 | 0.704 | 0.704 |
| GPT-4o-mini | 0.679 | 0.679 | 0.679 | 0.679 | 0.679 | 0.679 | 0.679 | 0.679 |
| GPT-4o | 0.677 | 0.677 | 0.677 | 0.677 | 0.677 | 0.677 | 0.677 | 0.677 |

**(f) Long-Context Scores by Task Category**

| Model | Arch. Understanding | Cross-File Refact. | Multi-Session Dev. | Bug Investigation | Feature Impl. | Code Comprehension | Integration Testing | Security Analysis |
|---|---|---|---|---|---|---|---|---|
| Gemini-2.5-Pro | 0.523 | 0.523 | 0.523 | 0.523 | 0.523 | 0.523 | 0.523 | 0.523 |
| Gemini-2.5-Flash | 0.565 | 0.565 | 0.565 | 0.565 | 0.565 | 0.565 | 0.565 | 0.565 |
| gpt5mini | 0.479 | 0.479 | 0.479 | 0.479 | 0.479 | 0.479 | 0.479 | 0.479 |
| claudesonnet4 | 0.492 | 0.492 | 0.492 | 0.492 | 0.492 | 0.492 | 0.492 | 0.492 |
| GPT-5 | 0.492 | 0.492 | 0.492 | 0.492 | 0.492 | 0.492 | 0.492 | 0.492 |
| claudesonnet3.7 | 0.477 | 0.477 | 0.477 | 0.477 | 0.477 | 0.477 | 0.477 | 0.477 |
| GPT-4.1-mini | 0.435 | 0.435 | 0.435 | 0.435 | 0.435 | 0.435 | 0.435 | 0.435 |
| o3-mini | 0.455 | 0.455 | 0.455 | 0.455 | 0.455 | 0.455 | 0.455 | 0.455 |
| GPT-4.1-2025-04-14 | 0.451 | 0.451 | 0.451 | 0.451 | 0.451 | 0.451 | 0.451 | 0.451 |
| o3 | 0.342 | 0.342 | 0.342 | 0.342 | 0.342 | 0.342 | 0.342 | 0.342 |
| o4-mini | 0.393 | 0.393 | 0.393 | 0.393 | 0.393 | 0.393 | 0.393 | 0.393 |
| GPT-4o-mini | 0.344 | 0.344 | 0.344 | 0.344 | 0.344 | 0.344 | 0.344 | 0.344 |
| GPT-4o | 0.348 | 0.348 | 0.348 | 0.348 | 0.348 | 0.348 | 0.348 | 0.348 |

Table 12: Detailed comparison of model performance by application domain.

### (a) Total Scores by Application Domain

| Model | Web Apps | API Services | Data Systems | ML/AI Systems | Gaming Sim. | Blockchain | Desktop Apps | Embedded Sys. | Mobile Apps | Network Tools |
|---|---|---|---|---|---|---|---|---|---|---|
| Gemini-2.5-Pro | 2.319 | 2.313 | 2.302 | 2.307 | 2.271 | 2.347 | 2.324 | 2.302 | 2.304 | 2.329 |
| Gemini-2.5-Flash | 2.302 | 2.312 | 2.304 | 2.310 | 2.252 | 2.323 | 2.309 | 2.301 | 2.310 | 2.321 |
| gpt5mini | 2.282 | 2.271 | 2.291 | 2.296 | 2.278 | 2.345 | 2.292 | 2.295 | 2.297 | 2.286 |
| claudesonnet4 | 2.283 | 2.273 | 2.287 | 2.314 | 2.244 | 2.262 | 2.296 | 2.301 | 2.303 | 2.317 |
| GPT-5 | 2.258 | 2.235 | 2.262 | 2.269 | 2.198 | 2.287 | 2.315 | 2.286 | 2.296 | 2.322 |
| claudesonnet3.7 | 2.289 | 2.291 | 2.278 | 2.285 | 2.261 | 2.295 | 2.283 | 2.273 | 2.290 | 2.306 |
| GPT-4.1-mini | 2.215 | 2.211 | 2.231 | 2.228 | 2.192 | 2.232 | 2.229 | 2.217 | 2.228 | 2.232 |
| o3-mini | 2.203 | 2.203 | 2.223 | 2.218 | 2.184 | 2.223 | 2.226 | 2.211 | 2.219 | 2.220 |
| GPT-4.1-2025-04-14 | 2.194 | 2.185 | 2.207 | 2.201 | 2.164 | 2.212 | 2.207 | 2.194 | 2.206 | 2.201 |
| o3 | 2.138 | 2.117 | 2.167 | 2.169 | 2.110 | 2.172 | 2.169 | 2.152 | 2.166 | 2.171 |
| o4-mini | 2.137 | 2.132 | 2.157 | 2.153 | 2.116 | 2.167 | 2.157 | 2.143 | 2.157 | 2.159 |
| GPT-4o-mini | 2.065 | 2.061 | 2.084 | 2.080 | 2.049 | 2.086 | 2.082 | 2.071 | 2.081 | 2.084 |
| GPT-4o | 2.064 | 2.057 | 2.082 | 2.081 | 2.051 | 2.084 | 2.081 | 2.070 | 2.081 | 2.082 |

### (b) Success Rates by Application Domain (%)

| Model | Web Apps | API Services | Data Systems | ML/AI Systems | Gaming Sim. | Blockchain | Desktop Apps | Embedded Sys. | Mobile Apps | Network Tools |
|---|---|---|---|---|---|---|---|---|---|---|
| Gemini-2.5-Pro | 99.88 | 99.88 | 99.88 | 99.88 | 99.88 | 99.88 | 99.88 | 99.88 | 99.88 | 99.88 |
| Gemini-2.5-Flash | 100.00 | 100.00 | 99.88 | 100.00 | 99.88 | 100.00 | 100.00 | 100.00 | 100.00 | 100.00 |
| gpt5mini | 100.00 | 100.00 | 100.00 | 100.00 | 100.00 | 100.00 | 100.00 | 100.00 | 100.00 | 100.00 |
| claudesonnet4 | 99.50 | 99.50 | 99.62 | 99.62 | 99.50 | 99.50 | 99.62 | 99.50 | 99.62 | 99.62 |
| GPT-5 | 100.00 | 100.00 | 100.00 | 100.00 | 100.00 | 100.00 | 100.00 | 100.00 | 100.00 | 100.00 |
| claudesonnet3.7 | 99.75 | 99.88 | 99.75 | 99.75 | 99.75 | 99.88 | 99.88 | 99.75 | 99.88 | 99.75 |
| GPT-4.1-mini | 100.00 | 100.00 | 100.00 | 100.00 | 100.00 | 100.00 | 100.00 | 100.00 | 100.00 | 100.00 |
| o3-mini | 100.00 | 100.00 | 100.00 | 100.00 | 100.00 | 100.00 | 100.00 | 100.00 | 100.00 | 100.00 |
| GPT-4.1-2025-04-14 | 100.00 | 100.00 | 100.00 | 100.00 | 100.00 | 100.00 | 100.00 | 100.00 | 100.00 | 100.00 |
| o3 | 100.00 | 100.00 | 100.00 | 100.00 | 100.00 | 100.00 | 100.00 | 100.00 | 100.00 | 100.00 |
| o4-mini | 99.62 | 99.75 | 99.75 | 99.75 | 99.62 | 99.75 | 99.75 | 99.62 | 99.75 | 99.75 |
| GPT-4o-mini | 100.00 | 100.00 | 100.00 | 100.00 | 100.00 | 100.00 | 100.00 | 100.00 | 100.00 | 100.00 |
| GPT-4o | 100.00 | 100.00 | 100.00 | 100.00 | 100.00 | 100.00 | 100.00 | 100.00 | 100.00 | 100.00 |

### (c) Software Engineering Scores by Application Domain

| Model | Web Apps | API Services | Data Systems | ML/AI Systems | Gaming Sim. | Blockchain | Desktop Apps | Embedded Sys. | Mobile Apps | Network Tools |
|---|---|---|---|---|---|---|---|---|---|---|
| Gemini-2.5-Pro | 0.374 | 0.375 | 0.375 | 0.375 | 0.374 | 0.376 | 0.375 | 0.375 | 0.375 | 0.376 |
| Gemini-2.5-Flash | 0.372 | 0.374 | 0.373 | 0.374 | 0.372 | 0.373 | 0.374 | 0.373 | 0.374 | 0.374 |
| gpt5mini | 0.375 | 0.375 | 0.376 | 0.376 | 0.376 | 0.377 | 0.376 | 0.376 | 0.376 | 0.375 |
| claudesonnet4 | 0.378 | 0.378 | 0.379 | 0.380 | 0.377 | 0.377 | 0.379 | 0.379 | 0.379 | 0.380 |
| GPT-5 | 0.366 | 0.365 | 0.367 | 0.367 | 0.365 | 0.368 | 0.368 | 0.367 | 0.368 | 0.369 |
| claudesonnet3.7 | 0.377 | 0.377 | 0.377 | 0.377 | 0.376 | 0.378 | 0.377 | 0.377 | 0.377 | 0.378 |
| GPT-4.1-mini | 0.358 | 0.358 | 0.360 | 0.359 | 0.357 | 0.359 | 0.359 | 0.358 | 0.359 | 0.359 |
| o3-mini | 0.354 | 0.354 | 0.356 | 0.355 | 0.353 | 0.355 | 0.356 | 0.355 | 0.355 | 0.355 |
| GPT-4.1-2025-04-14 | 0.351 | 0.350 | 0.353 | 0.352 | 0.350 | 0.353 | 0.353 | 0.351 | 0.353 | 0.352 |
| o3 | 0.341 | 0.339 | 0.344 | 0.344 | 0.338 | 0.344 | 0.344 | 0.342 | 0.344 | 0.344 |
| o4-mini | 0.352 | 0.351 | 0.354 | 0.353 | 0.350 | 0.354 | 0.354 | 0.352 | 0.354 | 0.354 |
| GPT-4o-mini | 0.340 | 0.339 | 0.342 | 0.341 | 0.338 | 0.342 | 0.342 | 0.340 | 0.342 | 0.342 |
| GPT-4o | 0.338 | 0.337 | 0.340 | 0.340 | 0.337 | 0.340 | 0.340 | 0.339 | 0.340 | 0.340 |

### (d) Functional Scores by Application Domain

| Model | Web Apps | API Services | Data Systems | ML/AI Systems | Gaming Sim. | Blockchain | Desktop Apps | Embedded Sys. | Mobile Apps | Network Tools |
|---|---|---|---|---|---|---|---|---|---|---|
| Gemini-2.5-Pro | 0.356 | 0.356 | 0.356 | 0.356 | 0.355 | 0.357 | 0.356 | 0.356 | 0.356 | 0.357 |
| Gemini-2.5-Flash | 0.358 | 0.358 | 0.358 | 0.358 | 0.357 | 0.358 | 0.358 | 0.358 | 0.358 | 0.358 |
| gpt5mini | 0.370 | 0.370 | 0.371 | 0.371 | 0.371 | 0.372 | 0.371 | 0.371 | 0.371 | 0.370 |
| claudesonnet4 | 0.347 | 0.347 | 0.348 | 0.349 | 0.346 | 0.346 | 0.348 | 0.348 | 0.348 | 0.349 |
| GPT-5 | 0.382 | 0.381 | 0.383 | 0.383 | 0.381 | 0.384 | 0.384 | 0.383 | 0.384 | 0.385 |
| claudesonnet3.7 | 0.347 | 0.347 | 0.347 | 0.347 | 0.346 | 0.348 | 0.347 | 0.347 | 0.347 | 0.348 |
| GPT-4.1-mini | 0.364 | 0.364 | 0.366 | 0.365 | 0.363 | 0.365 | 0.365 | 0.364 | 0.365 | 0.365 |
| o3-mini | 0.367 | 0.367 | 0.369 | 0.368 | 0.366 | 0.368 | 0.369 | 0.368 | 0.368 | 0.368 |
| GPT-4.1-2025-04-14 | 0.363 | 0.362 | 0.365 | 0.364 | 0.362 | 0.365 | 0.365 | 0.363 | 0.365 | 0.364 |
| o3 | 0.384 | 0.382 | 0.387 | 0.387 | 0.381 | 0.387 | 0.387 | 0.385 | 0.387 | 0.387 |
| o4-mini | 0.359 | 0.358 | 0.361 | 0.360 | 0.357 | 0.361 | 0.361 | 0.359 | 0.361 | 0.361 |
| GPT-4o-mini | 0.359 | 0.358 | 0.361 | 0.360 | 0.357 | 0.361 | 0.361 | 0.359 | 0.361 | 0.361 |
| GPT-4o | 0.361 | 0.360 | 0.363 | 0.362 | 0.360 | 0.363 | 0.363 | 0.361 | 0.363 | 0.363 |

Table 13: Detailed comparison of model performance by application domain (continued).

**(e) Quality Scores by Application Domain**

| Model | Web Apps | API Services | Data Systems | ML/AI Systems | Gaming Sim. | Blockchain | Desktop Apps | Embedded Sys. | Mobile Apps | Network Tools |
|---|---|---|---|---|---|---|---|---|---|---|
| Gemini-2.5-Pro | 0.768 | 0.768 | 0.768 | 0.768 | 0.767 | 0.769 | 0.768 | 0.768 | 0.768 | 0.769 |
| Gemini-2.5-Flash | 0.741 | 0.741 | 0.741 | 0.741 | 0.740 | 0.741 | 0.741 | 0.741 | 0.741 | 0.741 |
| gpt5mini | 0.745 | 0.744 | 0.745 | 0.745 | 0.745 | 0.746 | 0.745 | 0.745 | 0.745 | 0.744 |
| claudesonnet4 | 0.762 | 0.762 | 0.762 | 0.763 | 0.761 | 0.761 | 0.762 | 0.762 | 0.762 | 0.763 |
| GPT-5 | 0.732 | 0.731 | 0.732 | 0.732 | 0.730 | 0.733 | 0.733 | 0.732 | 0.733 | 0.734 |
| claudesonnet3.7 | 0.773 | 0.773 | 0.772 | 0.773 | 0.772 | 0.774 | 0.773 | 0.772 | 0.773 | 0.774 |
| GPT-4.1-mini | 0.739 | 0.738 | 0.740 | 0.739 | 0.738 | 0.740 | 0.740 | 0.739 | 0.740 | 0.740 |
| o3-mini | 0.726 | 0.725 | 0.727 | 0.726 | 0.724 | 0.727 | 0.727 | 0.726 | 0.727 | 0.727 |
| GPT-4.1-2025-04-14 | 0.720 | 0.719 | 0.721 | 0.720 | 0.718 | 0.721 | 0.721 | 0.720 | 0.721 | 0.721 |
| o3 | 0.721 | 0.720 | 0.723 | 0.723 | 0.719 | 0.723 | 0.723 | 0.722 | 0.723 | 0.723 |
| o4-mini | 0.704 | 0.703 | 0.706 | 0.705 | 0.702 | 0.706 | 0.706 | 0.704 | 0.706 | 0.706 |
| GPT-4o-mini | 0.679 | 0.678 | 0.681 | 0.680 | 0.677 | 0.681 | 0.681 | 0.679 | 0.681 | 0.681 |
| GPT-4o | 0.677 | 0.676 | 0.679 | 0.678 | 0.676 | 0.679 | 0.679 | 0.677 | 0.679 | 0.679 |

**(f) Long-Context Scores by Application Domain**

| Model | Web Apps | API Services | Data Systems | ML/AI Systems | Gaming Sim. | Blockchain | Desktop Apps | Embedded Sys. | Mobile Apps | Network Tools |
|---|---|---|---|---|---|---|---|---|---|---|
| Gemini-2.5-Pro | 0.523 | 0.523 | 0.523 | 0.523 | 0.522 | 0.524 | 0.523 | 0.523 | 0.523 | 0.524 |
| Gemini-2.5-Flash | 0.565 | 0.565 | 0.565 | 0.565 | 0.564 | 0.565 | 0.565 | 0.565 | 0.565 | 0.565 |
| gpt5mini | 0.479 | 0.478 | 0.479 | 0.479 | 0.479 | 0.480 | 0.479 | 0.479 | 0.479 | 0.478 |
| claudesonnet4 | 0.492 | 0.492 | 0.492 | 0.493 | 0.491 | 0.491 | 0.492 | 0.492 | 0.492 | 0.493 |
| GPT-5 | 0.492 | 0.491 | 0.492 | 0.492 | 0.490 | 0.493 | 0.493 | 0.492 | 0.493 | 0.494 |
| claudesonnet3.7 | 0.477 | 0.477 | 0.476 | 0.477 | 0.476 | 0.478 | 0.477 | 0.476 | 0.477 | 0.478 |
| GPT-4.1-mini | 0.435 | 0.434 | 0.436 | 0.435 | 0.434 | 0.436 | 0.436 | 0.435 | 0.436 | 0.436 |
| o3-mini | 0.455 | 0.454 | 0.456 | 0.455 | 0.453 | 0.456 | 0.456 | 0.455 | 0.456 | 0.456 |
| GPT-4.1-2025-04-14 | 0.451 | 0.450 | 0.452 | 0.451 | 0.449 | 0.452 | 0.452 | 0.451 | 0.452 | 0.452 |
| o3 | 0.342 | 0.341 | 0.344 | 0.344 | 0.340 | 0.344 | 0.344 | 0.343 | 0.344 | 0.344 |
| o4-mini | 0.393 | 0.392 | 0.395 | 0.394 | 0.391 | 0.395 | 0.395 | 0.393 | 0.395 | 0.395 |
| GPT-4o-mini | 0.344 | 0.343 | 0.346 | 0.345 | 0.342 | 0.346 | 0.346 | 0.344 | 0.346 | 0.346 |
| GPT-4o | 0.348 | 0.347 | 0.350 | 0.349 | 0.347 | 0.350 | 0.350 | 0.348 | 0.350 | 0.350 |

Table 14: Detailed comparison of model performance by architecture pattern.

### (a) Total Scores by Architecture Pattern

| Model | Monolithic | Microservices | MVC | Hexagonal | Event-Driven | Serverless | Layered | Component | Repository | Factory |
|---|---|---|---|---|---|---|---|---|---|---|
| Gemini-2.5-Pro | 2.309 | 2.305 | 2.330 | 2.350 | 2.314 | 2.329 | 2.316 | 2.295 | 2.307 | 2.315 |
| Gemini-2.5-Flash | 2.303 | 2.306 | 2.325 | 2.383 | 2.317 | 2.325 | 2.298 | 2.286 | 2.303 | 2.307 |
| gpt5mini | 2.295 | 2.271 | 2.321 | 2.458 | 2.300 | 2.308 | 2.282 | 2.275 | 2.289 | 2.291 |
| claudesonnet4 | 2.288 | 2.267 | 2.321 | 2.368 | 2.300 | 2.309 | 2.285 | 2.271 | 2.284 | 2.288 |
| GPT-5 | 2.262 | 2.225 | 2.284 | 2.347 | 2.271 | 2.280 | 2.297 | 2.283 | 2.291 | 2.295 |
| claudesonnet3.7 | 2.281 | 2.267 | 2.314 | 2.357 | 2.295 | 2.302 | 2.281 | 2.268 | 2.281 | 2.285 |
| GPT-4.1-mini | 2.219 | 2.208 | 2.242 | 2.284 | 2.228 | 2.235 | 2.221 | 2.209 | 2.218 | 2.221 |
| o3-mini | 2.213 | 2.196 | 2.236 | 2.275 | 2.221 | 2.228 | 2.210 | 2.198 | 2.212 | 2.215 |
| GPT-4.1-2025-04-14 | 2.194 | 2.180 | 2.216 | 2.250 | 2.202 | 2.210 | 2.194 | 2.182 | 2.194 | 2.197 |
| o3 | 2.147 | 2.123 | 2.174 | 2.207 | 2.158 | 2.166 | 2.150 | 2.139 | 2.151 | 2.154 |
| o4-mini | 2.146 | 2.132 | 2.168 | 2.199 | 2.153 | 2.161 | 2.146 | 2.135 | 2.147 | 2.150 |
| GPT-4o-mini | 2.072 | 2.058 | 2.093 | 2.118 | 2.079 | 2.087 | 2.072 | 2.062 | 2.074 | 2.076 |
| GPT-4o | 2.070 | 2.056 | 2.091 | 2.115 | 2.077 | 2.085 | 2.070 | 2.061 | 2.072 | 2.074 |

### (b) Success Rates by Architecture Pattern (%)

| Model | Monolithic | Microservices | MVC | Hexagonal | Event-Driven | Serverless | Layered | Component | Repository | Factory |
|---|---|---|---|---|---|---|---|---|---|---|
| Gemini-2.5-Pro | 99.88 | 99.88 | 99.88 | 99.88 | 99.88 | 99.88 | 99.88 | 99.88 | 99.88 | 99.88 |
| Gemini-2.5-Flash | 100.00 | 100.00 | 100.00 | 100.00 | 100.00 | 100.00 | 100.00 | 100.00 | 100.00 | 99.88 |
| gpt5mini | 100.00 | 100.00 | 100.00 | 100.00 | 100.00 | 100.00 | 100.00 | 100.00 | 100.00 | 100.00 |
| claudesonnet4 | 99.50 | 99.50 | 99.62 | 99.62 | 99.62 | 99.62 | 99.50 | 99.50 | 99.50 | 99.50 |
| GPT-5 | 100.00 | 100.00 | 100.00 | 100.00 | 100.00 | 100.00 | 100.00 | 100.00 | 100.00 | 100.00 |
| claudesonnet3.7 | 99.75 | 99.88 | 99.88 | 99.75 | 99.75 | 99.88 | 99.75 | 99.75 | 99.75 | 99.88 |
| GPT-4.1-mini | 100.00 | 100.00 | 100.00 | 100.00 | 100.00 | 100.00 | 100.00 | 100.00 | 100.00 | 100.00 |
| o3-mini | 100.00 | 100.00 | 100.00 | 100.00 | 100.00 | 100.00 | 100.00 | 100.00 | 100.00 | 100.00 |
| GPT-4.1-2025-04-14 | 100.00 | 100.00 | 100.00 | 100.00 | 100.00 | 100.00 | 100.00 | 100.00 | 100.00 | 100.00 |
| o3 | 100.00 | 100.00 | 100.00 | 100.00 | 100.00 | 100.00 | 100.00 | 100.00 | 100.00 | 100.00 |
| o4-mini | 99.75 | 99.75 | 99.75 | 99.75 | 99.75 | 99.75 | 99.75 | 99.75 | 99.75 | 99.75 |
| GPT-4o-mini | 100.00 | 100.00 | 100.00 | 100.00 | 100.00 | 100.00 | 100.00 | 100.00 | 100.00 | 100.00 |
| GPT-4o | 100.00 | 100.00 | 100.00 | 100.00 | 100.00 | 100.00 | 100.00 | 100.00 | 100.00 | 100.00 |

### (c) Software Engineering Scores by Architecture Pattern

| Model | Monolithic | Microservices | MVC | Hexagonal | Event-Driven | Serverless | Layered | Component | Repository | Factory |
|---|---|---|---|---|---|---|---|---|---|---|
| Gemini-2.5-Pro | 0.375 | 0.375 | 0.375 | 0.376 | 0.375 | 0.375 | 0.375 | 0.374 | 0.375 | 0.375 |
| Gemini-2.5-Flash | 0.373 | 0.373 | 0.373 | 0.375 | 0.373 | 0.373 | 0.373 | 0.372 | 0.373 | 0.373 |
| gpt5mini | 0.376 | 0.375 | 0.376 | 0.378 | 0.376 | 0.376 | 0.375 | 0.375 | 0.376 | 0.376 |
| claudesonnet4 | 0.379 | 0.378 | 0.379 | 0.381 | 0.379 | 0.379 | 0.378 | 0.378 | 0.378 | 0.379 |
| GPT-5 | 0.367 | 0.366 | 0.367 | 0.369 | 0.367 | 0.367 | 0.368 | 0.367 | 0.368 | 0.368 |
| claudesonnet3.7 | 0.377 | 0.377 | 0.378 | 0.379 | 0.378 | 0.378 | 0.377 | 0.377 | 0.377 | 0.377 |
| GPT-4.1-mini | 0.359 | 0.358 | 0.360 | 0.361 | 0.359 | 0.359 | 0.359 | 0.358 | 0.359 | 0.359 |
| o3-mini | 0.355 | 0.354 | 0.356 | 0.357 | 0.355 | 0.355 | 0.355 | 0.354 | 0.355 | 0.355 |
| GPT-4.1-2025-04-14 | 0.352 | 0.351 | 0.353 | 0.354 | 0.352 | 0.352 | 0.352 | 0.351 | 0.352 | 0.352 |
| o3 | 0.342 | 0.340 | 0.344 | 0.345 | 0.343 | 0.343 | 0.342 | 0.341 | 0.342 | 0.342 |
| o4-mini | 0.353 | 0.352 | 0.354 | 0.355 | 0.353 | 0.353 | 0.353 | 0.352 | 0.353 | 0.353 |
| GPT-4o-mini | 0.341 | 0.340 | 0.342 | 0.343 | 0.341 | 0.341 | 0.341 | 0.340 | 0.341 | 0.341 |
| GPT-4o | 0.339 | 0.338 | 0.340 | 0.341 | 0.339 | 0.339 | 0.339 | 0.338 | 0.339 | 0.339 |

### (d) Functional Scores by Architecture Pattern

| Model | Monolithic | Microservices | MVC | Hexagonal | Event-Driven | Serverless | Layered | Component | Repository | Factory |
|---|---|---|---|---|---|---|---|---|---|---|
| Gemini-2.5-Pro | 0.356 | 0.356 | 0.357 | 0.357 | 0.356 | 0.357 | 0.356 | 0.355 | 0.356 | 0.356 |
| Gemini-2.5-Flash | 0.358 | 0.358 | 0.358 | 0.360 | 0.358 | 0.358 | 0.357 | 0.357 | 0.358 | 0.358 |
| gpt5mini | 0.371 | 0.370 | 0.371 | 0.374 | 0.371 | 0.371 | 0.370 | 0.370 | 0.371 | 0.371 |
| claudesonnet4 | 0.348 | 0.347 | 0.348 | 0.350 | 0.348 | 0.348 | 0.347 | 0.347 | 0.347 | 0.348 |
| GPT-5 | 0.383 | 0.382 | 0.383 | 0.385 | 0.383 | 0.383 | 0.384 | 0.383 | 0.384 | 0.384 |
| claudesonnet3.7 | 0.347 | 0.347 | 0.348 | 0.349 | 0.348 | 0.348 | 0.347 | 0.347 | 0.347 | 0.347 |
| GPT-4.1-mini | 0.365 | 0.364 | 0.366 | 0.367 | 0.365 | 0.365 | 0.365 | 0.364 | 0.365 | 0.365 |
| o3-mini | 0.368 | 0.367 | 0.369 | 0.370 | 0.368 | 0.368 | 0.368 | 0.367 | 0.368 | 0.368 |
| GPT-4.1-2025-04-14 | 0.364 | 0.363 | 0.365 | 0.366 | 0.364 | 0.364 | 0.364 | 0.363 | 0.364 | 0.364 |
| o3 | 0.385 | 0.383 | 0.387 | 0.388 | 0.386 | 0.386 | 0.385 | 0.384 | 0.385 | 0.385 |
| o4-mini | 0.360 | 0.359 | 0.361 | 0.362 | 0.360 | 0.360 | 0.360 | 0.359 | 0.360 | 0.360 |
| GPT-4o-mini | 0.360 | 0.359 | 0.361 | 0.362 | 0.360 | 0.360 | 0.360 | 0.359 | 0.360 | 0.360 |
| GPT-4o | 0.362 | 0.361 | 0.363 | 0.364 | 0.362 | 0.362 | 0.362 | 0.361 | 0.362 | 0.362 |

Table 15: Detailed comparison of model performance by architecture pattern (continued).

**(e) Quality Scores by Architecture Pattern**

| Model | Monolithic | Microservices | MVC | Hexagonal | Event-Driven | Serverless | Layered | Component | Repository | Factory |
|---|---|---|---|---|---|---|---|---|---|---|
| Gemini-2.5-Pro | 0.768 | 0.768 | 0.769 | 0.769 | 0.768 | 0.769 | 0.768 | 0.767 | 0.768 | 0.768 |
| Gemini-2.5-Flash | 0.741 | 0.741 | 0.741 | 0.743 | 0.741 | 0.741 | 0.740 | 0.740 | 0.741 | 0.741 |
| gpt5mini | 0.745 | 0.744 | 0.745 | 0.748 | 0.745 | 0.745 | 0.744 | 0.744 | 0.745 | 0.745 |
| claudesonnet4 | 0.762 | 0.761 | 0.762 | 0.764 | 0.762 | 0.762 | 0.761 | 0.761 | 0.761 | 0.762 |
| GPT-5 | 0.732 | 0.731 | 0.732 | 0.734 | 0.732 | 0.732 | 0.733 | 0.732 | 0.733 | 0.733 |
| claudesonnet3.7 | 0.773 | 0.773 | 0.774 | 0.775 | 0.774 | 0.774 | 0.773 | 0.772 | 0.773 | 0.773 |
| GPT-4.1-mini | 0.739 | 0.738 | 0.740 | 0.741 | 0.739 | 0.739 | 0.739 | 0.738 | 0.739 | 0.739 |
| o3-mini | 0.726 | 0.725 | 0.727 | 0.728 | 0.726 | 0.726 | 0.726 | 0.725 | 0.726 | 0.726 |
| GPT-4.1-2025-04-14 | 0.720 | 0.719 | 0.721 | 0.722 | 0.720 | 0.720 | 0.720 | 0.719 | 0.720 | 0.720 |
| o3 | 0.721 | 0.720 | 0.723 | 0.724 | 0.722 | 0.722 | 0.721 | 0.720 | 0.721 | 0.721 |
| o4-mini | 0.704 | 0.703 | 0.706 | 0.707 | 0.705 | 0.705 | 0.704 | 0.703 | 0.704 | 0.704 |
| GPT-4o-mini | 0.679 | 0.678 | 0.681 | 0.682 | 0.680 | 0.680 | 0.679 | 0.678 | 0.679 | 0.679 |
| GPT-4o | 0.677 | 0.676 | 0.679 | 0.680 | 0.678 | 0.678 | 0.677 | 0.676 | 0.677 | 0.677 |

**(f) Long-Context Scores by Architecture Pattern**

| Model | Monolithic | Microservices | MVC | Hexagonal | Event-Driven | Serverless | Layered | Component | Repository | Factory |
|---|---|---|---|---|---|---|---|---|---|---|
| Gemini-2.5-Pro | 0.523 | 0.523 | 0.524 | 0.524 | 0.523 | 0.524 | 0.523 | 0.522 | 0.523 | 0.523 |
| Gemini-2.5-Flash | 0.565 | 0.565 | 0.565 | 0.567 | 0.565 | 0.565 | 0.564 | 0.564 | 0.565 | 0.565 |
| gpt5mini | 0.479 | 0.478 | 0.479 | 0.482 | 0.479 | 0.479 | 0.478 | 0.478 | 0.479 | 0.479 |
| claudesonnet4 | 0.492 | 0.491 | 0.492 | 0.494 | 0.492 | 0.492 | 0.491 | 0.491 | 0.491 | 0.492 |
| GPT-5 | 0.492 | 0.491 | 0.492 | 0.494 | 0.492 | 0.492 | 0.493 | 0.492 | 0.493 | 0.493 |
| claudesonnet3.7 | 0.477 | 0.477 | 0.478 | 0.479 | 0.478 | 0.478 | 0.477 | 0.476 | 0.477 | 0.477 |
| GPT-4.1-mini | 0.435 | 0.434 | 0.436 | 0.437 | 0.435 | 0.435 | 0.435 | 0.434 | 0.435 | 0.435 |
| o3-mini | 0.455 | 0.454 | 0.456 | 0.457 | 0.455 | 0.455 | 0.455 | 0.454 | 0.455 | 0.455 |
| GPT-4.1-2025-04-14 | 0.451 | 0.450 | 0.452 | 0.453 | 0.451 | 0.451 | 0.451 | 0.450 | 0.451 | 0.451 |
| o3 | 0.342 | 0.341 | 0.344 | 0.345 | 0.343 | 0.343 | 0.342 | 0.341 | 0.342 | 0.342 |
| o4-mini | 0.393 | 0.392 | 0.395 | 0.396 | 0.394 | 0.394 | 0.393 | 0.392 | 0.393 | 0.393 |
| GPT-4o-mini | 0.344 | 0.343 | 0.346 | 0.347 | 0.345 | 0.345 | 0.344 | 0.343 | 0.344 | 0.344 |
| GPT-4o | 0.348 | 0.347 | 0.350 | 0.351 | 0.349 | 0.349 | 0.348 | 0.347 | 0.348 | 0.348 |

Table 16: Detailed comparison of model performance by theme.

**(a) Total Scores by Theme**

| Model | Algorithms | Data Structures | System Design | Security | Performance | Testing | Maintenance | Integration |
|---|---|---|---|---|---|---|---|---|
| Gemini-2.5-Pro | 2.325 | 2.318 | 2.301 | 2.343 | 2.295 | 2.421 | 2.306 | 2.315 |
| Gemini-2.5-Flash | 2.309 | 2.312 | 2.299 | 2.325 | 2.297 | 2.430 | 2.301 | 2.312 |
| gpt5mini | 2.295 | 2.298 | 2.284 | 2.351 | 2.282 | 2.390 | 2.289 | 2.297 |
| claudesonnet4 | 2.289 | 2.294 | 2.276 | 2.275 | 2.275 | 2.402 | 2.284 | 2.291 |
| GPT-5 | 2.269 | 2.273 | 2.257 | 2.336 | 2.260 | 2.386 | 2.265 | 2.274 |
| claudesonnet3.7 | 2.285 | 2.289 | 2.272 | 2.322 | 2.275 | 2.403 | 2.280 | 2.287 |
| GPT-4.1-mini | 2.223 | 2.228 | 2.212 | 2.226 | 2.215 | 2.363 | 2.219 | 2.226 |
| o3-mini | 2.217 | 2.222 | 2.206 | 2.214 | 2.209 | 2.330 | 2.213 | 2.220 |
| GPT-4.1-2025-04-14 | 2.197 | 2.202 | 2.186 | 2.191 | 2.189 | 2.346 | 2.193 | 2.200 |
| o3 | 2.155 | 2.160 | 2.144 | 2.197 | 2.147 | 2.323 | 2.151 | 2.158 |
| o4-mini | 2.150 | 2.155 | 2.139 | 2.131 | 2.142 | 2.277 | 2.146 | 2.153 |
| GPT-4o-mini | 2.077 | 2.082 | 2.066 | 2.061 | 2.069 | 2.184 | 2.073 | 2.080 |
| GPT-4o | 2.075 | 2.080 | 2.064 | 2.056 | 2.067 | 2.200 | 2.071 | 2.078 |

**(b) Success Rates by Theme (%)**

| Model | Algorithms | Data Structures | System Design | Security | Performance | Testing | Maintenance | Integration |
|---|---|---|---|---|---|---|---|---|
| Gemini-2.5-Pro | 99.88 | 99.88 | 99.88 | 99.90 | 99.88 | 99.90 | 99.88 | 99.88 |
| Gemini-2.5-Flash | 100.00 | 100.00 | 99.88 | 100.00 | 100.00 | 100.00 | 100.00 | 100.00 |
| gpt5mini | 100.00 | 100.00 | 100.00 | 100.00 | 100.00 | 100.00 | 100.00 | 100.00 |
| claudesonnet4 | 99.50 | 99.62 | 99.50 | 99.60 | 99.50 | 99.60 | 99.62 | 99.50 |
| GPT-5 | 100.00 | 100.00 | 100.00 | 100.00 | 100.00 | 100.00 | 100.00 | 100.00 |
| claudesonnet3.7 | 99.75 | 99.88 | 99.75 | 99.70 | 99.88 | 99.90 | 99.75 | 99.88 |
| GPT-4.1-mini | 100.00 | 100.00 | 100.00 | 100.00 | 100.00 | 100.00 | 100.00 | 100.00 |
| o3-mini | 100.00 | 100.00 | 100.00 | 100.00 | 100.00 | 100.00 | 100.00 | 100.00 |
| GPT-4.1-2025-04-14 | 100.00 | 100.00 | 100.00 | 100.00 | 100.00 | 100.00 | 100.00 | 100.00 |
| o3 | 100.00 | 100.00 | 100.00 | 100.00 | 100.00 | 100.00 | 100.00 | 100.00 |
| o4-mini | 99.75 | 99.75 | 99.62 | 99.50 | 99.75 | 99.70 | 99.75 | 99.75 |
| GPT-4o-mini | 100.00 | 100.00 | 100.00 | 100.00 | 100.00 | 100.00 | 100.00 | 100.00 |
| GPT-4o | 100.00 | 100.00 | 100.00 | 100.00 | 100.00 | 100.00 | 100.00 | 100.00 |

**(c) Software Engineering Scores by Theme**

| Model | Algorithms | Data Structures | System Design | Security | Performance | Testing | Maintenance | Integration |
|---|---|---|---|---|---|---|---|---|
| Gemini-2.5-Pro | 0.375 | 0.375 | 0.375 | 0.375 | 0.375 | 0.375 | 0.375 | 0.375 |
| Gemini-2.5-Flash | 0.373 | 0.373 | 0.373 | 0.373 | 0.373 | 0.373 | 0.373 | 0.373 |
| gpt5mini | 0.376 | 0.376 | 0.376 | 0.376 | 0.376 | 0.376 | 0.376 | 0.376 |
| claudesonnet4 | 0.379 | 0.379 | 0.379 | 0.379 | 0.379 | 0.379 | 0.379 | 0.379 |
| GPT-5 | 0.367 | 0.367 | 0.367 | 0.367 | 0.367 | 0.367 | 0.367 | 0.367 |
| claudesonnet3.7 | 0.377 | 0.377 | 0.377 | 0.377 | 0.377 | 0.377 | 0.377 | 0.377 |
| GPT-4.1-mini | 0.359 | 0.359 | 0.359 | 0.359 | 0.359 | 0.359 | 0.359 | 0.359 |
| o3-mini | 0.355 | 0.355 | 0.355 | 0.355 | 0.355 | 0.355 | 0.355 | 0.355 |
| GPT-4.1-2025-04-14 | 0.352 | 0.352 | 0.352 | 0.352 | 0.352 | 0.352 | 0.352 | 0.352 |
| o3 | 0.342 | 0.342 | 0.342 | 0.342 | 0.342 | 0.342 | 0.342 | 0.342 |
| o4-mini | 0.353 | 0.353 | 0.353 | 0.353 | 0.353 | 0.353 | 0.353 | 0.353 |
| GPT-4o-mini | 0.341 | 0.341 | 0.341 | 0.341 | 0.341 | 0.341 | 0.341 | 0.341 |
| GPT-4o | 0.339 | 0.339 | 0.339 | 0.339 | 0.339 | 0.339 | 0.339 | 0.339 |

**(d) Functional Scores by Theme**

| Model | Algorithms | Data Structures | System Design | Security | Performance | Testing | Maintenance | Integration |
|---|---|---|---|---|---|---|---|---|
| Gemini-2.5-Pro | 0.356 | 0.356 | 0.356 | 0.356 | 0.356 | 0.356 | 0.356 | 0.356 |
| Gemini-2.5-Flash | 0.358 | 0.358 | 0.358 | 0.358 | 0.358 | 0.358 | 0.358 | 0.358 |
| gpt5mini | 0.371 | 0.371 | 0.371 | 0.371 | 0.371 | 0.371 | 0.371 | 0.371 |
| claudesonnet4 | 0.348 | 0.348 | 0.348 | 0.348 | 0.348 | 0.348 | 0.348 | 0.348 |
| GPT-5 | 0.383 | 0.383 | 0.383 | 0.383 | 0.383 | 0.383 | 0.383 | 0.383 |
| claudesonnet3.7 | 0.347 | 0.347 | 0.347 | 0.347 | 0.347 | 0.347 | 0.347 | 0.347 |
| GPT-4.1-mini | 0.365 | 0.365 | 0.365 | 0.365 | 0.365 | 0.365 | 0.365 | 0.365 |
| o3-mini | 0.368 | 0.368 | 0.368 | 0.368 | 0.368 | 0.368 | 0.368 | 0.368 |
| GPT-4.1-2025-04-14 | 0.364 | 0.364 | 0.364 | 0.364 | 0.364 | 0.364 | 0.364 | 0.364 |
| o3 | 0.385 | 0.385 | 0.385 | 0.385 | 0.385 | 0.385 | 0.385 | 0.385 |
| o4-mini | 0.360 | 0.360 | 0.360 | 0.360 | 0.360 | 0.360 | 0.360 | 0.360 |
| GPT-4o-mini | 0.360 | 0.360 | 0.360 | 0.360 | 0.360 | 0.360 | 0.360 | 0.360 |
| GPT-4o | 0.362 | 0.362 | 0.362 | 0.362 | 0.362 | 0.362 | 0.362 | 0.362 |

Table 17: Detailed comparison of model performance by theme (continued).

**(e) Quality Scores by Theme**

| Model | Algorithms | Data Structures | System Design | Security | Performance | Testing | Maintenance | Integration |
|---|---|---|---|---|---|---|---|---|
| Gemini-2.5-Pro | 0.768 | 0.768 | 0.768 | 0.768 | 0.768 | 0.768 | 0.768 | 0.768 |
| Gemini-2.5-Flash | 0.741 | 0.741 | 0.741 | 0.741 | 0.741 | 0.741 | 0.741 | 0.741 |
| gpt5mini | 0.745 | 0.745 | 0.745 | 0.745 | 0.745 | 0.745 | 0.745 | 0.745 |
| claudesonnet4 | 0.762 | 0.762 | 0.762 | 0.762 | 0.762 | 0.762 | 0.762 | 0.762 |
| GPT-5 | 0.732 | 0.732 | 0.732 | 0.732 | 0.732 | 0.732 | 0.732 | 0.732 |
| claudesonnet3.7 | 0.773 | 0.773 | 0.773 | 0.773 | 0.773 | 0.773 | 0.773 | 0.773 |
| GPT-4.1-mini | 0.739 | 0.739 | 0.739 | 0.739 | 0.739 | 0.739 | 0.739 | 0.739 |
| o3-mini | 0.726 | 0.726 | 0.726 | 0.726 | 0.726 | 0.726 | 0.726 | 0.726 |
| GPT-4.1-2025-04-14 | 0.720 | 0.720 | 0.720 | 0.720 | 0.720 | 0.720 | 0.720 | 0.720 |
| o3 | 0.721 | 0.721 | 0.721 | 0.721 | 0.721 | 0.721 | 0.721 | 0.721 |
| o4-mini | 0.704 | 0.704 | 0.704 | 0.704 | 0.704 | 0.704 | 0.704 | 0.704 |
| GPT-4o-mini | 0.679 | 0.679 | 0.679 | 0.679 | 0.679 | 0.679 | 0.679 | 0.679 |
| GPT-4o | 0.677 | 0.677 | 0.677 | 0.677 | 0.677 | 0.677 | 0.677 | 0.677 |

**(f) Long-Context Scores by Theme**

| Model | Algorithms | Data Structures | System Design | Security | Performance | Testing | Maintenance | Integration |
|---|---|---|---|---|---|---|---|---|
| Gemini-2.5-Pro | 0.523 | 0.523 | 0.523 | 0.523 | 0.523 | 0.523 | 0.523 | 0.523 |
| Gemini-2.5-Flash | 0.565 | 0.565 | 0.565 | 0.565 | 0.565 | 0.565 | 0.565 | 0.565 |
| gpt5mini | 0.479 | 0.479 | 0.479 | 0.479 | 0.479 | 0.479 | 0.479 | 0.479 |
| claudesonnet4 | 0.492 | 0.492 | 0.492 | 0.492 | 0.492 | 0.492 | 0.492 | 0.492 |
| GPT-5 | 0.492 | 0.492 | 0.492 | 0.492 | 0.492 | 0.492 | 0.492 | 0.492 |
| claudesonnet3.7 | 0.477 | 0.477 | 0.477 | 0.477 | 0.477 | 0.477 | 0.477 | 0.477 |
| GPT-4.1-mini | 0.435 | 0.435 | 0.435 | 0.435 | 0.435 | 0.435 | 0.435 | 0.435 |
| o3-mini | 0.455 | 0.455 | 0.455 | 0.455 | 0.455 | 0.455 | 0.455 | 0.455 |
| GPT-4.1-2025-04-14 | 0.451 | 0.451 | 0.451 | 0.451 | 0.451 | 0.451 | 0.451 | 0.451 |
| o3 | 0.342 | 0.342 | 0.342 | 0.342 | 0.342 | 0.342 | 0.342 | 0.342 |
| o4-mini | 0.393 | 0.393 | 0.393 | 0.393 | 0.393 | 0.393 | 0.393 | 0.393 |
| GPT-4o-mini | 0.344 | 0.344 | 0.344 | 0.344 | 0.344 | 0.344 | 0.344 | 0.344 |
| GPT-4o | 0.348 | 0.348 | 0.348 | 0.348 | 0.348 | 0.348 | 0.348 | 0.348 |

