# OpenReview forum: "LoCoBench: A Benchmark for Long-Context Large Language Models in Complex Software Engineering"
_ICLR.cc/2026/Conference — ICLR 2026 Conference Desk Rejected Submission_

### Official Review · Reviewer_ASAo · 2025-10-17

**Soundness:** 2
**Presentation:** 2
**Contribution:** 3
**Rating:** 4
**Confidence:** 3

**Summary:**

This paper introduces LoCoBench, a benchmark designed to evaluate LLMs on complex software engineering tasks under long-context settings. The authors propose a detailed data synthesis pipeline that produces 1,000 projects and 8 task types, resulting in a total of 8,000 scenarios. They use LoCoBench to comprehensively evaluate several major closed-source models.

**Strengths:**

- Sound pipeline. The authors employ a constraint-satisfaction approach to synthesize 1,000 specifications, generate code bases from these specifications, and create 8 scenarios per code base. They also perform compilation- and execution-based dependency validation. The overall setup appears logical and appropriate.

- Substantial experimental effort. The authors conduct broad, multifaceted evaluations of leading closed-source models, differing from most existing benchmarks that primarily assess repository-level software engineering capabilities via unit test–based Pass@N metrics.

**Weaknesses:**

- While LoCoBench is a comprehensive code dataset, its connection to long-context scenarios is not clearly established. In other word, it is unclear what exactly the authors mean by "context". In datasets such as the SWE-bench series, “context” includes not only the problem statement but the entire target repository. Yet Table 1 lists SWE-bench contexts ranging only from 10 to 50K tokens—how were these numbers computed? In realistic settings, an LLM agent retrieves relevant parts of a repository (via dense/sparse retrieval or agentic search) rather than ingesting all code lines at once. The authors should formally define “context” and explain how it is measured to clarify the distinction between LoCoBench and prior work.

- The statement on line 64—“but not the long-context capabilities needed for realistic software development”—is inaccurate. On the one hand, as discussed in previous point, SWE-bench codebase can be quite large; On the other hand, SWE-bench issues are not limited to bug fixing but also include feature requests and other tasks (even thought its paper claim that they only choose bug-fix issues). The SWE-bench framework can already model a broad range of realistic development activities, especially as datasets such as SWE-bench Live continue to expand. Therefore, the authors need to clarify why they believe that SWE-bench (and its derivative datasets) do not represent the “long-context capabilities needed for realistic software development.”

- Although the paper emphasizes dataset scale, a larger or longer-context dataset does not necessarily imply higher quality. Even the 500-instance SWE-bench-verified dataset is already computationally demanding for LLM-agent evaluation. I am concerned that evaluating models on LoCoBench may require excessive computational and time resources.

- While synthetic data are common for scalable training, they are less suitable for evaluation. Synthetic tasks may embed biases from the generation process, making the evaluation potentially less representative of real-world software development. This limits the external validity of the results.

- The paper uses weights of 0.4/0.3/0.2/0.1 for computing the final score, but the rationale for this allocation is not explained. Since small changes in weights could significantly affect the ranking, a justification or sensitivity analysis is necessary to ensure interpretability.

- The authors emphasize “software architecture” in their task design, but it remains unclear why this aspect is critical for evaluating LLMs. Any insights or empirical findings on this point would be valuable.

I hope the authors can address these concerns. I will reconsider my score after rebuttal.

Typos:

- Line 166: “We target coverage ratios ¿0.7 for all scenarios.” — “¿” appears to be a corrupted character.

- Table 1: “Short (¡10K)” — “¡” also appears to be a corrupted character (appears twice).

**Questions:**

1. What is the definition of "context" in this paper? Why isn't SWE-bench series benchmark a long-context real-world software engineering benchmark?
2. Why does LoCoBench choose to use synthetic data?
3. Why does LoCoBench use weights of 0.4/0.3/0.2/0.1 for computing the final score?

---

### Official Review · Reviewer_XsAM · 2025-10-24

**Soundness:** 2
**Presentation:** 1
**Contribution:** 2
**Rating:** 2
**Confidence:** 2

**Summary:**

LoCoBench is a new benchmark for evaluating LLMs' long-context capabilities on realistic, large-scale software engineering tasks. It is constructed via a 5-phase pipeline that synthesizes 1,000 validated projects across 10 languages, yielding 8,000 scenarios spanning 10K–1M tokens per instance. The benchmark covers eight task categories including architectural understanding, cross-file refactoring, feature implementation, bug investigation, and multi-session development, designed to stress long-range reasoning across full codebases. LoCoBench introduces 17 evaluation metrics (including 6 new ones) and aggregates them into a unified LoCoBench Score (LCBS) to capture engineering quality, correctness, code quality, and long-context utilization. Experiments on leading LLMs show substantial performance gaps in long-context scenarios across languages, domains, and architecture patterns, establishing long-context software development as a persistent open challenge .

**Strengths:**

* The proposed dataset is of a large scale, and covers diverse software engineering domains and programming languages.
* The paper introduces 6 new evaluation metrics for a more comprehensive assessment of a model's coding capability.

**Weaknesses:**

* The paper does not provide sufficient technical details, making it difficult to assess the soundness of the benchmark. The 5-phase data construction pipeline only spans half a page in the main body. Even after reading Appendix E, I'm still not quite clear about a few key points. For example, (i) how were the specs generated initially? Were they crafted by humans or LLMs? How were the topics of projects determined?  (ii) The validation pipeline in E.2.2 does not guarantee any functional or executional correctness. Did you construct any runtime environment and execute the code?
* Similarly, the newly proposed evaluation metrics deserve more in-depth descriptions in the paper. Defining the associated tasks and terms (e.g., "architecture patterns") for each metric would be very helpful. In addition, it is not clear how difficult it will be for practitioners to calculate those metrics when evaluating their own models.

**Questions:**

* Typo in L166

---

### Official Review · Reviewer_QZvy · 2025-10-30

**Soundness:** 1
**Presentation:** 1
**Contribution:** 2
**Rating:** 2
**Confidence:** 4

**Summary:**

This paper proposes LoCoBench, a benchmark of synthesized software projects which can be used to assess how well large language models handle large contexts.

The paper mentions a five stage (generating specs, code, and evaluation scenarios, and conducting Q& and LLM scoring) pipeline to create LoCoBench.

LoCoBench consists of 8000 tasks in 1000 projects in 10 different languages, divided over 36 application domains. The projects are synthesized in such a way that there is variance in, e.g., number of files, file sizes, cyclomatic complexity, etc.

The benchmark is used to rank 13 LLMs (from the Gemin, GPT, and Claude families)

**Strengths:**

- The paper represents an impressive amount of work
- Understanding the limits of context size and the impact this has on LLM performance is a relevant problem, certainly in the area of LLMs for code.

**Weaknesses:**

A key problem with this paper is that it contains an overwhelming amount of information, without giving the high level information needed to understand the relevance of all this information.

Crucial information about, e.g., the generation process (specs are created through constraint satisfaction, E.1.1) are postponed to the appendix, or not included at all (e.g., is an LLM used to generate the code in E.2.1 or are classical program generators used?).

Also, the metrics introduced are hailed as a key contribution, but are not even defined. As an example, the paper mentions an Integration Test Performance metric. But then it just refers to a book by Binder, which is over 1000 pages long. It is unclear what exactly is measured here. (And there are 1000s of ways in which one could attempt to measure "integration test performance" -- assuming the test suite indeed contains an integration test suite). The appendix offers no clue either, just stating "System-wide functionality assessment, based on established integration testing frameworks", which is meaningless.

The paper is proud that it includes 6 new evaluation metrics, but I would find it much better if existing ones could have been reused. The need for these metrics is insufficiently explained.

Besides failing to provide a clear conceptual overview, the paper also suffers from trying to include more and more and more. Why 10 languages? Why 8 tasks? A better approach would be to design the smalles possible benchmark that would require LLMs to work with large contexts.

Related to all this is the fact that the objective of the paper is not clear. Appearantly, the benchmark can be used to rank LLMs. But why is this benchmark better for ranking than other benchmarks?

In Table 1 a comparison with other benchmarks is made. This table suggests that LoCoBench is "real world". This seems false to me. Perhaps the tasks are inspired by real world tasks. But the generated code is completely artificial, and so are the tasks. In fact, a more relevant column would be whether the code is synthesized or real, with LoCoBench being synthesized, and various of the others real.

The lack of realism of LoCoBench casts doubt about the relevance of the results. Here the lack of realism should be offset by benefits in increased control, permitting assessments of the characteristics of the projects vs LLM performance. This may be the true goal of the paper, but this is not stated anywhere.

**Questions:**

- What is the goal of LoCoBench?
- Can you provide details on the codebase generation step
- Can you provide details on the "Integration Test Performance" metric

---

### Official Review · Reviewer_JMvm · 2025-11-01

**Soundness:** 3
**Presentation:** 2
**Contribution:** 3
**Rating:** 6
**Confidence:** 3

**Summary:**

This paper introduces LoCoBench, which is designed to evaluate LLMs capability in solving long-context software development tasks. The benchmark is synthesized following 5-phase pipeline and generates 8000 scenarios across 10 languages. The authors also introduces 6 new evaluation metrics within 17 comprehensive metrics. Experiments are conducted on major LLMs across 10 languages, suggesting a significant variance across programming languages, domains, and architectural patterns and showing a gap in long-context and multi-file reasoning.

**Strengths:**

1. LoCoBench has a large scale of long-context software engineering task instances, covering 8000 scenarios, 10 languages and 10K to 1M tokens.

2. LoCoBench involves 6 new long-context metrics in evaluation. It also introduces comprehensive evaluation metrics to evaluate LLMs in different dimensions, providing a rich insight on the LLM behaviors.

3. Besides the comprehensive detailed evaluation, LoCoBench also provide an aggregated score to intuitively exhibit models capacity and compare major LLMs' performance in various degrees, pointing out their subtle design priorities.

**Weaknesses:**

1. While the benchmark includes a sufficient number of instances across multiple domains and subdomains, the projects and scenarios are generated by LLMs rather than derived from real-world repositories. Moreover, the data synthesis pipeline is described only at a high level and lacks clarity. The paper does not provide sufficient details on how the authors automatically ensure specification quality, especially given that the quality constraints described in the appendix appear to be somewhat subjective.

2. The authors also provide limited information regarding manual quality checks or execution validation procedures applied to these model-generated projects. Consequently, it remains unclear whether the synthesized projects are reliable or if they exhibit artificial characteristics that diverge from real repositories and may bias the evaluation results.

3. The authors introduce the final LoCoBench Score as a specific weighted sum of four dimensions. However, the paper only briefly states that these weights are “empirically determined to reflect relative importance,” without describing how this empirical determination was done. Additionally, within each dimension, the metrics vary, yet their scores are computed simply as averages after normalization. It remains unclear why to use averaging within each dimension and how the weights were determined.

**Questions:**

1. Can the authors further explain about their validation approach applied? Have they conducted any manual validation to check the synthesized quality?

2. Can the authors justify the specific weighting in the LoCoBench Score? It would be helpful if the authors could explain more on why they chose to use averaging within each dimension and how to decide the final weights for each dimension.

3. In Figure 10 (Top), the performance scores across architecture patterns are presented using a line chart. Could the authors clarify why a line chart was chosen instead of a bar chart? Does the depicted trend reflect any underlying features or relationships among the patterns?

4. A typo: in Page 9: It provides findingds -> findings

---

### Note · Program_Chairs · 2026-01-17
**Submission Desk Rejected by Program Chairs**

The following references in this submission do not refer to real documents and/or have major errors in bibliographic information:

 G Ann Campbell and Patroklos P Papapetrou. Defects in agile software development: an industrial perspective. In Empirical Software Engineering, volume 18, pp. 1077-1096. Springer, 2013.
Yutao Dong, Bowen Jiang, Yaojie Chen, Hongzheng Dai, Xingjian Yao, Ming Chen, and Yongji Zhang. Effibench: Benchmarking the efficiency of code generation. In arXiv preprint arXiv:2402.02991, 2024a.